# Planar chlorination engineering induced symmetry-broken single-atom site catalyst for enhanced $CO_2$ electroreduction

Shengjie Wei [1,2], Jiexin Zhu [3,4] ✉, Xingbao Chen[4], Rongyan Yang[5], Kailong Gu[3], Lei Li [6] ✉, Ching-Yu Chiang [7] ✉, Liqiang Mai [4] ✉ & Shenghua Chen[3] ✉

Breaking the geometric symmetry of traditional metal-$N_4$ sites and further boosting catalytic activity are significant but challenging. Herein, planar chlorination engineering is proposed for successfully converting the traditional Zn-$N_4$ site with low activity and selectivity for $CO_2$ reduction reaction ($CO_2RR$) into highly active Zn-$N_3$ site with broken symmetry. The optimal catalyst Zn-SA/CNCl-1000 displays a highest faradaic efficiency for CO ($FE_{CO}$) around $97 \pm 3\%$ and good stability during 50 h test at high current density of 200 mA/cm² in zero-gap membrane electrode assembly (MEA) electrolyzer, with promising application in industrial catalysis. At -0.93 V vs. RHE, the partial current density of CO ($J_{CO}$) and the turnover frequency (TOF) value catalyzed by Zn-SA/CNCl-1000 are $271.7 \pm 1.4$ mA/cm² and $29325 \pm 151$ h⁻¹, as high as 29 times and 83 times those of Zn-SA/CN-1000 without planar chlorination engineering. The in-situ extended X-ray absorption fine structure (EXAFS) measurements and density functional theory (DFT) calculation reveal the adjacent C-Cl bond induces the self-reconstruction of Zn-$N_4$ site into the highly active Zn-$N_3$ sites with broken symmetry, strengthening the adsorption of *COOH intermediate, and thus remarkably improving $CO_2RR$ activity.

Electrochemical $CO_2RR$ is an effective and environmental-friendly route to convert $CO_2$ into hydrocarbon fuels, which is appealing to alleviate the greenhouse effect and the energy shortage[1-6]. The CO as one of the important products during 2e-reduction of $CO_2$, has been widely utilized in Fischer-Tropsch synthesis for high-value hydrocarbons[7,8]. During the 2e-reduction of $CO_2$ into CO, effectively strengthening the adsorption of key *COOH intermediate and lowering

the energy barrier of protonation process are the key issues to improve catalytic activity. Besides, effectively suppressing the competing hydrogen evolution reaction (HER), achieving high $FE_{CO}$ within wide potential range and high current density are main challenges for $CO_2RR$[9].

Metal isolated single-atom site (ISAS) catalysts as the frontier in heterogeneous catalysis, have aroused much research interests due to

[1]Center Excellence for Environmental Safety and Biological Effects, Beijing Key Laboratory for Green Catalysis and Separation, Department of Chemistry, College of Chemistry and Life Science, Beijing University of Technology, Beijing 100124, China. [2]School of Materials Science and Engineering, Nankai University, Tianjin 300350, P. R. China. [3]National Innovation Platform (Center) for Industry-Education Integration of Energy Storage Technology, School of Chemical Engineering and Technology, Xi'an Jiaotong University, Xi'an 710049, P. R. China. [4]State Key Laboratory of Advanced Technology for Materials Synthesis and Processing, Wuhan University of Technology, Wuhan 430070 Hubei, P. R. China. [5]Key Laboratory of Pollution Processes and Environmental Criteria of Ministry of Education, Tianjin Key Laboratory of Environmental Remediation and Pollution Control, College of Environmental Science and Engineering of Nankai University, Tianjin 300350, P. R. China. [6]Hefei National Research Center for Physical Sciences at the Microscale, University of Science and Technology of China, Hefei, Anhui 230026, P. R. China. [7]National Synchrotron Radiation Research Center, Hsinchu 30076, Taiwan. ✉e-mail: jxzhu@whut.edu.cn; leili@mail.ustc.edu.cn; chiang.cy@nsrrc.org.tw; mlq518@whut.edu.cn; shenghchen@xjtu.edu.cn

their utmost atomic utilization efficiency and good catalytic performance[10–15]. Among them, metal-$N_4$ sites similar to the metalloporphyrin complex anchored on N-doped carbon catalysts (metal-$N_4$/CN) with good structural stability have been extensively utilized for $CO_2$RR[9,16–27]. However, the metal-$N_4$/CN catalysts have the planar-like $D_{4h}$ symmetry and symmetric charge distribution, bringing the limitation for electronic configuration regulation of catalytic site[28] and axial adsorption of the intermediates[29]. Therefore, their catalytic activities are promising to be further improved by breaking the planar-like $D_{4h}$ symmetry of metal-$N_4$ sites, redistributing the electronic distribution, optimizing the orbital hybridization interaction, strengthening the axial adsorption of the intermediates for $CO_2$RR.

Recently, the reported atomic-level chlorination engineering of metal-$N_4$/CN catalysts for boosting electro-catalytic activities mainly focused on axial chlorination engineering by introducing axial Cl coordinating atom on metal-$N_4$ sites with $C_{2v}$ symmetry[30–34], which limited the axial adsorption of reactants and did not thoroughly break the geometric symmetry of bonding and electronic distribution as exhibited in Supplementary Fig. 1. By comparison, developing the planar chlorination engineering is promising to thoroughly break the planar-like $D_{4h}$ symmetry of metal-$N_4$ sites and to induce their self-reconstruction into highly active sites for boosting $CO_2$RR activity.

Herein, the planar chlorination engineering of metal ISAS catalysts was proposed and designed for successfully converting the traditional Zn-$N_4$ site with low activity and selectivity for $CO_2$RR into highly active Zn-$N_3$ site with broken geometric symmetry. The optimal catalyst Zn-SA/CNCl-1000 had $FE_{CO}$ above 90% over a broad potential window from -0.63 V to -0.93 V vs. RHE, with the maximum $FE_{CO}$ of 97.5 ± 3.0% at −0.63 V vs. RHE, much higher than that of Zn-SA/CN-1000 catalyst without planar chlorination engineering (the maximum $FE_{CO}$ of 30.5 ± 1.0% at −0.75 V vs. RHE). The $J_{CO}$ and TOF value catalyzed by Zn-SA/CNCl-1000 were 271.7 ± 1.4 mA/cm$^2$ and 29325 ± 151 h$^{-1}$ at −0.93 V vs. RHE, around 29 times and 83 times those of Zn-SA/CN-1000 without Cl-doping, demonstrating that the planar chlorination engineering of Zn-$N_4$ sites remarkably improved both catalytic activity and selectivity for $CO_2$RR. Besides, the Zn-SA/CNCl-1000 also exhibited stability during 50 h test at high current density of 200 mA/cm$^2$ with the potential in industrial application. The in-situ EXAFS measurements and DFT calculation revealed the adjacent C-Cl bond induced the self-reconstruction of Zn-$N_4$ site into highly active Zn-$N_3$ catalytic site, breaking the planar-like $D_{4h}$ symmetry of Zn-$N_4$ site, strengthening the adsorption of key *COOH intermediate, and remarkably improving $CO_2$RR activity and selectivity.

## Results

### Synthesis and characterization of Zn-SA/CNCl catalysts

As shown in Fig. 1a, we utilized the NaCl-co-pyrolysis strategy during the synthesis of Zn-SA/CNCl catalysts, which was similar to the reported work[35]. The Zeolite Imidazole Framework-8 (ZIF-8) and NaCl were pyrolyzed together under argon atmosphere for 3 h, during which the NaCl as the chlorine source in a separate ceramic boat was placed in the upstream direction of ZIF-8 powder, as shown in Supplementary Fig. 2. The tetrahedral Zn-$N_4$ sites from ZIF-8 gradually evolved into planar Zn-$N_4$ sites at elevated temperatures. When the pyrolysis temperature above the melting point of NaCl (801ºC), the volatile Cl species from the melted NaCl evaporated and was captured by Zn-$N_4$/CN catalysts, with the formation of Zn-$N_4Cl_1$ sites and C-Cl bond. As the pyrolysis temperatures increased from 850 °C to 1000 °C, the Zn-Cl bond of Zn-$N_4Cl_1$ sites was gradually broken with the formation of Zn-$N_4$ sites and successful introduction of C-Cl bonds adjacent to the planar Zn-$N_4$ sites by planar chlorination engineering. We characterized the Zn-SA/CNCl catalysts by the high angle annular dark field scanning transmission electron microscopy (HAADF-STEM) images in Fig. 1b, and Supplementary Figs. 3 and 4. No Zn or ZnO nanoparticles were found in the Zn-SA/CNCl catalysts. The corresponding

energy dispersive X-ray (EDX) spectroscopy elemental mapping of Zn-SA/CNCl catalysts in Fig. 1c, and Supplementary Figs. 3 and 4 revealed the successful introduction of Cl element and the homogeneous distribution of C, N, Zn and Cl elements on the Zn-SA/CNCl catalysts. We directly observed the Zn ISASs were atomically dispersed on the Zn-SA/CNCl-1000 by aberration-corrected scanning transmission electron microscopy (AC-STEM) in Fig. 1d. By comparison, the Zn-SA/CN-1000 as reference sample without Cl-doping was synthesized by pyrolysis of pure ZIF-8 at 1000 °C under argon atmosphere. As shown in Supplementary Figs. 5 and 6, the HAADF-STEM image and corresponding EDX spectroscopy elemental mapping of Zn-SA/CN-1000 confirmed the absence of Zn-based nanoparticles and Cl element. The Zn contents of Zn-based samples were determined by inductively coupled plasma optical emission spectrometry (ICP-OES) measurements. The Zn contents of Zn-SA/CNCl-850, Zn-SA/CNCl-920, Zn-SA/CNCl-1000 and Zn-SA/CN-1000 were 4.23 wt%, 2.65 wt%, 1.13 wt% and 3.24 wt%, respectively. We measured the X-ray diffraction (XRD) patterns of Zn-based samples in Supplementary Fig. 7, which confirmed the absence of Zn or ZnO. There were only two broad peaks of the characteristic carbon (002) and (100)/(101) diffractions, which located at around 25° and 44°, respectively. To analyze the elemental compositions, the X-ray photoelectron spectroscopy (XPS) measurements of Zn-based samples were performed, as shown in Fig. 1e–g, Supplementary Figs. 8–11 and Supplementary Table 1. The N 1$s$ spectra were composed of pyridinic N, Zn-N, pyrrolic N, graphitic N and oxidized N species. The Cl 2$p$ spectra of Zn-SA/CNCl catalysts in Fig. 1e–g were composed of covalent Cl from C-Cl bond and ionic Cl from Zn-Cl bond. As the pyrolysis temperatures increasing from 850 °C to 1000 °C, the ratios of ionic Cl gradually decreased from 56.1% to 6.9%, indicating that higher temperatures were advantageous to break Zn-Cl bond. The Brunauer–Emmett–Teller (BET) surface areas of Zn-based samples were determined by nitrogen sorption isotherm experiments in Supplementary Figs. 12–15. The BET surface areas of Zn-SA/CNCl-850, Zn-SA/CNCl-920, Zn-SA/CNCl-1000 and Zn-SA/CN-1000 were 738, 999, 998 and 768 m$^2$/g, respectively. We measured the Raman spectra of Zn-based samples in Supplementary Fig. 16. The two prominent peaks at around 1340 cm$^{-1}$ (D peak) and 1580 cm$^{-1}$ (G peak) reflected the degree of disorder and graphitization of carbon substrates. The Zn-based samples had similar ratios between the intensities of D peak and G peak ($I_D$: $I_G$), indicating the similar structure of carbon substrates.

To further analyze the coordination environment of Zn-SA/CNCl catalysts at atomic level, we carried out the X-ray absorption near-edge structure (XANES) and extended X-ray absorption fine structure (EXAFS) measurements at Zn K-edge. We compared the normalized XANES curves of Zn-SA/CNCl catalysts, Zn foil and ZnO at Zn K-edge in Fig. 2a. The near-edge absorption of Zn-SA/CNCl catalysts located between those of Zn foil and ZnO, indicating that the Zn sites from Zn-SA/CNCl catalysts carried positive charges. As shown in Fig. 2b, we analyzed the corresponding $k^3$-weighted Fourier transform EXAFS (FT-EXAFS) curves in $R$ space without correction of radical distance phase. For Zn foil, there was a dominant peak at around 2.2 Å assigned to Zn-Zn bond. For ZnO, there were two prominent peaks at around 1.5 Å and 2.9 Å, assigned to the Zn-O pathway and Zn-O-Zn pathway, respectively. By contrast, there was only one dominant peak at around 1.5 Å in Zn-SA/CNCl catalysts, revealing the sole existence of Zn ISAS without formation of Zn or ZnO. As shown in Supplementary Figs. 17–19 and Supplementary Tables 2–4, we constructed three models for EXAFS fitting of Zn-based samples. Due to the stronger contribution of Zn-N path in the first coordination shell of Zn, obtaining the accurate coordination number of Zn-Cl bond is challenging by EXAFS fitting at Zn-K edge. As exhibited in Supplementary Figs. 20 and 21 and Supplementary Table 5, we analyzed and fitted the second coordination shell of Zn-SA/CNCl catalysts, revealing the main contribution of the second coordination shell was from Zn-C path.

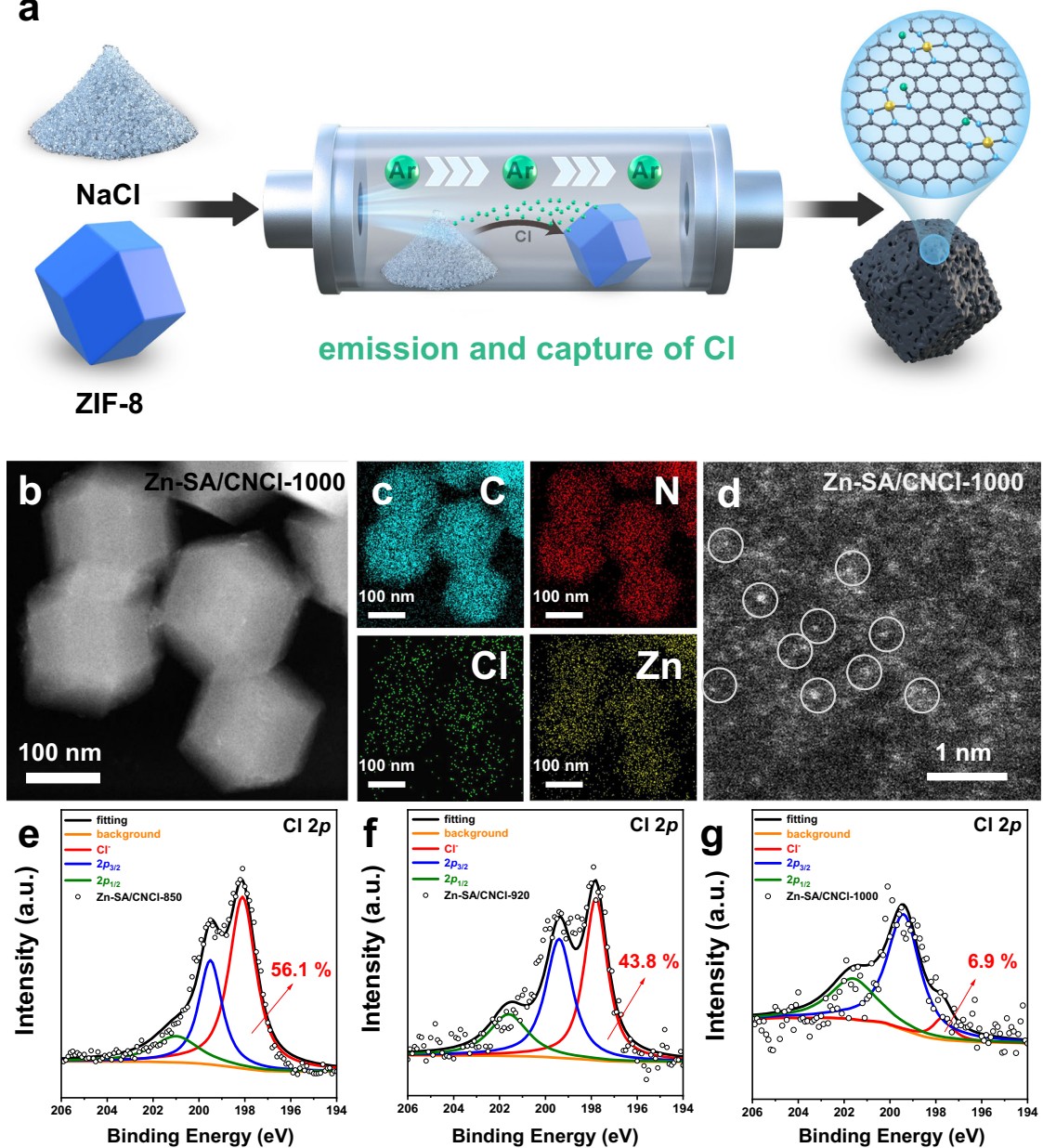

**Fig. 1 | The synthetic procedure and characterization of Zn-SA/CNCl catalysts.** **a** The schematic illustration of NaCl-co-pyrolysis strategy, which was similar to the synthetic method reported by the ref. 35. The gray, blue, yellow and green balls represented C, N, Zn and Cl atoms, respectively. **b, c** The HAADF-STEM image of Zn-SA/CNCl-1000 and the corresponding EDX spectroscopy elemental mapping results. **d** The AC-STEM image of Zn-SA/CNCl−1000. Only partial Zn ISASs were marked by white circles. **e–g** The XPS spectra for the Cl 2*p* of Zn-SA/CNCl-850, Zn-SA/CNCl-920 and Zn-SA/CNCl-1000, respectively.

In order to study the existence form of Cl element in the Zn-SA/CNCl catalysts, we carried out the EXAFS measurement at Cl K-edge, which could effectively eliminate the effect of the Zn-N bond. The XANES spectra of Zn-SA/CNCl catalysts at Cl K-edge were shown in Supplementary Fig. 22 and the corresponding FT-EXAFS results in *R* space were exhibited in Fig. 2c. As shown in Fig. 2c, the two prominent peaks at around 1.2 Å and 2.1 Å were assigned to the Cl-C path and Cl-Zn path, respectively. The two prominent Cl-C and Cl-Zn paths co-existed in Zn-SA/CNCl-850 and Zn-SA/CNCl-920. While the intensity of Cl-Zn path in Zn-SA/CNCl-1000 had a noticeable decline, compared with those of Zn-SA/CNCl-850 and Zn-SA/CNCl-920. By contrast, the intensity of Cl-C path in Zn-SA/CNCl-1000 was similar to those of Zn-SA/CNCl-850 and Zn-SA/CNCl-920. These results revealed the gradual transformation from the Zn-$N_4Cl_1$ site into Zn-$N_4$ site as the pyrolysis

temperatures of Zn-SA/CNCl samples increasing from 850 °C to 1000 °C. As the pyrolysis temperatures increasing from 850 °C to 1000 °C, the intensity of Cl-Zn path gradually decreased and the peak location of Cl-Zn path gradually moved toward higher *R* space, revealing that the Zn-Cl bonds were gradually elongated and finally were broken, which was attributed to the gradual transformation from the Zn-$N_4Cl_1$ site into Zn-$N_4$ site. By contrast, the intensity of Cl-C path had no obvious changes as the pyrolysis temperatures increasing from 850 °C to 1000 °C, indicating the C-Cl bond was much stronger than Zn-Cl bond under elevated temperatures. The fitting results of Zn-SA/CNCl samples at Cl K-edge were exhibited in Supplementary Fig. 23 and Supplementary Table 6.

The WT analysis of Zn-SA/CNCl-850, Zn-SA/CNCl-920 and Zn-SA/CNCl-1000 at Cl-K edge were compared in Fig. 2d–f. The prominent

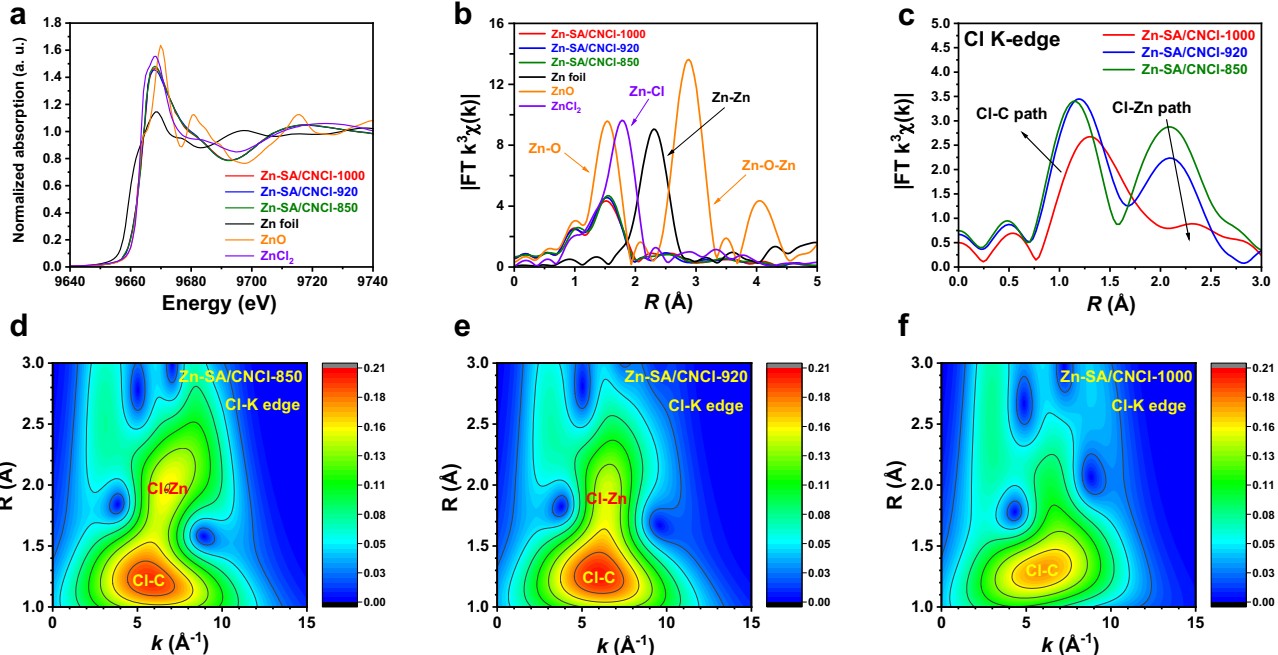

**Fig. 2 | The characterization of Zn-SA/CNCl catalysts by XANES and EXAFS measurements at Zn K-edge and Cl K-edge. a**, **b** The XANES spectra and corresponding FT-EXAFS results in *R* space of Zn-SA/CNCl-850, Zn-SA/CNCl-920, Zn-SA/CNCl-1000, Zn foil, ZnO and ZnCl₂ as reference samples at Zn K-edge. **c** The FT-EXAFS results in *R* space of Zn-SA/CNCl-850, Zn-SA/CNCl-920, and Zn-SA/CNCl-1000 at Cl K-edge. **d**–**f** The WT analysis of Zn-SA/CNCl-850, Zn-SA/CNCl-920 and Zn-SA/CNCl-1000 at Cl K-edge, respectively.

peak at around 6 Å⁻¹ in *k* space and 1.2 Å in *R* space was assigned to the Cl-C path. While the prominent peak at around 7 Å⁻¹ in *k* space and 2.0 Å in *R* space was assigned to the Cl-Zn path. Obviously, the Cl-C path and Cl-Zn path co-existed in the Zn-SA/CNCl-850 and Zn-SA/CNCl-920. As the pyrolysis temperatures gradually increasing from 850 °C to 1000 °C, the peak intensity of Cl-Zn path had a noticeable decay, revealing the breakage of Zn-Cl bond during the gradual transformation from Zn-N₄Cl₁ sites into Zn-N₄ sites. As shown in Fig. 2f, the Zn-SA/CNCl-1000 had a prominent peak of Cl-C path while the intensity of Cl-Zn path was rather weak. Therefore, by combination of EXAFS measurement at Cl K-edge and XPS measurement for Cl 2*p*, we concluded that the C-Cl bond and Zn-Cl bond co-existed in Zn-SA/CNCl-850 and Zn-SA/CNCl-920 samples. While the Cl element mainly existed as C-Cl bond in the Zn-SA/CNCl-1000 sample.

We summarized the evolution of catalytic sites in Supplementary Fig. 24. The C-Cl bond, Zn-N₄Cl₁ and Zn-N₄ sites co-existed in Zn-SA/CNCl-850 and Zn-SA/CNCl-920. As the pyrolysis temperature gradually increasing from 850 °C to 1000 °C, the Zn-Cl bond was gradually broken, with the co-existence of C-Cl bond and Zn-N₄ site. As shown in Supplementary Fig. 25, we also simulated the theoretical XANES curve of the optimized model of Zn-SA/CNCl-1000 in Supplementary Fig. 24, revealing the model of Zn-N₄ site with adjacent C-Cl bond was a rational model to simulate the Zn-SA/CNCl-1000. As shown in Supplementary Figure 26, in order to study the effect of Cl element for simulating XANES spectra, we simulated the theoretical XANES spectra of Zn-N₄Cl/CNCl-1 and Zn-N₄/CNCl-1 models in Supplementary Fig. 24, revealing the gradual transformation from the Zn-N₄Cl₁ site into the Zn-N₄ site did not induce remarkable change of XANES spectrum at Zn-K edge. The fitting results of the first and second coordination shells of Zn-SA/CN-1000 were shown in Supplementary Figs. 27 and 28 and Supplementary Table 7, revealing the existence of Zn-N₄ site. Therefore, we achieved the evolution of Zn-N₄Cl₁ site into Zn-N₄ site with C-Cl bond by regulating the pyrolysis temperatures, i.e., the transformation from the axial chlorination engineering into the planar chlorination engineering of Zn-N₄ ISAS.

## The catalytic performance for CO₂RR of Zn-based catalysts

The catalytic performance for CO₂RR catalyzed by Zn-based catalysts was investigated in an alkaline flow cell electrolyzer with 1 M KOH electrolyte[6]. We compared the total current density for CO₂RR in Supplementary Fig. 29. The Zn-SA/CNCl-1000 exhibited the highest total current density. The total current densities catalyzed by Zn-SA/CNCl catalysts increased gradually as the pyrolysis temperature increasing, and were higher than that of Zn-SA/CN-1000 catalyst. The comparison of linear sweep voltammetry (LSV) curves in flow cell electrolyzer with flowing Ar or CO₂ were exhibited in Supplementary Fig. 30, indicating the improved current densities were attributed to CO₂RR. Only CO and H₂ as the gaseous products were detected by gas chromatography (GC). No liquid products were found which was measured by ¹H nuclear magnetic resonance (NMR) spectroscopy in Supplementary Fig. 31. The corresponding FE_CO were compared in Fig. 3a. The Zn-SA/CN-1000 had the lowest FE_CO, with the maximum FE_CO of 30.5 ± 1.0% at −0.75 V vs. RHE. The FE_CO catalyzed by Zn-SA/CNCl-850 and Zn-SA/CNCl-920 were similar, fluctuating between 45% to 60% from −0.57 V to −0.87 V vs. RHE. By comparison, the Zn-SA/CNCl-1000 catalyst displayed FE_CO above 90% over a broad potential window from −0.63 V to −0.93 V vs. RHE, with the maximum FE_CO of 97.5 ± 3.0% at −0.63 V vs. RHE. The J_CO at different potentials was compared in Fig. 3b. The Zn-SA/CN-1000 exhibited poor CO₂RR activity, with the maximum J_CO of 22.3 ± 3.5 mA/cm² at −0.81 V vs. RHE. By comparison, after introducing planar chlorination engineering, the Zn-SA/CNCl-1000 catalyst exhibited much improved J_CO than Zn-SA/CN-1000 catalyst, with the maximum J_CO of 271.7 ± 1.4 mA/cm² at −0.93 V vs. RHE, which were 28.9 times, 5.7 times and 4.0 times those of Zn-SA/CN-1000, Zn-SA/CNCl-850 and Zn-SA/CNCl-920 catalysts, respectively. These results confirmed that the planar chlorination engineering of Zn-N₄ sites effectively boosted the activity and selectivity for CO₂RR. As the Zn ISASs were gradually evolved from Zn-N₄Cl₁ sites into Zn-N₄ sites, the J_CO and FE_CO improved correspondingly, revealing the superior regulation of planar chlorination engineering than axial chlorination engineering for improving CO₂RR activity on

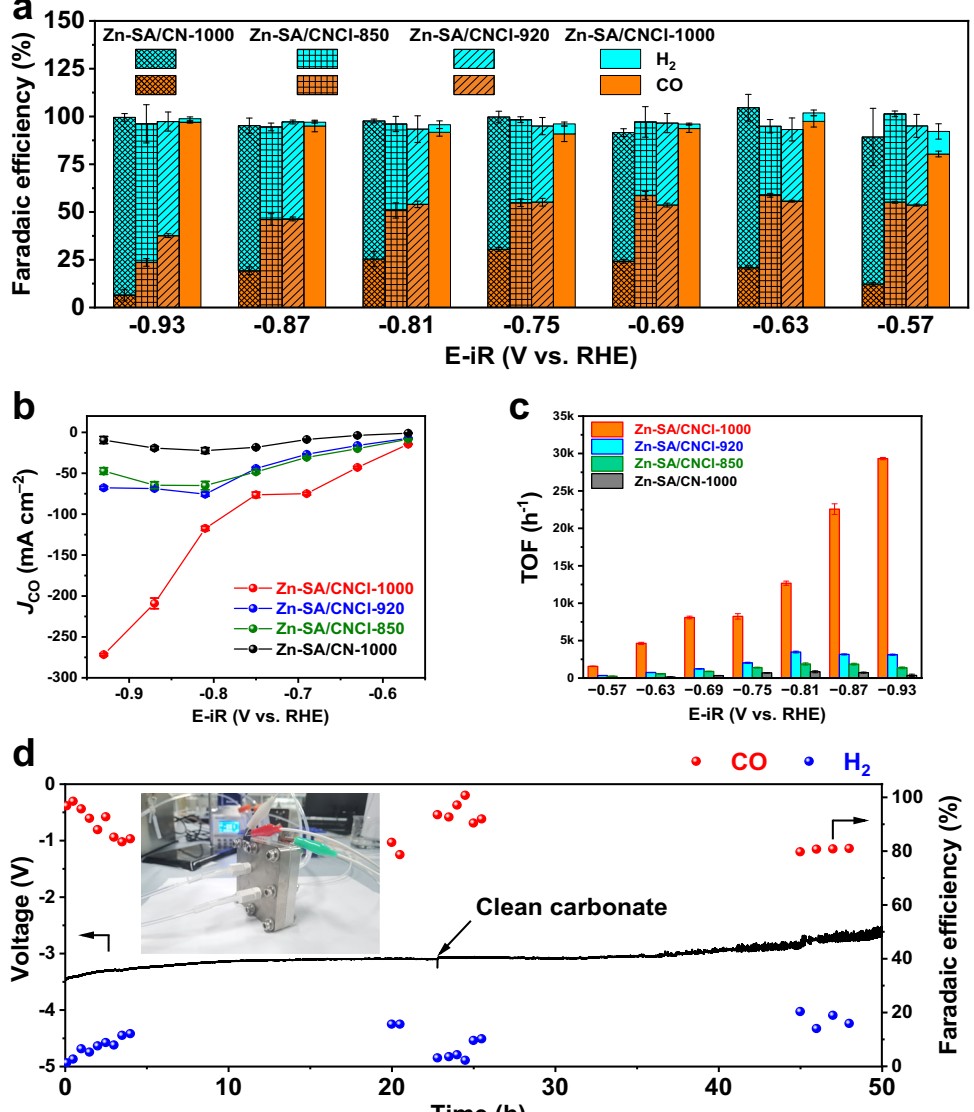

**Fig. 3 | The catalytic performance for CO₂RR of Zn-based catalysts. a** The comparison of FE$_{CO}$ at different potentials in 1 M KOH (pH = 14.0 ± 0.2). **b** The comparison of J$_{CO}$ during CO₂RR. **c** The comparison of TOF values. (The potentials in Fig. 3a–c are provided after iR correction. The measured electrolyte resistance was 1.0 ± 0.15 Ω. The mass loading of electrode is 1 mg cm⁻². The CO₂ flow rate was set to 50 sccm). **d** The stability test of Zn-SA/CNCl-1000 catalyst during 50 h at current density of 200 mA/cm² in 0.5 M KHCO₃ (pH = 8.3 ± 0.3) in zero-gap membrane flow reactor. The inset is photograph of membrane electrode assembly (MEA). All the tests were performed at room temperature. The error bars in Fig. 3 represent s.d. obtained from three independent experiments.

Zn-N₄ sites. We compared the TOF values in Fig. 3c. At −0.93 V vs. RHE, the TOF value catalyzed by Zn-SA/CNCl-1000 was 29325 ± 151 h⁻¹, which were 82.8 times, 21.5 times and 9.4 times those of Zn-SA/CN-1000 (354.0 ± 162.6 h⁻¹), Zn-SA/CNCl-850 (1364.0 ± 115.3 h⁻¹) and Zn-SA/CNCl-920 (3118.5 ± 82.8 h⁻¹) catalysts, respectively, demonstrating the remarkable advantage of planar chlorination engineering of metal ISAS catalysts. As shown in Supplementary Table 8, compared with other reported catalysts for CO₂RR into CO in flow cell, the Zn-SA/CNCl-1000 catalyst exhibited comparable CO₂RR activity in the J$_{CO}$ and FE$_{CO}$. As shown in Fig. 3d, we tested the stability of Zn-SA/CNCl-1000 at high current density of 200 mA/cm² in zero-gap membrane electrode assembly (MEA) reactor[6]. The photograph of MEA was shown in Supplementary Fig. 32. After catalysis for 50 h, the FE$_{CO}$ was above 80%, demonstrating the good stability of Zn-SA/CNCl-1000 at high current density. After CO₂RR, the Zn-SA/CNCl-1000 was characterized by HAADF-STEM image and AC-STEM image in Supplementary Fig. 33, confirming the atomic dispersion of Zn element of Zn-SA/CNCl-1000 after catalysis. The XPS spectra of Zn-SA/CNCl-1000 after CO₂RR in

Supplementary Fig. 34 exhibited the existence of covalent Cl from C-Cl bond, indicating the good structural stability of Zn-SA/CNCl-1000 after CO₂RR.

We compared the electrochemical active surface area (ECSA) of Zn-based catalysts in Supplementary Fig. 35. All the Zn-based catalysts showed similar ECSA, indicating that the improved electrochemical performances came from the optimization of the intrinsic activity of the active sites. The in-situ attenuated total reflection surface-enhanced infrared adsorption spectroscopy (ATR-SEIRAS) is an effective technique to detect the key intermediates for CO₂RR. We performed the in-situ ATR-SEIRAS measurement for CO₂RR catalyzed by Zn-SA/CNCl-1000 in Supplementary Fig. 36. The two prominent peaks at around 1720 cm⁻¹ and 1215 cm⁻¹ are attributed to the stretching vibrations of C=O bond and C-OH bond from *COOH intermediate, respectively[36,37]. The signal of *COOH appeared at −0.3 V vs. RHE, and the intensity of *COOH signal gradually increased as the potential decreased from −0.6 V to −1.2 V vs. RHE, indicating that the Zn-SA/CNCl-1000 catalyst effectively facilitated the formation of *COOH

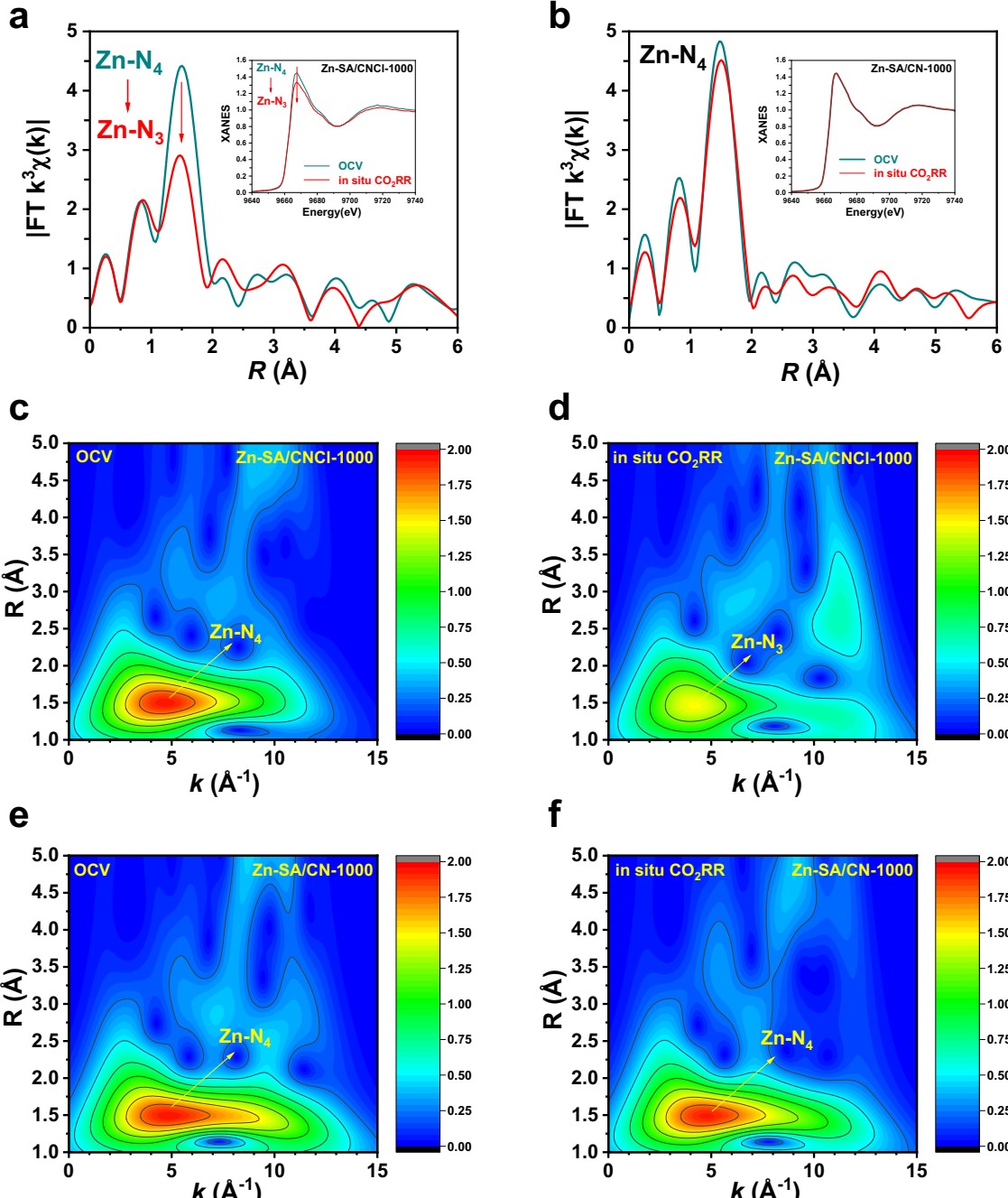

**Fig. 4 | The in-situ XANES and EXAFS measurements for CO₂RR. a, b** The FT-EXAFS spectra in *R* space and the XANES spectra (inset) for CO₂RR catalyzed by Zn-SA/CNCl-1000 and Zn-SA/CN-1000, respectively. **c, d** The WT analysis of Zn-SA/ CNCl-1000 at OCV and −0.9 V vs. RHE during in-situ CO₂RR, respectively. **e, f** The WT analysis of Zn-SA/CN-1000 at OCV and −0.9 V vs. RHE during in-situ CO₂RR, respectively. The potentials are provided without iR correction.

species as key intermediate for CO formation. As shown in Supplementary Figs. 37 and 38, we also tested the catalytic performance of $ZnO_x/CN$ catalyst for CO₂RR, revealing the poor catalytic activity of $ZnO_x/CN$ catalyst for CO₂RR.

## The in-situ XANES and EXAFS measurements for CO₂RR

To reveal the regulatory mechanism of planar chlorination engineering on Zn-N₄ sites for CO₂RR, in-situ XANES and EXAFS measurements were carried out at Zn K-edge. The photograph of the in-situ EXAFS experiments for CO₂RR was exhibited in Supplementary Fig. 39. We measured the XANES spectra catalyzed by Zn-SA/CNCl-1000 for CO₂RR at open circuit voltage (OCV) and −0.9 V vs. RHE, as shown in the inset of Fig. 4a. The intensity of XANES white-line (WL) obviously

decreased at −0.9 V vs. RHE compared with the spectrum at OCV, indicating the filling of *p* electrons on Zn catalytic sites during CO₂RR[38]. The corresponding FT-EXAFS spectra in *R* space were shown in Fig. 4a. The intensity of main peak at around 1.5 Å remarkably decreased at −0.9 V vs. RHE, demonstrating the decrease of coordination number of Zn-N bond during in-situ CO₂RR. The fitting results in Supplementary Fig. 40 and Supplementary Table 9 demonstrated the Zn catalytic sites of Zn-SA/CNCl-1000 were evolved from Zn-N₄ sites at OCV into Zn-N₃ sites at −0.9 V vs. RHE. By comparison, for Zn-SA/CN-1000 without Cl-doping, there was no obvious changes in both XANES spectra and FT-EXAFS spectra in *R* space from OCV to −0.9 V vs. RHE, as shown in Fig. 4b. The fitting results in Supplementary Fig. 40 and Supplementary Table 9 revealed the existence of Zn-N₄ sites at both OCV and

−0.9 V vs. RHE catalyzed by Zn-SA/CN-1000. The WT contour plots of Zn-based catalysts at OCV and −0.9 V vs. RHE were analyzed in Fig. 4c−f. Compared with the WT contour plot of Zn-SA/CNCl-1000 at OCV in Fig. 4c, the intensity of main peak (4.2 Å$^{-1}$ in $k$ space and 1.5 Å in $R$ space) of Zn-SA/CNCl-1000 at −0.9 V vs. RHE remarkably decreased, as shown in Fig. 4d, revealing the evolution from Zn-N$_4$ sites at OCV into Zn-N$_3$ sites at −0.9 V vs. RHE. By contrast, there was no obvious changes in the WT contour plots of Zn-SA/CN-1000 at OCV in Fig. 4e and −0.9 V vs. RHE in Fig. 4f. These results revealed that the planar chlorination engineering effectively induced the self-reconstruction of Zn-N$_4$ sites into low-coordinated Zn-N$_3$ sites during CO$_2$RR. As shown in Supplementary Fig. 41, we also performed the potential dependent study for CO$_2$RR catalyzed by Zn-SA/CNCl-1000 by in-situ XANES and EXAFS measurements, revealing the gradual transformation from Zn-N$_4$ site into low-coordinated Zn-N$_3$ site as the potentials gradually decreased from OCV to −1.0 V *vs*. RHE.

## The DFT calculation for CO$_2$RR

To understand the self-reconstruction of Zn-N$_4$ catalytic site for CO$_2$RR induced by planar chlorination engineering, DFT calculation was performed. As shown in Fig. 5a and Supplementary Fig. 42, we constructed ten possible Zn-N$_4$/CNCl models with C-Cl bond at different positions. We compared the relative energies of different Zn-N$_4$/CNCl models in Supplementary Fig. 42. The Zn-N$_4$/CNCl-1 was the most stable model with the lowest energy defined as 0 eV while the Zn-N$_4$/CNCl-2 was the second stable model with energy of 1.51 eV. As the C-Cl bond moved away from the Zn-N$_4$ sites, the relative energy increased, demonstrating the more stable models with C-Cl bond adjacent to Zn-N$_4$ sites. As shown in Supplementary Figs. 43 and 44, the AC-STEM images with EDX spectroscopy elemental mapping and electron energy loss spectroscopy (EELS) measurement revealed that the C and Cl elements were indeed located in the vicinity of Zn species.

Considering the regulation of C-Cl bond for Zn-N$_4$ sites will decrease as the C-Cl bond gradually moves away from the Zn-N$_4$ sites, Zn-N$_4$/CNCl-1, Zn-N$_4$/CNCl-2 and Zn-N$_4$/CNCl-3 with adjacent C-Cl bond of Zn-N$_4$ sites are involved in the following discussion. The CO$_2$RR into CO was the combination of reduction and protonation. Due to the large electronegativity of Cl and N element, the proton will enrich around the coordinated N atom with adjacent C-Cl bond. Therefore, the Gibbs energy changes ($\Delta G$) for the protonation of coordinated N atom on Zn-N$_4$/CNCl-1, Zn-N$_4$/CNCl-2, Zn-N$_4$/CNCl-3 and Zn-N$_4$/CN models were compared in Fig. 5b and the corresponding optimized structures were exhibited in Supplementary Fig. 45. We denoted the protonation of coordinated N atom on Zn-based models as Zn-N$_4$/CNCl-1(H), Zn-N$_4$/CNCl-2(H) and Zn-N$_4$/CNCl-3(H), respectively. The $\Delta G$ for the protonation of coordinated N atom on Zn-N$_4$/CNCl-1, Zn-N$_4$/CNCl-2, Zn-N$_4$/CNCl-3 and Zn-N$_4$/CN models were 0.79 eV, −0.66 eV, 1.06 eV and 1.20 eV, respectively, indicating the adjacent C-Cl bond induced the easier protonation of coordinated N atom than the pristine Zn-N$_4$ site.

As shown in Supplementary Fig. 46, we calculated the energy changes for CO$_2$RR on Zn-N$_4$/CNCl and Zn-N$_4$/CN models without the protonation of coordinated N atom, revealing the direct CO$_2$RR pathways on Zn-N$_4$/CNCl models were unable to induce the self-reconstruction of Zn-N$_4$ site into the low-coordinated Zn-N$_3$ catalytic sites. Besides, as exhibited in Supplementary Fig. 47, the energy changes of protonation of coordinated N atom on Zn-N$_4$/CNCl-1 and Zn-N$_4$/CNCl-2 models were lower than their energy barriers of rate-determining steps (RDS) for direct CO$_2$RR, indicating the protonation of coordinated N atom was easier to occur than direct CO$_2$RR on Zn-N$_4$/CNCl-1 and Zn-N$_4$/CNCl-2 models. Therefore, we considered the CO$_2$RR pathways on Zn-N$_4$/CNCl(H) models after the protonation of coordinated N atom. The optimized structures of $^*$COOH and $^*$CO intermediates on Zn-N$_4$/CNCl(H) models were shown in Supplementary Fig. 48. The protonation of coordinated N atom on Zn-N$_4$/CNCl(H)

models induced the break of Zn-N bond and the self-reconstruction of Zn-N$_4$ site into Zn-N$_3$ site with broken planar-like symmetry during CO$_2$RR, which was also confirmed by in-situ EXAFS measurement. The energy changes for CO$_2$RR on Zn-N$_4$/CNCl(H) and Zn-N$_4$/CN models were shown in Fig. 5c. The energy barriers of RDS on Zn-N$_4$/CNCl-1(H), Zn-N$_4$/CNCl-2(H), Zn-N$_4$/CNCl-3(H) and Zn-N$_4$/CN models were 0.55 eV, 0.59 eV, 0.65 eV and 1.20 eV, respectively, revealing the self-reconstruction of Zn-N$_4$ site into Zn-N$_3$ site remarkably improved the CO$_2$RR activity. Compared with pristine Zn-N$_4$ site, after self-reconstruction of Zn-N$_4$ site into Zn-N$_3$ site, the low-coordinated Zn-N$_3$ site from Zn-N$_4$/CNCl-1(H), Zn-N$_4$/CNCl-2(H) and Zn-N$_4$/CNCl-3(H) effectively strengthened the adsorption of $^*$COOH intermediate and accelerated the reaction rate, which was consistent with the experimental results. Therefore, the catalytic pathways on Zn-N$_4$/CNCl models were summarized in Fig. 5d. The adjacent C-Cl bond induced the easier protonation of coordinated N atom of Zn-N$_4$/CNCl models. The protonation of coordinated N atom subsequently induced the self-reconstruction of Zn-N$_4$ site into highly active Zn-N$_3$ site, which remarkably strengthened the adsorption of $^*$COOH intermediate and lowered the energy barriers of RDS. The catalytic pathways of HER as the competing reaction of CO$_2$RR were exhibited in Fig. 5e. The energy barriers for HER on Zn-N$_4$/CNCl-1(H), Zn-N$_4$/CNCl-2(H), Zn-N$_4$/CNCl-3(H) and Zn-N$_4$/CN models were 0.53 eV, 0.50 eV, 0.03 eV and 0.91 eV, respectively. Although the planar chlorination engineering of Zn-N$_4$ site simultaneously facilitated CO$_2$RR and HER compared with pristine Zn-N$_4$ site, the higher concentration of CO$_2$ and lower concentration of proton in the flow cell electrolyzer with 1 M KOH electrolyte was more advantageous for CO$_2$RR. The difference between limiting potentials for CO$_2$RR and HER ($\Delta U = U_L(CO_2) - U_L(H_2)$) reflected the selectivity of CO$_2$RR. As shown in Fig. 5f, the $\Delta U$ on Zn-N$_4$/CNCl-1(H), Zn-N$_4$/CNCl-2(H), Zn-N$_4$/CNCl-3(H) and Zn-N$_4$/CN models were −0.02 V, −0.09 V, −0.62 V and −0.29 V, respectively. Compared with Zn-N$_4$/CN model, the more positive $\Delta U$ on Zn-N$_4$/CNCl-1(H) and Zn-N$_4$/CNCl-2(H) demonstrated the higher selectivity for CO$_2$RR[21]. While the selectivity for CO$_2$RR on Zn-N$_4$/CNCl-3(H) was inferior than Zn-N$_4$/CN model. Considering the Zn-N$_4$/CNCl-1 and Zn-N$_4$/CNCl-2 were more stable than Zn-N$_4$/CNCl-3, Zn-N$_4$/CNCl-1 and Zn-N$_4$/CNCl-2 as dominant models were more easier formed rather than Zn-N$_4$/CNCl-3 model. Therefore, the in-situ EXAFS measurement and DFT calculation revealed the planar chlorination engineering of Zn-N$_4$ site induced the protonation of coordinated N atom adjacent to C-Cl bond and the self-reconstruction of Zn-N$_4$ site into Zn-N$_3$ site, which effectively strengthened the adsorption of $^*$COOH intermediate and remarkably boosted CO$_2$RR activity.

## Discussion

In summary, we rationally designed the planar chlorination engineering of Zn-N$_4$ sites for improving CO$_2$RR. The planar chlorination engineering induced the self-reconstruction of Zn-N$_4$ site into Zn-N$_3$ site with broken symmetry for remarkably boosting the CO$_2$RR activity, with around 29 times and 83 times improvement in $J_{CO}$ and TOF value at −0.93 V vs. RHE. The Zn-SA/CNCl-1000 also exhibited FE$_{CO}$ above 90% over a broad potential window from −0.63 V to −0.93 V vs. RHE and good stability during 50 h test at high current density of 200 mA/cm$^2$. This work reveals the potential of planar chlorination engineering for improving the CO$_2$RR activity and promising application in industrial catalysis by breaking the geometric symmetry of traditional metal-N$_4$ sites.

## Methods

### Reagents

Zinc nitrate hexahydrate (Zn(NO$_3$)$_2$·6H$_2$O, Aladdin, 99.99%), 2-methylimidazole (Acros, 99%), methanol (Beijing Chemical Reagent, AR), N,N-dimethylformamide (DMF, Sinopharm Chemical, AR), ethanol (Sinopharm chemical, AR), NaCl (Aladdin, 99.8%, GR), argon gas

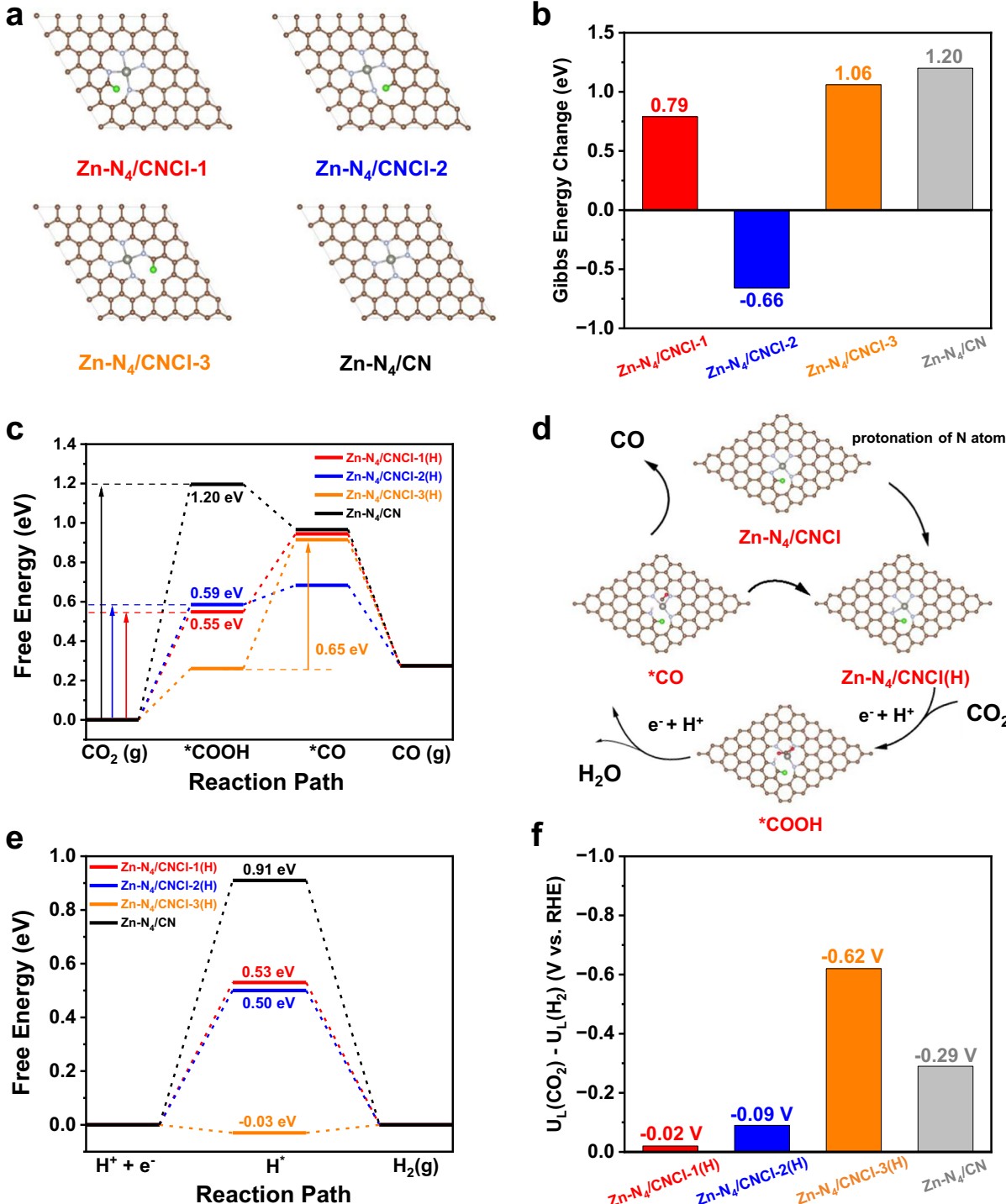

**Fig. 5 | DFT calculation of self-reconstruction of Zn-N$_4$ catalytic site for CO$_2$RR induced by planar chlorination engineering. a** The optimized structures of Zn-N$_4$/CNCl−1, Zn-N$_4$/CNCl-2, Zn-N$_4$/CNCl-3 and Zn-N$_4$/CN models. The brown, light blue, gray and green balls represented C, N, Zn and Cl atoms, respectively. **b** The comparison of Gibbs energy changes of Zn-N$_4$/CNCl-1, Zn-N$_4$/CNCl-2, Zn-N$_4$/CNCl-3 and Zn-N$_4$/CN models during the protonation of coordinated N atom adjacent to

C-Cl bond. **c** The catalytic pathways for CO$_2$RR on Zn-N$_4$/CNCl-1(H), Zn-N$_4$/CNCl-2(H), Zn-N$_4$/CNCl-3(H) and Zn-N$_4$/CN models. **d** The schematic illustration of catalytic mechanism on Zn-N$_4$/CNCl models. **e** The catalytic pathways for HER on different models. **f** The comparison of U$_L$(CO$_2$)-U$_L$(H$_2$) of Zn-N$_4$/CNCl−1(H), Zn-N$_4$/CNCl-2(H), Zn-N$_4$/CNCl-3(H) and Zn-N$_4$/CN models.

(99.999%), carbon dioxide (99.999%), potassium hydroxide (Aladdin, 99%), potassium bicarbonate (Aladdin, 99%), Nafion solution (5 wt%, Alfa Aesar), and iridium oxide (Aladdin, 99.9%). Aqueous solutions were all prepared with pure water obtained from a Milli-Q water system (Millipore, 18.2 MΩ cm).

## Preparation of ZIF-8

ZIF-8 was synthesized by referring a reported article[39]. Firstly, DMF and methanol were mixed with the volume ratio of 4:1 (V$_{DMF}$:V$_{methanol}$ = 4:1) to prepare a a mixed solvent. The Zn(NO$_3$)$_2$·6H$_2$O (1069 mg) was dissolved in 15 ml DMF/methanol mixed solvent and the mixer was under

ultrasound to obtain the homogeneous solution A. The 2-methylimidazole (1161 mg) was dissolved in another 10 ml DMF/methanol mixed solvent by ultrasonication to prepare the homogeneous solution B. Then, the solution B was pour into solution A and the mixer was stirred for several seconds to prepare homogeneous solution. After crystallization of ZIF-8 for 12 h at room temperature with moderate stirring, we collected the ZIF-8 powder by centrifugation (15,000 r.p.m. 5 min) and the ZIF-8 powder was washed by methanol for three times. Finally, the ZIF-8 powder was put in a oven at 50 °C for several hours to evaporate methanol.

## Preparation of Zn-SA/CNCl catalysts

We utilized the NaCl-co-pyrolysis strategy to synthesize Zn-SA/CNCl catalysts, similar to the synthetic method reported by our prior work[35]. The ZIF-8 and NaCl as the precursors were under co-pyrolysis for preparing Zn-SA/CNCl catalysts. The photographs of synthetic procedure were exhibited in Supplementary Fig. 2. 3 g NaCl in a smaller ceramic boat was put in the upstream of 150 mg ZIF-8 powder. The NaCl and ZIF-8 as precursors were put together in a larger ceramic boat with a lid and the larger ceramic boat was put into a tube furnace. The air in the tube furnace was removed by a vacuum pump and then the argon gas (99.999%) was input into the tube furnace. The above treatment was repeated for three times. After heating the precursors at 1000 °C (heating rate: 5 °C/min) for 3 h under flowing argon gas (99.999%), the Zn-SA/CNCl-1000 catalyst was prepared.

During synthesis, the pyrolysis temperature of Zn-SA/CNCl-1000 was 1000 °C, higher than that of Zn-SA/CNCl SAzyme (950 °C) reported in the reference[35]. The higher pyrolysis temperature facilitated the fracture of Zn-Cl bond. Therefore, the Cl element in Zn-SA/CNCl-1000 in this work mainly existed as C-Cl bond and the Zn-N$_4$ site with adjacent C-Cl bond was the active site in Zn-SA/CNCl-1000. While the Cl element in Zn-SA/CNCl SAzyme reported in the reference[35] existed as Zn-Cl bond and C-Cl bond and the Zn atom-pair (Zn-N$_4$Cl and Zn-N$_4$ sites) was the active site in Zn-SA/CNCl SAzyme, which was different from the Zn-SA/CNCl-1000 catalyst in this work.

Similarly, we synthesized the Zn-SA/CNCl-850 and Zn-SA/CNCl-920 catalysts at 850 °C and 920 °C for 3 h, respectively. After pyrolysis, the catalysts were cooled to room temperature naturally under flowing argon gas (99.999%).

## Preparation of Zn-SA/CN-1000 catalyst

We synthesized the Zn-SA/CN-1000 catalyst by pyrolysis of pure ZIF-8 powder without NaCl. We heated the ZIF-8 powder to 1000 °C for 3 h with a heating rate of 5 °C/min under flowing argon gas (99.999%). After pyrolysis, the Zn-SA/CN-1000 catalyst was cooled to room temperature naturally under flowing argon gas (99.999%).

## Characterization

The Rigaku RU-200b X-ray powder (XRD) diffractometer with Cu Kα radiation ($\lambda = 1.5418$ Å) was utilized to obtain the X-ray powder (XRD) patterns of Zn-based samples. The inductively coupled plasma optical emission spectrometry (ICP-OES) measurement was carried out to determine the Zn contents of Zn-based samples. The high-resolution HAADF-STEM images and corresponding elemental mapping results of Zn-based samples were measured by a JEOL-2100F FETEM with electron acceleration energy of 200 kV. To analyze the atomic structures of Zn-based samples, AC-STEM images of Zn-based samples were obtained. During AC-STEM measurement, the ARM-200CF (JEOL, Tokyo, Japan) transmission electron microscope was operated at 200 keV and was equipped with double spherical aberration (Cs) correctors. The probe had the attainable resolution of 78 picometers, defined by the objective pre-field. For AC-STEM with EDX spectroscopy elemental mapping measurement, the Titan ETTEM Themis was operated at 200 keV. For AC-STEM with EELS, the Spectra300 (ThermoFisher Scientific, USA) transmission electron microscope was operated at 300 keV and was equipped with double spherical aberration (Cs) correctors. The probe had the attainable resolution of 50 picometers, defined by the objective pre-field. During the X-ray photoelectron spectroscopy (XPS) measurement of Zn-based samples, an ESCALAB 250 Xi X-ray photoelectron spectrometer with Al Kα radiation was used. The binding energy of C 1s was set to 284.8 eV to calibrate the binding energies of all Zn-based samples. In order to prevent the surface charging of samples, we switched on an electron flood gun during XPS measurement. The method of Brunauer-Emmett-Teller (BET) with a QuadraSorb SI automated surface area and pore size analyzer (Quantachrome Instruments) at 77 K was used to measure the nitrogen sorption isotherm experiments of Zn-based samples. All Zn-based samples were degassed at 200 °C before measurement. The distribution of mesopores and micropores of Zn-based samples were analyzed by the Barrett-Joyner-Halenda (BJH) method.

## EXAFS measurement

The spectra of X-ray absorption fine structure at Zn K-edge were collected at the TPS-21A beamline of the National Synchrotron Radiation Research Center (NSRRC, Hsinchu, Taiwan), operated at 3 GeV with a maximum current of 500 mA. The EXAFS measurement was performed at room temperature in fluorescence mode by utilization of a Lytle detector. The powder of catalysts were pelletized as a disks (diameter of 8 mm) with polyvinylidene fluoride (PVDF) powder as binders. The energies of spectra at Zn K-edge were corrected by the absorption edge of Zn foil. The fluorescence mold was performed for in situ XANES and EXAFS measurements. During in situ EXAFS measurement, the Zn-based catalysts with loading of 2 mg/cm$^2$ on carbon paper (the area of a circle with the diameter of 3 mm) served as the working electrode. The 15 cm and 3 cm platinum wire with a diameter 0.2 mm (Nilaco, 99.95%) served as the counter electrode and reference electrode, respectively. The electric potential of the platinum reference electrode was calibrated using the Ag/AgCl (KCl-saturated) electrode (SciKET, LEDONLAB). The CO$_2$-saturated 0.5 M KHCO$_3$ solution served as electrolyte (2 ml). The XANES and EXAFS measurement at Cl K-edge was recorded at the 4B7A station in Beijing Synchrotron Radiation Facility in TEY mode.

## EXAFS data analysis

The Demeter (version 0.9.26) software package was utilized during the XAFS data normalization and Fourier transformed data fitting of Zn-based samples. During fitting, k$^3$ weights, $k$-range (3.4--12.4 Å$^{-1}$), and $R$ range (1--2 Å) were applied for ex-situ results; $k$-range (3.4--9.7 Å$^{-1}$), and $R$ range (1--2 Å) for in-situ experiments.

During the Wavelet Transform analysis, the χ(k) data format was imported into the Larch Python code. The parameters were listed as follows: $R$ range was 1–5 Å; $k$ range was 2–14 Å$^{-1}$; k-weight was 3; and the Morlet function ($\kappa = 6$, $\sigma = 1$) was used as the mother wavelet to provide the overall distribution.

For fitting the EXAFS data at Cl K-edge, k$^3$ weights, the $k$ range (3.0–10.0 Å$^{-1}$), and $R$ range (1.0–3.0 Å) were applied for Zn-SA/CNCl samples.

During the Wavelet Transform analysis, the χ(k) data format was imported into the Larch Python code. The parameters were listed as follows: $R$ range was 1–5 Å; $k$ range was 2–10 Å$^{-1}$; k-weight was 2; and the Morlet function ($\kappa = 6$, $\sigma = 1$) was used as the mother wavelet to provide the overall distribution.

## Electrochemical CO$_2$RR measurements

All electrochemical tests were carried out by utilization of an electrochemical workstation (Autolab PGSTAT 204, Metrohm). The catalytic performance of the Zn-based catalysts for CO$_2$RR was evaluated in flow-cell and zero-gap membrane reactor. For the measurement in flow-cell, the electrocatalyst inks were composed of 20 mg catalysts, 1.9 ml ethanol, and 0.1 ml Nafion solution by ultrasonication for 1 h. We

sprayed the catalyst ink onto the gas diffusion electrode (GDE, Sigracet 28BC) with the catalyst loading of $1\,mg\,cm^{-2}$. The electrode area was $1.0\,cm^2$ during measurement. The Ni foam ($1 \times 1\,cm$, $0.5\,mm$ thickness, Suzhou Taili Foam Metal Factory) and Hg/HgO electrode (CHI152, Shanghai Yueci Electronic Co.) served as the counter electrode and reference electrode, respectively. We separated the working electrode with Zn-based catalysts and counter electrode by an anionic exchange membrane (AEM, $2 \times 2\,cm$, $50\,\mu m$ thickness Sustainion® X37-50 grade 60, Dioxide Materials). The membrane was pre-treated by $1\,M$ KOH for 2 days and storage in $1\,M$ KOH solutions. $1\,M$ KOH (pH = $14.0 \pm 0.2$) solution served as the electrolyte and the flow rate of $CO_2$ was 50 sccm by utilization of a gas mass flow meter (CS200, Beijing Sevenstar flow Co., LTD). The collected potential was converted into the reversible hydrogen electrode (RHE) using $E$ (versus RHE) = $E$ (versus Hg/HgO) + 0.098 V + 0.059 × pH and was iR corrected using automatic compensation of workstation with 85% level. The measured electrolyte resistance was $1.0 \pm 0.15\,\Omega$.

During the stability measurement in two-electrode zero-gap membrane reactor, we utilized an AEM ($2 \times 2\,cm$, $50\,\mu m$ thickness, Sustainion® X37-50 grade 60, Dioxide Materials) in order to inhibit cation shuttle. The Zn-SA/CNCl-1000 catalyst ($1.0\,mg\,cm^{-2}$) loaded on GDE with $1.0\,cm^2$ electrode area served as a cathode, and the $IrO_2$ (Aladdin, 99.9%) catalyst on a titanium mesh (100 mesh, $1 \times 1\,cm$) served as an anode. The cathode side was supplied with humified $CO_2$ gas with a flow rate of $30\,ml\,min^{-1}$. The anode side was circulated with $0.5\,M$ $KHCO_3$ (pH = $8.3 \pm 0.3$) aqueous solution with a flow rate of $100\,mL\,min^{-1}$.

The electrochemical surface area of electrocatalysts was estimated by performing CV in the potential range of 1.0 to 1.1 V vs. RHE at different scan rates ($v$) of 10, 20, 30, 40, 50, 60, 70, 80, 90 and $100\,mV\,s^{-1}$, followed by extracting the slope from the resulting $\Delta j = (j_a - j_c)/2$ vs. $v$ plots ($j_a$ and $j_c$ represent the anodic and cathodic current densities of the catalysts at 1.05 V vs. RHE). The electrodes for ECSA test were prepared by droping $10\,\mu L$ electrocaralysts inks (10 mg catalysts, 0.75 ml ethanol, $0.2\,ml\,H_2O$, and 0.05 ml Nafion solution) on glassy-carbon electrode (GCE, diameter = 0.5 cm) yielding a catalyst loading of $0.5\,mg\,cm^{-2}$. All the electrochemical measurements were conducted at room temperature.

## Product analysis

The gaseous products for $CO_2RR$ were detected and analyzed by online GC (GC2014C, Shimadzu), which was equipped with a thermal conductivity detector (TCD) and flame ionization detector (FID) detector. The argon (Praxair 99.999%) served as the carrier gas, and we calibrated the GC with $H_2$ and CO. The analysis time of each gaseous product was 7 min. The FEs for gaseous products were calculated as follows:

$$FE\,(\%) = \frac{NF \times \left(\frac{v}{60}\right) \times \left(\frac{y}{24.5 \times 10^9}\right)}{i} \times 100\% \qquad (1)$$

In the above equation, the $N$ represents the number of electrons required for products ($N$ is equal to 2 for $H_2$ and CO), y (ppm) represents the volume concentration of the gaseous product, $v$ (sccm) represents the measured gas flow rate (from the outlet of flow cell using Soap Film Flow Meter), $i$ (A) represents the collected cell current, $F$ is the Faraday constant ($96485\,C\,mol^{-1}$).

We measured the collected electrolytes to quantify the liquid products by utilization of NMR (Bruker Avance III, 600 M). The dimethyl sulfoxide (DMSO) with known concentration served as an interior label.

We calculated the TOF ($h^{-1}$) values for $CO_2RR$ into CO based on the following equation:

$$TOF = \frac{I_{CO}/NF}{m_{cat} \times \omega/M_{Zn}} \times 3600 \qquad (2)$$

In the above equation, the $I_{CO}$ (A) represents the partial current for CO. The $N$ represents the number of electrons required ($N$ is equal to 2 for CO). $F$ is the Faraday constant ($96485\,C\,mol^{-1}$). The $m_{cat}$ (g) represents the catalyst mass in the electrode. The $\omega$ represents metal loading in the catalyst. The $M_{Zn}$ represents the atomic mass of Zn element ($65.38\,g\,mol^{-1}$).

### In-situ ATR-SEIRAS measurements

We collected the in-situ ATR-SEIRAS spectra by an FT-IR spectrometer (Nicolet iS50, Thermo Scientific), which was equipped with an MCT-A detector. We prepared the catalyst inks by mixing 5 mg electrocatalysts, 0.95 ml ethanol, and $50\,\mu l$ of Nafion solution. An Au film was chemically deposited on the Si prism. Then, we dropped $10\,\mu l$ of catalyst ink onto the central surface area ($\sim 1\,cm^2$) of a hemicylindrical Au coated Si prism yielding a catalyst loading of $0.1\,mg\,cm^{-2}$. In a spectroelectrochemical cell, the catalyst on the Si prism served as the working electrode. A Pt wire ($\Phi 1\,mm \times 37\,mm$) served as the counter electrode. The Ag/AgCl electrode (CHI111, Shanghai Yueci Electronic Co.) served as the reference electrode. $0.5\,M$ $KHCO_3$ (pH = $8.3 \pm 0.3$) solution (50 ml) served as the electrolyte. All the in-situ ATR-SEIRAS spectra were collected at a resolution of $4\,cm^{-1}$, and each single-beam spectrum was an average of 200 scans. We utilized an Autolab PGSTAT 204 electrochemical workstation (Metrohm) for potential control. During the measurement, high pure $CO_2$ (20 sccm) was continuously introduced into the electrolyte.

### DFT calculation

The present spin-polarized first principle DFT calculations were carried out by Vienna Ab initio Simulation Package(VASP)[40] with the projector augmented wave (PAW) method[41]. The exchange-functional was treated by utilization of the generalized gradient approximation (GGA) of Perdew-Burke-Ernzerhof (PBE) functional[42]. We set the energy cutoff for the plane wave basis expansion to 500 eV. We set the force on each atom less than $0.02\,eV/\text{Å}$ for convergence criterion of geometry relaxation. The Brillouin zone integration was performed by utilizing $3 \times 3 \times 1$ k-point for sampling. A convergence energy threshold of $10^{-5}$ eV was applied for the self-consistent calculations. To consider the van der Waals interaction[43], the DFT-D3 method was used. We constructed the computational model based on a 6×6 supercell with single-layer graphene. To avoid the interaction between periodic structures, we added a 15 Å vacuum along the z direction.

We calculated the free energies for $CO_2RR$ by utilizing the following equation[44]:

$$\Delta G = \Delta E_{DFT} + \Delta E_{ZPE} - T\Delta S \qquad (3)$$

In the above equation, the $\Delta E_{DFT}$ represented the DFT energy difference, and the $\Delta E_{ZPE}$ and $T\Delta S$ terms were obtained based on vibration analysis.

## Data availability

The data generated in this study are provided in the article and the Supplementary Information file. The source data of the figures are available on Figshare.

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

## Acknowledgements

This work was supported by the National Key Research and Development Program of China (No. 2022YFB2404300, L.M.), the National Natural Science Foundation of China (No. 52273231, L.M.), (No. 22109123, L.M.), (No. 22405261, L.L.) and (No. 22409159, S.C.), the National Postdoctoral Program for Innovative Talents of China (No. BX20220159, S.W.), China Postdoctoral Science Foundation (2023M731785, S.W.), (2023TQ0341, L.L.), (2023M743369, L.L.), the Natural Science Foundation of Hubei Province (No. 2022CFD089, L.M.), Natural Science Basic Research Program of Shaanxi (Program No. 2024JC-YBQN-0119, S.C.) and (No. 2023SYJ04, S.C.), the Fundamental Research Funds for the Central Universities (WK2060000068, L.L.), the Postdoctoral Fellowship Program of CPSF (GZB20230706, L.L.), and the Anhui Provincial Natural Science Foundation (2408085QB046, L.L.). Prof. Shenghua Chen acknowledges the Young Talent Support Plan of Xi'an Jiaotong University (71211223010707, S.C.). We thank the TPS-21A beamline for XAFS measurements in the National Synchrotron Radiation Research Center (NSRRC, Hsinchu, Taiwan). We thank Dr. Lei Zheng, Dr. Chenyan Ma and Dr. Shuhu Liu from 4B7A station in Beijing Synchrotron Radiation Facility (BSRF) for the EXAFS measurement at Cl K-edge. The numerical calculations in this paper have been done on the supercomputing system in the Supercomputing Center of the University of Science and Technology of China.

## Author contributions

S.W., J.Z., L.M., and S.C. conceived the idea and wrote the paper. S.W. performed the synthesis and characterization of catalysts, collected and analyzed the data. S.C., J.Z., X.C., and K.G. evaluated the catalytic performance for $CO_2RR$. R.Y. measured and analyzed the XPS data. L.L. performed the DFT calculations. C.-Y.C. performed the XANES and EXAFS measurements and analyzed the data.

## Competing interests

The authors declare no competing interests.
