## [Transparent Peer Review file · Nature Communications]

Planar chlorination engineering induced symmetry-broken single-atom site catalyst for enhanced CO₂ electroreduction.

Corresponding Author: Professor Liqiang Mai

Version 0:

Reviewer comments:

Reviewer #1

(Remarks to the Author)

In this work, Wei et al. reported a planar chlorination method for symmetric fracture of metal-N₄. The Zn-N₃ active site synthesized by this method has achieved efficient CO electroreduction. High-quality characterizations and DFT calculations explained the structure and evolution process of materials. Overall, this is a very interesting discovery. I think this manuscript can be published in Nature communications after addressing the below concerns:

- 1) Limited information can be seen in Supplementary Figure 4. I suggest providing higher resolution STEM images to confirm the absence of Zn-based nanoparticles in Zn-SA/CN-1000.
- 2) The EXAFS of Zn-SA/CNCl-1000 very similar to Zn-SA/CNCl-850 and Zn-SA/CNCl-920, but no Zn-Cl pathway was observed in Figure 2f. Please recheck the fitting results.
- 3) At present, the evolution of the structure from Zn-N₄Cl₁ to Zn-N₄ is not decisive, as there is still a few ion Cl⁻ present in Zn-SA/CNCl-1000. The simulated XANES curve from the optimized model should be compared with experimental results.
- 4) The authors claim that Figure 4b shows the EXAFS of Zn-SA/CN-1000 has no obvious change. However, by comparing Figure 4a and b directly, two samples exhibit similar trends.
- 5) Although the author conducted the in-situ characterizations to confirm the structural evolution process of Zn atoms, the conclusion is still relatively vague. The authors are suggested to provide more characterization to confirm the changes in the coordination structure of Zn, such as EELS.

Reviewer #2

(Remarks to the Author)

The manuscript "Planar chlorination engineering induced symmetry-broken single-atom site catalyst for enhanced CO₂ electroreduction" by Wei et al proposes chlorination of the Zn-nitrogen co-doped catalyst to boost its selectivity to CO production. They observed that the treatment of the catalyst modified with chlorination at different temperatures results in drastically different catalytic properties. The obtained catalytic results are intriguing. However, I am not convinced that the structure model, proposed by authors, is indeed supported by provided experimental evidences.

The presence of chlorine species in the vicinity of active Zn species is deduced solely from the XAS data. However, the presented XAS data seem to be inconclusive, in my opinion. In EXAFS data for the as-prepared samples only very minor differences are observed for the samples treated at different temperatures, while XANES spectra seem to be nearly identical (Figure 2). This suggests that the only difference between different samples is just a slight difference in the structural disorder. To show the differences in the number of Cl atoms in the first coordination shell of Zn, authors carried out EXAFS data fitting. However, it could be challenging to detect reliably the presence of small amount of Cl atoms on the background of strong contribution of Zn-N bonds. Have authors tried to perform fitting without Zn-Cl path, with one or two non-equivalent

Zn-N paths only? Does the inclusion of Zn-Cl path really makes the fit significantly better?

Overall, the employed fitting model is not clear. How many different paths were included in the fit? The R-factors reported in Supplementary Table 1 all seem to be very small, which is highly unusual for such disordered system. All the fitting results are reported without uncertainties, which makes them essentially useless. Why Zn-Cl bond distance differs so strongly for the samples treated at 850 deg C and 920 deg C? It seems that the disorder factors for Zn-Cl and Zn-N bonds were constrained to be the same – this should be clearly stated in text. Were any other constraints employed? Please show also EXAFS fitting results in k-space for all samples, not just for one as is currently done in Figure S16.

Furthermore, after all these troubles the main conclusion based on the ex-situ EXAFS data analysis seems to be that the sample treated at 1000 deg C does not have any Zn-Cl bonds, and its local structure is remarkably similar to that of the sample prepared in the absence of chlorine. So, how then authors can conclude that chlorine is actually doing something for the catalytic mechanism? How can they claim that the C-Cl bonds are indeed located in the vicinity of Zn species?

To conclude, major revision is necessary, since the author's conclusions do not seem to be supported by experimental data.

Reviewer #3

(Remarks to the Author)

In this article, Wei et al reported distorted Zn-N3 site which is reconstructed from Zn-N4 moiety by the presence of Cl and found to be highly active for CO₂ reduction reaction. The Zn-N3 site with broken symmetry believed to strengthen the adsorption of COOH intermediate (supported by DFT calculations) which is responsible for improving the catalytic activity. The distorted Zn single site showed around 90% Faradaic efficiency for the conversion of CO₂ to CO in the potential range from -0.63V to -0.93 V Versus RHE. The authors did extensive ex-situ characterization of the materials via XAS, STEM, XRD, XPS, RAMAN which is appreciated. Moreover, they performed some in-situ XAS and SEIRAS spectroscopy to understand the mechanistic details. However, in my opinion, there are many studies conducted in this direction in the past and the major concern about whether the single-atom catalyst remains as it is or agglomerates under reaction conditions. Such as Yang et al found that Ni(I) center is highly dynamic in nature under operando conditions (<https://www.nature.com/articles/s41560-017-0078-8>). Even Zn SACs are already reported to be active for electrochemical CO₂ reduction with high Faradaic efficiency (<https://doi.org/10.1002/anie.201805871>). Hence, I found a lack of novelty in this work to be published in such a prestigious journal. I have the following comments that the authors can take into account to improve their manuscript in the future.

(a) There are no details of in-situ XAS measurements such as what is exactly a Lytle detector? What was the gas flow? Did the author perform a potential dependent study? Usually presence of bubble leads to noisy XAS spectrum. How did the authors overcome this? Please see the review by Sarma et al. (<https://doi.org/10.1021/acs.chemrev.2c00495>).

(b) The authors conducted in-situ ATR-SEIRAS measurements over Si prism and claimed that there are C=O and C-OH features. However, when these experiments were conducted authors should keep in mind that Si absorbs the IR strongly in the range of 1500-1000 cm⁻¹. Hence any data in this range might not be so meaningful and might have occurred due to some artifact. A more interesting region will be Metal-Carbonyl which appears in the range of 1900-2100 cm⁻¹. Also in the zero voltage potential why do there are already some features? May be these features come from some organic that is already present in the Nafion or residue from ethanol. For more details please see <https://doi.org/10.1021/acs.nanolett.9b01582>

(c) The authors should have include a Zn-Cl reference sample in their XAS measurements which would have helped the readers to follow the XAS results. There is consistent presence of a strong feature in the range of 2-3 Å in the figure 2 (d), (e) and (f). Does the author know the origin of this feature?

(d) The fitting results of the XAS data in supplementary table 1 lack errors and also details. When is the asterisk or star represented? What did the authors fit in achieving those results?

Version 1:

Reviewer comments:

Reviewer #1

(Remarks to the Author)

OK for publication.

Reviewer #2

(Remarks to the Author)

During the revision authors have addressed some of my prior comments. Nonetheless, I feel that two my most important points were not answered.

First, I am still not convinced that there are any Zn-Cl bonds in any of the investigated samples. In my previous review, I have asked authors to carry out EXAFS fitting without Zn-Cl bonds included, with one or two non-equivalent Zn-N paths only, to demonstrate that the inclusion of Cl indeed improves the fits noticeably. This has not been done. In the revised version of the manuscript authors also show XANES spectrum simulated for their structure model. However, it looks rather featureless, and similar to the spectrum of singly dispersed Zn species in general. Does really the presence of Cl in this structure affect the XANES spectrum in any noticeably way? There is also no explanation in the manuscript, how the simulations of XANES spectra were carried out. Which code was used? Was there any attempt to optimize the structure to improve the agreement with experimental data?

In the revised version of the manuscript, authors also provide XPS data to show the presence of Cl in their catalysts. This is an important observation. However, XPS results cannot be directly compared with XAS, since the former is surface sensitive technique while the latter – bulk sensitive. More importantly, XPS does not show that there indeed are Zn-Cl bonds in the material – just that there is Cl somewhere in this system.

To demonstrate the presence or the absence of Zn-Cl bonds in their EXAFS fits, authors argue in their reply that if EXAFS fitting gives a negative coordination number for Zn-Cl bond, then this surely means that there are no Zn-Cl bonds in the material. However, the negative coordination numbers can also simply mean the chosen structure model is inadequate, e.g., that the imposed constraint that the disorder factors for Zn-Cl and Zn-N bonds is the same cannot be applied. This can also be simply a result of the instability of non-linear fitting, and can disappear, if different initial guesses for structure parameters are chosen.

Even more importantly, authors still have not provided a satisfactory answer to my last question from the previous review: if their XAS data suggest that the sample treated at 1000 deg C does not have any Zn-Cl bonds, and XPS also shows that the amount of Cl in this sample is rather low, how can they conclude that chlorine is actually doing something for the catalytic mechanism? How can they claim that the C-Cl bonds are indeed located in the vicinity of Zn species?

To conclude, major revision is still necessary, since the author's conclusions do not seem to be supported by experimental data.

Reviewer #3

(Remarks to the Author)

In this revised manuscript, Wei et al. has conducted significant revision of the concerns raised by the reviewers which I highly appreciate. However, I am still little bit puzzled by their explanation of existence of only single atom as there are still some additional features (between 2.5 -3.5 Å) in the EXAFS regions of figure 20 of supplementary information. Keep in mind that EXAFS is less sensitive with distances. Hence, any features that you observe in the high radial distance might be low in intensity but can contribute significantly. The contribution from Cl cannot be ignored as there is clear evidence of presence of Cl from the XPS results. Following are some additional comments.

(a) Did the authors performed any catalytic test with ZnOx clusters over the same support?

(b) Since the authors already have the DFT models, I suggest also fit of the second shell to make sure that there is no contribution of other component beyond 2 Å.

(c) The authors claim that the intensity of the wave in the range of 2-3 Å (red box b) was similar to that of the wave in the range of 0-0.75 Å. I agree that there is no bond that can be in the range 0-1 Å. However, as I said the intensity of the EXAFS decreases with radial distance. So, to compare between 0-1 Å and 1.5-2.5 Å is not the same. The features in figure 2 (d), (e) and (f) cannot be simply ignored.

As the novelty of the work lies on the restructuring of the Zn coordination, this is very important that the authors carefully make their conclusion to be published in such a prestigious journal.

Version 2:

Reviewer comments:

Reviewer #2

(Remarks to the Author)

During the revision, authors have made the manuscript much more convincing, in particular, by adding Cl K-edge XAS data. I appreciate authors' efforts in improving the manuscript. I have just a few additional comments.

1) Please add Cl K-edge EXAFS data in k-space to Supplementary Information

2) Please provide fits of Cl K-edge EXAFS data that would confirm the presence of Cl-Zn bonds.

3) Please provide details of Cl K-edge XAS data processing in the Methods section. Also, currently it is stated in the Methods section that that only XANES data were collected, while in the manuscript EXAFS data are discussed as well.

Reviewer #3

(Remarks to the Author)

The authors have implemented all my previous concerns and suggestions. I do not have any further questions.

Response to the Reviewers

Reviewer #1:

Comment : In this work, Wei et al. reported a planar chlorination method for symmetric fracture of metal-N₄. The Zn-N₃ active site synthesized by this method has achieved efficient CO electroreduction. High-quality characterizations and DFT calculations explained the structure and evolution process of materials. Overall, this is a very interesting discovery. I think this manuscript can be published in Nature communications after addressing the below concerns:

Reply: We sincerely appreciate the comments provided by Reviewer #1. We extend our sincere appreciation to the reviewer for the time, expertise, and constructive feedback, which have undoubtedly contributed to the improvement of our manuscript.

Comment 1: Limited information can be seen in Supplementary Figure 4. I suggest providing higher resolution STEM images to confirm the absence of Zn-based nanoparticles in Zn-SA/CN-1000.

Reply: We sincerely appreciate the comments provided by Reviewer #1 regarding providing higher resolution STEM images to confirm the absence of Zn-based nanoparticles in Zn-SA/CN-1000.

As shown in **Figure R1**, we measured the higher resolution STEM images of Zn-SA/CN-1000. No Zn-based nanoparticles were found in Zn-SA/CN-1000.

As shown in **Supplementary Fig. 20**, the atomic dispersion of Zn element in Zn-SA/CN-1000 was also confirmed by EXAFS measurement and fitting. There was only one dominant peak at around 1.5 Å, which was attributable to the Zn-N bond from Zn-N₄ sites, without the Zn-Zn bond at around 2.2 Å from Zn nanoparticles or Zn-O-Zn bond at around 2.9 Å from ZnO nanoparticles.

Therefore, by combination of higher resolution STEM image and EXAFS measurement, we concluded that no Zn-based nanoparticles existed in the

Zn-SA/CN-1000. We added the **Figure R1** as the **Supplementary Fig. 5** in the revised Supplementary Information.

Figure R1. The higher resolution STEM images of Zn-SA/CN-1000.

Supplementary Fig. 20. a, The fitting result of Zn-SA/CN-1000 in k space at Zn K-edge. **b,** The fitting result of Zn-SA/CN-1000 in R space at Zn K-edge.

Comment 2: The EXAFS of Zn-SA/CNCl-1000 very similar to Zn-SA/CNCl-850 and Zn-SA/CNCl-920, but no Zn-Cl pathway was observed in Figure 2f. Please recheck the fitting results.

Reply: We sincerely appreciate the comments provided by Reviewer #1 regarding rechecking the fitting result of Zn-SA/CNCI-1000.

We also performed XPS measurement to determine whether the Zn-Cl pathway should be considered during EXAFS fitting. The XPS spectra for the Cl 2p of Zn-SA/CNCI samples were shown in **Figs. 1e-1g**. The ionic Cl from Zn-Cl bond of Zn-SA/CNCI-850, Zn-SA/CNCI-920 and Zn-SA/CNCI-1000 were 56.1%, 43.8% and 6.9%, respectively. Compared with Zn-SA/CNCI-850 and Zn-SA/CNCI-920, the ionic Cl of Zn-SA/CNCI-1000 is only 6.9% with trace amounts and the signal has poor signal-to-noise ratio. Therefore, we conclude that the Zn-Cl bond exists in Zn-SA/CNCI-850 and Zn-SA/CNCI-920 samples while it is difficult to determine whether the Zn-Cl bond exists in Zn-SA/CNCI-1000 sample.

Figs. 1e-1g. The XPS spectra for the Cl 2p of Zn-SA/CNCI-850, Zn-SA/CNCI-920 and Zn-SA/CNCI-1000, respectively.

We also attempt to introduce the Zn-Cl pathway during the EXAFS fitting of Zn-SA/CNCI-1000. As shown in **Table R1**, the corresponding fitting result exhibited that the coordination numbers of Zn-N bond and Zn-Cl bond were 4.18 and -0.16, respectively. The coordination number of Zn-Cl bond was negative (-0.16), which indicated that introducing the Zn-Cl pathway during the EXAFS fitting of Zn-SA/CNCI-1000 was inappropriate.

Similarly, we also introduced the Zn-Cl pathway during the EXAFS fitting of Zn-SA/CNCI-850 and Zn-SA/CNCI-920 by the same fitting method. The coordination numbers of Zn-Cl bond in Zn-SA/CNCI-850 and Zn-SA/CNCI-920 were 0.36 and

0.27, respectively, revealing that introducing the Zn-Cl pathway for fitting Zn-SA/CNCl-850 and Zn-SA/CNCl-920 was appropriate.

Therefore, the Zn-Cl pathway was not involved during the EXAFS fitting of Zn-SA/CNCl-1000, which was different from those of Zn-SA/CNCl-850 and Zn-SA/CNCl-920.

Table R1. The structural parameters for fitting Zn-SA/CNCl-1000 when introducing Zn-Cl pathway. ($S_0^2 = 0.78$)

Sample	Shell	N	R(Å)	$\sigma^2(\text{\AA}^2)$	$\Delta E_0(\text{eV})$	R factor
Zn-SA/CNCl-1000	Zn-N	4.18 ± 0.17	2.047 ± 0.003	0.00985 ± 0.0008	4.96 ± 0.75	0.00098
	Zn-Cl	-0.16 ± 0.08	2.192 ± 0.041			
Zn-SA/CNCl-920	Zn-N	3.99 ± 0.12	2.004 ± 0.012	0.0097 ± 0.0003	-1.79 ± 0.26	0.0010394
	Zn-Cl	0.27 ± 0.05	2.215 ± 0.015			
Zn-SA/CNCl-850	Zn-N	4.07 ± 0.18	2.023 ± 0.003	0.0106 ± 0.0005	0.43 ± 0.31	0.0026757
	Zn-Cl	0.36 ± 0.08	2.217 ± 0.021			

Comment 3: At present, the evolution of the structure from $\text{Zn-N}_4\text{Cl}_1$ to Zn-N_4 is not decisive, as there is still a few ion Cl- present in Zn-SA/CNCl-1000. The simulated XANES curve from the optimized model should be compare with experimental results.

Reply: We sincerely appreciate the comments provided by Reviewer #1. As analyzed in **Comment 2**, the Zn-Cl pathway is negligible during the EXAFS fitting of Zn-SA/CNCl-1000 and the XPS spectrum for the Cl 2p of Zn-SA/CNCl-1000 also reveals that the ionic Cl from Zn-Cl bond of Zn-SA/CNCl-1000 is rather weak. Therefore, only the Zn-N pathway is considered without Zn-Cl pathway during the

EXAFS fitting of Zn-SA/CNCl-1000.

We also simulate the XANES curve of the optimized model of Zn-SA/CNCl-1000 in **Supplementary Scheme 2**. As shown in **Figure R2a**, the XANES spectrum of Zn-SA/CNCl-1000 and the theoretical spectrum of Zn-N₄/CNCl-1 model (**Figure R2b**) exhibited similar features, revealing the Zn-N₄/CNCl-1 model was a rational model to simulate the Zn-SA/CNCl-1000.

We added the **Figure R2** as the **Supplementary Fig. 19** in the revised Supplementary Information and Manuscript (Line 206-209).

Figure R2. **a**, The XANES spectrum of Zn-SA/CNCl-1000 and the theoretical spectrum of Zn-N₄/CNCl-1 model. **b**, The corresponding Zn-N₄/CNCl-1 model.

Comment 4: The authors claim that Figure 4b shows the EXAFS of Zn-SA/CN-1000 has no obvious change. However, by comparing Figure 4a and b directly, two samples exhibit similar trends.

Reply: We sincerely appreciate the comments provided by Reviewer #1.

Indeed, during in situ CO₂RR, the intensity of Zn-N peak from Zn-SA/CN-1000 had a slight decrease after applying a voltage of -0.9 V vs. RHE, compared with the curve at OCV. The Zn-SA/CN-1000 exhibited similar trends as that of Zn-SA/CNCl-1000. However, compared with Zn-SA/CNCl-1000, the decrease of Zn-N peak intensity from Zn-SA/CN-1000 was inconspicuous, which was probably attributable to the perturbation after applying a voltage of -0.9 V vs. RHE. Besides, as

shown in Figs. 4e-4f, the intensity of Zn-N peak in WT contour plots of Zn-SA/CN-1000 also exhibited no observable changes at OCV and at -0.9 V vs. RHE.

Due to the inconspicuous change of Zn-SA/CN-1000 during the in situ EXAFS measurements for CO₂RR, we can not conclude that the coordination number of Zn-N bond from Zn-SA/CN-1000 has a substantive change for CO₂RR, which is different from that of Zn-SA/CNCl-1000.

Comment 5: Although the author conducted the in-situ characterizations to confirm the structural evolution process of Zn atoms, the conclusion is still relatively vague. The authors are suggested to provide more characterization to confirm the changes in the coordination structure of Zn, such as EELS.

Reply: We sincerely appreciate the comments provided by Reviewer #1 regarding the in-situ characterizations by EELS measurement to confirm the structural evolution process of Zn atoms for CO₂RR.

The electron energy loss spectroscopy (EELS) in the scanning transmission electron microscopy (STEM) can provide some information about the local element distribution around metal catalytic sites. However, it is difficult to achieve the in-situ EELS measurement for CO₂RR by current technology. Before the EELS measurement, the sample is dispersed in solvent and then is dropped on ultrathin carbon film. After drying, the sample on ultrathin carbon film is inserted into the TEM column under Ar atmosphere. During the EELS measurement, the sample is loaded on a ultrathin carbon film in TEM column, which is difficult to apply a constant voltage on the sample and immerse the sample in CO₂-saturated 0.5 M KHCO₃ solution. Therefore, the current EELS technology can not provide the in-situ characterization of catalytic sites for CO₂RR.

Currently, the in-situ EXAFS measurement is a practical and advanced technology to provide the in-situ structural information of catalytic sites for CO₂RR. During the in-situ EXAFS measurement for CO₂RR, an in-situ electrochemical cell

was employed, as shown in **Figure R3**. The catalyst on the carbon paper was immersed in the CO₂-saturated 0.5 M KHCO₃ solution, with a constant voltage during in-situ EXAFS measurement. While the emission and collection of X-ray light were outside the in-situ electrochemical cell.

Figure R3. The photograph of the in-situ EXAFS experiments for CO₂RR.

Reviewer #2:

Comment: *The manuscript “Planar chlorination engineering induced symmetry-broken single-atom site catalyst for enhanced CO₂ electroreduction” by Wei et al proposes chlorination of the Zn-nitrogen co-doped catalyst to boost its selectivity to CO production. They observed that the treatment of the catalyst modified with chloring at different temperatures results in drastically different catalytic properties. The obtained catalytic results are intriguing. However, I am not convinced that the structure model, proposed by authors, is indeed supported by provided experimental evidences.*

Reply: We sincerely appreciate the comments provided by Reviewer #2. We analyzed the structural models of Zn-based catalysts in detail in the revised manuscript by combination of XPS measurement for Cl 2p and EXAFS fitting. More details for EXAFS fitting were provided in the revised manuscript. We extend our sincere appreciation to the reviewer for the time, expertise, and constructive feedback, which have undoubtedly contributed to the improvement of our manuscript.

Comment 1: *The presence of chlorine species in the vicinity of active Zn species is deduced solely from the XAS data. However, the presented XAS data seem to be inconclusive, in my opinion. In EXAFS data for the as-prepared samples only very minor differences are observed for the samples treated at different temperatures, while XANES spectra seem to be nearly identical (Figure 2). This suggests that the only difference between different samples is just a slight difference in the structural disorder. To show the differences in the number of Cl atoms in the first coordination shell of Zn, authors carried out EXAFS data fitting. However, it could be challenging to detect reliably the presence of small amount of Cl atoms on the background of strong contribution of Zn-N bonds. Have authors tried to perform fitting without Zn-Cl path, with one or two non-equivalent Zn-N paths only? Does the inclusion of Zn-Cl path really makes the fit significantly better?*

Reply: We sincerely appreciate the comments provided by Reviewer #2 regarding the rationality of Zn-Cl pathway during the EXAFS fitting of Zn-SA/CNCl-850 and Zn-SA/CNCl-920.

Firstly, we also performed XPS measurement to determine whether the Zn-Cl pathway should be considered during EXAFS fitting. The XPS spectra for the Cl 2p of Zn-SA/CNCl samples were shown in **Figs. 1e-1g**. The ionic Cl from Zn-Cl bond of Zn-SA/CNCl-850, Zn-SA/CNCl-920 and Zn-SA/CNCl-1000 were 56.1%, 43.8% and 6.9%, respectively. Since the noticeable peak intensity of ionic Cl from Zn-Cl bond exists in Zn-SA/CNCl-850 and Zn-SA/CNCl-920, the inclusion of Zn-Cl pathway during the EXAFS fitting of Zn-SA/CNCl-850 and Zn-SA/CNCl-920 is necessary and rational.

By contrast, compared with Zn-SA/CNCl-850 and Zn-SA/CNCl-920, the ionic Cl of Zn-SA/CNCl-1000 is only 6.9% with trace amounts and the signal has poor signal-to-noise ratio. Therefore, we conclude that the Zn-Cl bond exists in Zn-SA/CNCl-850 and Zn-SA/CNCl-920 samples while it is difficult to determine whether the Zn-Cl bond exists in Zn-SA/CNCl-1000 sample.

We also attempt to introduce the Zn-Cl pathway during the EXAFS fitting of Zn-SA/CNCl-1000. As shown in **Table R1**, the corresponding fitting result exhibited that the coordination numbers of Zn-N bond and Zn-Cl bond were 4.18 and -0.16, respectively. The coordination number of Zn-Cl bond was negative (-0.16), which indicated that introducing the Zn-Cl pathway during the EXAFS fitting of Zn-SA/CNCl-1000 was inappropriate.

Similarly, we also introduced the Zn-Cl pathway during the EXAFS fitting of Zn-SA/CNCl-850 and Zn-SA/CNCl-920 by the same fitting method. The coordination numbers of Zn-Cl bond in Zn-SA/CNCl-850 and Zn-SA/CNCl-920 were 0.36 and 0.27, respectively, revealing that introducing the Zn-Cl pathway for fitting Zn-SA/CNCl-850 and Zn-SA/CNCl-920 was appropriate.

Therefore, the Zn-Cl pathway was not involved during the EXAFS fitting of Zn-SA/CNCl-1000, which was different from those of Zn-SA/CNCl-850 and Zn-SA/CNCl-920.

Table R1. The structural parameters for fitting Zn-SA/CNCl-1000 when introducing Zn-Cl pathway. ($S_0^2 = 0.78$)

Sample	Shell	N	R(Å)	$\sigma^2(\text{Å}^2)$	$\Delta E_0(\text{eV})$	R factor
Zn-SA/CNCl-1000	Zn-N	4.18 ± 0.17	2.047 ± 0.003	0.00985 ± 0.0008	4.96 ± 0.75	0.00098
	Zn-Cl	-0.16 ± 0.08	2.192 ± 0.041			
Zn-SA/CNCl-920	Zn-N	3.99 ± 0.12	2.004 ± 0.012	0.0097 ± 0.0003	-1.79 ± 0.26	0.0010394
	Zn-Cl	0.27 ± 0.05	2.215 ± 0.015			
Zn-SA/CNCl-850	Zn-N	4.07 ± 0.18	2.023 ± 0.003	0.0106 ± 0.0005	0.43 ± 0.31	0.0026757
	Zn-Cl	0.36 ± 0.08	2.217 ± 0.021			

We also performed the EXAFS measurement of ZnCl₂ as reference sample. The comparison of FT-EXAFS spectra in R space of Zn-SA/CNCl samples and ZnCl₂ was exhibited in **Figure R4**. The prominent peak at around 1.8 Å of ZnCl₂ was attributable to the Zn-Cl bond. As the pyrolysis temperatures of Zn-SA/CNCl samples increasing from 850°C to 1000°C, the intensity of main peak gradually decreased and the peak value gradually moved toward lower R space. Considering the Zn-Cl bond length is longer than that of Zn-N bond length, the gradual shift of peak value toward lower R space is attributable to the transformation of Zn-N₄Cl₁ site into the Zn-N₄ site. The gradually reduced intensity of main peak was attributable to the gradual disappearance of Zn-Cl bond. We added the **Figure R4** as the **Supplementary Fig. 16** in the revised Supplementary Information and Manuscript (Line 186-187).

Combination of the XPS measurement and EXAFS fitting results, we concluded that the Zn-Cl bond and Zn-N bonds co-existed in the Zn-SA/CNCl-850 and Zn-SA/CNCl-920. While the Zn-Cl bond was negligible in Zn-SA/CNCl-1000.

As shown in **Table R1**, the coordination number of Zn-N bond and Zn-Cl bond from Zn-SA/CNCl-850 were 4.07 and 0.36, respectively. The coordination number of Zn-N bond and Zn-Cl bond from Zn-SA/CNCl-920 were 3.99 and 0.27, respectively. These results indicated the Zn-N₄ site and Zn-N₄Cl₁ site co-existed in Zn-SA/CNCl-850 and Zn-SA/CNCl-920. While the coordination numbers of Zn-Cl bond from Zn-SA/CNCl-850 and Zn-SA/CNCl-920 were much lower than that of Zn-N bond. Therefore, the Zn-N₄ site is still the predominant component in both the Zn-SA/CNCl-850 and Zn-SA/CNCl-920 rather than Zn-N₄Cl₁ site.

In summary, the Zn element predominantly existed as the Zn-N₄ site in both Zn-SA/CNCl-850, Zn-SA/CNCl-920 and Zn-SA/CNCl-1000 with only partial Zn-N₄Cl₁ site in Zn-SA/CNCl-850 and Zn-SA/CNCl-920, which led to the minor differences between Zn-SA/CNCl samples at different temperatures.

Figure R4. The comparison of FT-EXAFS spectra in R space of Zn-SA/CNCl-850, Zn-SA/CNCl-920, Zn-SA/CNCl-1000 and ZnCl₂ as reference sample.

Comment 2: Overall, the employed fitting model is not clear. How many different paths were included in the fit? The R-factors reported in Supplementary Table 1 all seem to be very small, which is highly unusual for such disordered system. All the fitting results are reported without uncertainties, which makes them essentially useless. Why Zn-Cl bond distance differs so strongly for the samples treated at 850 deg C and 920 deg C? It seems that the disorder factors for Zn-Cl and Zn-N bonds were constrained to be the same-this should be clearly stated in text. Were any other constraints employed? Please show also EXAFS fitting results in k-space for all samples, not just for one as is currently done in Figure S16.

Reply: We sincerely appreciate the comments provided by Reviewer #2.

Combination of the XPS measurement and EXAFS fitting results, we concluded that the Zn-Cl bond and Zn-N bonds co-existed in the Zn-SA/CNCl-850 and Zn-SA/CNCl-920. While the Zn-Cl bond was negligible in Zn-SA/CNCl-1000.

Therefore, the Zn-N path and Zn-Cl path were considered together for fitting the Zn-SA/CNCl-850 and Zn-SA/CNCl-920. While only the Zn-N path was involved for fitting Zn-SA/CNCl-1000. The Cl element was absent in the Zn-SA/CN-1000 as control sample. Therefore, only the Zn-N path was considered for fitting Zn-SA/CN-1000.

The R-factors reflected the goodness of fitting results. The smaller R-factors, the better fitting results. Besides, the goodness of fitting results is relative to the signal quality of collected EXAFS spectra of Zn-based samples.

The EXAFS measurement was performed at room temperature by fluorescence mode. As shown in **Table R2**, the Zn contents of Zn-SA/CNCl-850, Zn-SA/CNCl-920, Zn-SA/CNCl-1000 and Zn-SA/CN-1000 were 4.23 wt%, 2.65 wt%, 1.13 wt% and 3.24 wt%, respectively, higher than most of metal contents from the reported metal single-atom sites catalysts. Therefore, the fluorescence mode and the higher Zn loading in the catalysts lead to the higher signal quality of EXAFS spectra and the smaller R-factors during EXAFS fitting.

We sincerely appreciate the comments provided by Reviewer #2 to recheck the rationality of R-factors. The R-factor of Zn-SA/CNCI-850 was 0.0000530 in the original Supplementary Table 1, which seemed to be problematic. Therefore, we carefully rechecked the EXAFS fitting results of all Zn-based samples. We sincerely apologized that the parameter value of dk was set to 1 in error during the EXAFS fitting of Zn-SA/CNCI-850. After correcting the parameter value of dk ($dk = 0.5$), the same as other Zn-based samples, the corrected R-factor of Zn-SA/CNCI-850 was 0.0026757. After careful examination, the EXAFS fitting results of other Zn-based samples were rational. We extend our sincere appreciation to the reviewer for the time, expertise, and constructive feedback, which have undoubtedly contributed to the improvement of our manuscript.

The corrected fitting results of Zn-SA/CNCI-850 were shown in **Figure R5** and **Table R1**. Therefore, we corrected the EXAFS fitting results of Zn-SA/CNCI-850 in the revised **Fig. 2d** and **Supplementary Table 1** in the revised Manuscript and Supplementary Information.

Figure R5. **a**, The fitting result of Zn-SA/CNCI-850 in k space. **b**, The fitting result of Zn-SA/CNCI-850 in R space.

Table R1. The structural parameters for fitting Zn-SA/CNCl-850. ($S_0^2 = 0.78$)

Sample	Shell	N	R(\AA)	$\sigma^2(\text{\AA}^2)$	$\Delta E_0(\text{eV})$	R factor
Zn-SA/CNCl-850	Zn-N	4.07 ± 0.18	2.023 ± 0.003	0.0106 ± 0.0005	0.43 ± 0.31	0.0026757
	Zn-Cl	0.36 ± 0.08	2.217 ± 0.021			

As shown in **Table R3**, we compared the R factors for EXAFS fitting and metal contents between Zn-SA/CNCl-1000 and other reported metal single-atom sites catalysts. The R factor and metal content of Zn-SA/CNCl-1000 were 0.0022350 and 1.13 wt% Zn, which were comparable to those of other reported metal single-atom sites catalysts. For example, the R factor and metal content of Fe-ISAs/CN (*Angew. Chem. Int. Ed.* **2017**, *56*, 6937.) were 0.0005 and 2.16 wt% Fe. The R factor and metal content of Fe-SAs/NPS-HC (*Nat. Commun.* **2018**, *9*, 5422.) were 0.0026 and 1.54 wt% Fe. Therefore, the R factors in the Zn-based samples were reasonable, compared with other reported metal single-atom sites catalysts.

Table R2. The summary of R factor and Zn content of different Zn-based samples.

Sample	R factor	Zn Content (wt%)
Zn-SA/CNCI-850	0.0026757	4.23
Zn-SA/CNCI-920	0.0010394	2.65
Zn-SA/CNCI-1000	0.0022350	1.13
Zn-SA/CNCI-1000 (CO ₂ RR@OCV)	0.0021748	1.13
Zn-SA/CNCI-1000 (CO ₂ RR@-0.9V)	0.0052816	1.13
Zn-SA/CN-1000	0.0002944	3.24
Zn-SA/CN-1000 (CO ₂ RR@OCV)	0.0005161	3.24
Zn-SA/CN-1000 (CO ₂ RR@-0.9V)	0.0005811	3.24

Table R3. The comparison of R factors for EXAFS fitting and metal contents between Zn-SA/CNCI-1000 and other reported metal single-atom sites catalysts.

Sample	R factor	Metal content (wt%)	Reference
Zn-SA/CNCI-1000	0.0022350	Zn: 1.13	This work
FeN ₃ P-SAzyme	0.003	Fe: 2.59	Nat. Catal. 2021 , 4 , 407-417.
FeN ₄ -SAzyme	0.001	Fe: 1.98	
Fe-SAs/NPS-HC	0.0026	Fe: 1.54	Nat. Commun. 2018 , 9 , 5422.
Co ₁ -N ₃ PS/HC	0.007	Co: 0.39	Angew. Chem. Int. Ed. 2021 , 60 , 3212.
Co-N ₄ /HC	0.005	Co: 0.42	
Fe-ISAs/CN	0.0005	Fe: 2.16	Angew. Chem. Int. Ed. 2017 , 56 , 6937.
In-SAs/NC	0.003	In: 1.25	Angew. Chem. Int. Ed. 2020 , 59 , 22465.

We sincerely appreciate the recommendation from Reviewer #2 to provide the uncertainties for fitting results, which have undoubtedly contributed to the improvement of our manuscript.

As shown in the revised **Supplementary Table 1**, we added the uncertainties for fitting results of Zn-based samples.

The revised **Supplementary Table 1**. Structural parameters extracted from the Zn K-edge EXAFS fitting. ($S_0^2 = 0.78$)

Sample	Shell	N	R(Å)	$\sigma^2(\text{\AA}^2)$	$\Delta E_0(\text{eV})$	R factor
Zn foil	Zn-Zn ¹	6*	2.644 ± 0.005	0.0096 ± 0.0004	3.64 ± 0.66	0.0007218
	Zn-Zn ²	6*	2.766 ± 0.007	0.0185 ± 0.0023		
Zn-SA/CNCl-850	Zn-N	4.07 ± 0.18	2.023 ± 0.003	0.0106 ± 0.0005	0.43 ± 0.31	0.0026757
	Zn-Cl	0.36 ± 0.08	2.217 ± 0.021			
Zn-SA/CNCl-920	Zn-N	3.99 ± 0.12	2.004 ± 0.012	0.0097 ± 0.0003	-1.79 ± 0.26	0.0010394
	Zn-Cl	0.27 ± 0.05	2.215 ± 0.015			
Zn-SA/CNCl-1000	Zn-N	4.02 ± 0.14	2.045 ± 0.003	0.0098 ± 0.0005	4.22 ± 0.39	0.0022350
Zn-SA/CN-1000	Zn-N	4.01 ± 0.08	2.001 ± 0.001	0.0078 ± 0.0002	-2.80 ± 0.24	0.0002944
Zn-SA/CNCl-1000 (CO ₂ RR@OCV)	Zn-N	4.04 ± 0.17	2.030 ± 0.003	0.0094 ± 0.0006	3.69 ± 0.42	0.0021748
Zn-SA/CNCl-1000 (CO ₂ RR@-0.9V)	Zn-N	3.44 ± 0.35	2.008 ± 0.007	0.0134 ± 0.0014	2.24 ± 0.89	0.0052816
Zn-SA/CN-1000 (CO ₂ RR@OCV)	Zn-N	4.31 ± 0.11	2.024 ± 0.002	0.0100 ± 0.0003	2.43 ± 0.23	0.0005161
Zn-SA/CN-1000 (CO ₂ RR@-0.9V)	Zn-N	4.07 ± 0.09	2.021 ± 0.002	0.0093 ± 0.0003	2.70 ± 0.22	0.0005811

We sincerely appreciate the comment provided by Reviewer #2 regarding the difference of Zn-Cl bond length between Zn-SA/CNCl-850 and Zn-SA/CNCl-920 in **Supplementary Table 1**. After correcting the EXAFS fitting result of Zn-SA/CNCl-850, the Zn-Cl bond length of Zn-SA/CNCl-850 was 2.217 Å, which was similar to that of Zn-SA/CNCl-920 (2.215 Å).

We sincerely appreciate the comments provided by Reviewer #2 regarding the constraint of the disorder factors for Zn-Cl and Zn-N bonds. The Zn-Cl and Zn-N bonds were both in the first coordination shells of Zn atom and their bond lengths were similar. Therefore, the difference of disorder factors between Zn-Cl and Zn-N bonds was inapparent.

As shown in **Figure R6**, the reference (*Energy Environ. Sci.* **2018**, *11*, 2348-2352.) reported the FeCl₁N₄/CNS catalyst. The disorder factors of Fe-N bond (1.92 Å) and Fe-Cl bond (2.19 Å) were 0.0065 Å² and 0.0072 Å², respectively, which were similar to each other.

Table S1. Structural parameters of FeCl₁N₄/CNS extracted from the EXAFS fitting.

$$(|S_{0=0.85}^2|)$$

Sample	Scattering Pair	CN	R (Å)	σ^2 (10 ⁻³ Å ²)	ΔE_0 (eV)	R factor
FeCl ₁ N ₄ /CNS	Fe-N	1.4	1.92	6.5	2.5	0.0048
	Fe-NCl	3.6	2.19	7.2	3.1	
Fe foil	Fe-Fe ₁	8	2.46	4.4	4.6	0.0063
	Fe-Fe ₂	6	2.84	4.5	4.1	

Figure R6. The Table S1 from the reference (*Energy Environ. Sci.* **2018**, *11*, 2348-2352.).

As shown in **Figure R7**, the reference (*Nat. Catal.* **2021**, *4*, 407-417.) reported the FeN₃P-SAzyme. The disorder factors of Fe-N/O bond (1.97 Å) and Fe-P bond (2.26 Å) in the first coordination shells of Fe were 0.007 Å² and 0.008 Å², respectively, which were similar to each other. While the disorder factor of Fe-C bond

(2.80 Å) in the second coordination shells of Fe was 0.021 Å².

Supplementary Table 1. Fe K-edge EXAFS curve Fitting Parameters^a

sample	path	N	R (Å)	σ^2 (Å ²)	ΔE_0 (eV)	R , %
Fe foil ^b	Fe–Fe1	8	2.47	0.006	0.7	0.01
	Fe–Fe2	6	2.85	0.007		
FePc ^c	Fe–N	4	1.93	0.004	3.2	0.01
FeN ₃ P-SAzyme ^d	Fe–N/O	3.6	1.97	0.007	0.6	0.3
	Fe–P	1.1	2.26	0.008		
	Fe–C	4.3	2.80	0.021		
FeN ₄ -SAzyme ^d	Fe–N/O	5.8	2.00/	0.009	4.1	0.1
	Fe–C	4.2	2.76	0.040		

Figure R7. The Supplementary Table 1 from the reference (*Nat. Catal.* **2021**, *4*, 407-417.).

Therefore, due to the similar disorder factors of different bonds in the first coordination shells, some recently reported references also constrained the disorder factors of different bonds in the first coordination shells during EXAFS fitting.

For example, as shown in **Figure R8**, the reference (*Angew. Chem. Int. Ed.* **2021**, *60*, 27324.) reported the FeN₄Cl₁/NC catalyst. The disorder factors of Fe-N bond (2.05 Å) and Fe-Cl bond (2.19 Å) in the first coordination shell of Fe were both 0.0135 Å².

Table S5. EXAFS fitting parameters at the Fe K-edge for various samples ($S_0^2=0.74$).

Sample	Shell	$N^{[a]}$	$R(\text{Å})^{[b]}$	$\sigma^2 \times 10^3 (\text{Å}^2)^{[c]}$	ΔE_0 (eV) ^[d]	R factor
Fe foil	Fe-Fe	8*	2.471	4.8	6.4	0.002
	Fe-Fe	6*	2.85	5.9	5.2	
FeCl ₂	Fe-Cl	2.2	2.20	8.4	-10.3	0.001
	Fe-Cl	3.8	2.48	17.8	-1.8	
FeCl ₃	Fe-Cl	6.1	2.23	10.6	-11.4	0.006
FePc	Fe-N	4.0	1.99	8.1	8.0	0.012
	Fe-C	4.8	2.98	6.6	6.3	
FeN _x /NC	Fe-N	4.2	2.04	15.4	2.3	0.015
FeN _x Cl _y /NC	Fe-N	3.5	2.05	13.5	-0.3	0.019
	Fe-Cl	1.2	2.19			

[a] N : coordination numbers; [b] R : bond distance; [c] σ^2 : Debye-Waller factors; [d] ΔE_0 : the inner potential correction. R factor: goodness of fit.

Figure R8. The Table S5 from the reference (*Angew. Chem. Int. Ed.* **2021**, *60*, 27324.).

As shown in **Figure R9**, the reference (*J. Am. Chem. Soc.* **2020**, *142*, 2404-2412.) reported the Fe-N/P-C-700 catalyst. The disorder factors of Fe-N bond (1.99 Å) and Fe-P bond (2.35 Å) in the first coordination shell of Fe were both 0.00249 Å².

Table S2. EXAFS data fitting results of Fe-N/P-C-700 and Fe-N/P-C-800.

Sample	Path	N	$R(\text{Å})$	σ^2 (10^{-3}Å^2)	ΔE_0	R-factor
Fe-N/P-C-700	Fe-N	3.1±0.3	1.99	2.49±0.18	-2.0 ± 1.1	0.02
	Fe-P	0.9±0.1	2.35	2.49±0.18		
Fe-N/P-C-800	Fe-P	2±0.2	2.19	8.71±0.45	2.4 ± 3.4	0.017
	Fe-P	4±0.3	2.28	8.71±0.45		
	Fe-Fe	2±0.3	2.57	2.22±0.30		
	Fe-Fe	4±0.4	2.67	2.22±0.30		

Figure R9. The Table S2 from the reference (*J. Am. Chem. Soc.* **2020**, *142*, 2404-2412.).

As shown in **Figure R10**, the reference (*Adv. Funct. Mater.* **2022**, *32*, 2209499.) reported the Co-N₄-O/MX catalyst. The disorder factors of Co-N bond (1.91 Å) and Co-O bond (2.10 Å) in the first coordination shell of Co were both 0.0101 Å².

Table S2. EXAFS fitting parameters at the Co K-edge ($S_0^2=0.84$)

Sample	Path	C.N.	R (Å)	$\sigma^2 \times 10^3$ (Å ²)	ΔE (eV)	R factor
Co foil	Co-Co	12*	2.49±0.01	6.4±0.1	8.6±0.2	0.001
	Co-N	3.9±2.9	1.91±0.04			
	Co-O	0.8±0.6	2.10±0.15	10.1±6.3	-2.0±3.6	0.017

C.N.: coordination numbers; R : bond distance; σ^2 : Debye-Waller factors; ΔE : the inner potential correction. R factor: goodness of fit.

Figure R10. The Table S2 from the reference (*Adv. Funct. Mater.* **2022**, *32*, 2209499.).

Therefore, the disorder factors of Zn-Cl and Zn-N bonds were constrained to be the same during the EXAFS fitting. This fitting method was appropriate since all the Zn-based samples were fitted by the same fitting method. Therefore, the fitting results of all the Zn-based sample can be compared with each other.

We sincerely appreciate the comments provided by Reviewer #2 and we add the description about the constraint of the disorder factors of Zn-Cl and Zn-N bonds in the revised **Supplementary Table 1**. No other parameters were fixed, constrained, or correlated during EXAFS fitting.

We sincerely appreciate the comments provided by Reviewer #2 regarding showing the EXAFS fitting results in k-space for all samples. The fitting results in k-space for Zn-SA/CNCl-850, Zn-SA/CNCl-920, Zn-SA/CNCl-1000 and Zn-SA/CN-1000 were exhibited in **Figure R11**. We added the fitting results in k-space for Zn-based samples as **Supplementary Figure 17** in the revised

Supplementary Information.

Figure R11. a-d, The fitting results in k-space for Zn-SA/CNCI-850, Zn-SA/CNCI-920, Zn-SA/CNCI-1000 and Zn-SA/CN-1000, respectively.

Comment 3: Furthermore, after all these troubles the main conclusion based on the ex-situ EXAFS data analysis seems to be that the sample treated at 1000 deg C does not have any Zn-Cl bonds, and its local structure is remarkably similar to that of the sample prepared in the absence of chlorine. So, how then authors can conclude that chlorine is actually doing something for the catalytic mechanism? How can they claim that the C-Cl bonds are indeed located in the vicinity of Zn species?

To conclude, major revision is necessary, since the author's conclusions do not seem to be supported by experimental data.

Reply: We sincerely appreciate the comments provided by Reviewer #2 regarding the catalytic role of chlorine element in Zn-SA/CNCl-1000 during catalysis.

By combination of the XPS measurement for Cl 2p and the EXAFS fitting results, we revealed the Zn-Cl bond was negligible in the Zn-SA/CNCl-1000. The Zn element in both Zn-SA/CNCl-1000 and Zn-SA/CN-1000 existed as the Zn-N₄ sites, which were analyzed by EXAFS fitting results.

As shown in **Fig. 1g**, the existence of Cl element in the Zn-SA/CNCl-1000 was confirmed by XPS measurement for Cl 2p. The Cl element in the Zn-SA/CNCl-1000 mainly existed as the covalent Cl from C-Cl bonds. Therefore, in the Zn-SA/CNCl-1000, the Zn-N₄ sites were anchored on N, Cl-co-doped graphene with C-Cl bonds. While the Zn-N₄ sites in Zn-SA/CN-1000 were anchored on N-doped graphene without C-Cl bonds.

Although both the Zn element in Zn-SA/CNCl-1000 and Zn-SA/CN-1000 existed as Zn-N₄ sites, the chemical environment of Zn-N₄ sites were different. Therefore, the existence of C-Cl bonds in Zn-SA/CNCl-1000 has a potential effect on the catalytic performance of Zn-N₄ sites during catalysis.

Fig. 1g. The XPS spectrum for the Cl 2p of Zn-SA/CNCl-1000.

As shown in **Fig. 4**, the in-situ XANES and EXAFS measurements for CO₂RR catalyzed by Zn-SA/CNCl-1000 and Zn-SA/CN-1000 were performed under the same conditions. At -0.9 V vs. RHE, the Zn-N₄ sites in Zn-SA/CNCl-1000 were in situ transformed into low-coordination Zn-N₃ catalytic sites. While at -0.9 V vs. RHE, the Zn-N₄ sites in Zn-SA/CN-1000 also existed as the Zn-N₄ sites.

The different chemical environment of Zn-N₄ sites between Zn-SA/CNCl-1000 and Zn-SA/CN-1000 induced different structural changes of Zn-N₄ sites for CO₂RR. Therefore, the existence of C-Cl bonds in Zn-SA/CNCl-1000 played an important role during the in situ structural evolution from the Zn-N₄ sites into Zn-N₃ sites for CO₂RR.

As shown in **Fig. 5**, the DFT calculation revealed that the adjacent C-Cl bond effectively induced the structural evolution of Zn-N₄ site into Zn-N₃ site, which was consistent with the in-situ XANES and EXAFS measurements for CO₂RR. As shown in **Fig. 5c**, compared with the Zn-N₄ site without adjacent C-Cl bond, the in-situ formed Zn-N₃ catalytic site induced by adjacent C-Cl bond, exhibited much higher activity for CO₂RR, which was consistent with the experimental results for CO₂RR in **Fig. 3**.

Therefore, by combination of in-situ XANES and EXAFS measurements for CO₂RR and DFT calculation, we concluded that the C-Cl bonds in Zn-SA/CNCl-1000 played an important role to induce the structural evolution of Zn-N₄ site into Zn-N₃

site, which remarkably boosted the catalytic activity for CO₂RR.

In order to demonstrate the C-Cl bond was favourable to be located in the vicinity of Zn-N₄ site, we constructed ten possible Zn-N₄/CNCl models with C-Cl bond at different positions, as shown in **Supplementary Fig. 32**. We compared the relative energies of different Zn-N₄/CNCl models. The Zn-N₄/CNCl-1 was the most stable model with the lowest energy defined as 0 eV while the Zn-N₄/CNCl-2 was the second stable model with energy of 1.51 eV. As the C-Cl bond moved away from the Zn-N₄ sites, the relative energy increased, demonstrating the more stable models with C-Cl bond adjacent to Zn-N₄ sites.

Supplementary Fig. 32. The optimized structures of ten possible Zn-N₄/CNCl models with C-Cl bond at different positions and the comparison of relative energies.

As shown in **Fig. 4** and **Fig. 5**, the in-situ XANES and EXAFS measurements for CO₂RR and DFT calculation revealed the adjacent C-Cl bond of Zn-N₄ site in Zn-SA/CNCl-1000 effectively induced the in situ structural evolution of Zn-N₄ site into Zn-N₃ site. As the C-Cl bond gradually moved away from the Zn-N₄ site, the effect of C-Cl bond on Zn-N₄ site gradually decreased, which was unfavourable to induce the in situ structural evolution of Zn-N₄ site into Zn-N₃ site.

By comparing the relative energies of different Zn-N₄/CNCl models with C-Cl bond at different positions, the Zn-N₄/CNCl models with C-Cl bond adjacent to Zn-N₄ site were more easily to be formed, which was favourable to induce the in situ structural evolution of Zn-N₄ site into Zn-N₃ site, consistent with the results of in-situ XANES and EXAFS measurements for CO₂RR.

Reviewer #3:

Comment : In this article, Wei et al reported distorted Zn-N₃ site which is reconstructed from Zn-N₄ moiety by the presence of Cl and found to be highly active for CO₂ reduction reaction. The Zn-N₃ site with broken symmetry believed to strengthen the adsorption of COOH intermediate (supported by DFT calculations) which is responsible for improving the catalytic activity. The distorted Zn single site showed around 90% Faradaic efficiency for the conversion of CO₂ to CO in the potential range from -0.63V to -0.93 V Versus RHE. The authors did extensive ex-situ characterization of the materials via XAS, STEM, XRD, XPS, RAMAN which is appreciated. Moreover, they performed some in-situ XAS and SEIRAS spectroscopy to understand the mechanistic details. However, in my opinion, there are many studies conducted in this direction in the past and the major concern about whether the single-atom catalyst remains as it is or agglomerates under reaction conditions. Such as Yang et al found that Ni(I) center is highly dynamic in nature under operando conditions (<https://www.nature.com/articles/s41560-017-0078-8>). Even Zn SACs are already reported to be active for electrochemical CO₂ reduction with high Faradaic efficiency (<https://doi.org/10.1002/anie.201805871>). Hence, I found a lack of novelty in this work to be published in such a prestigious journal. I have the following comments that the authors can take into account to improve their manuscript in the future.

Reply: We sincerely appreciate the comments provided by Reviewer #3 regarding the novelty of our work. We extend our sincere appreciation to the reviewer for the time, expertise, and constructive feedback, which have undoubtedly contributed to the improvement of our manuscript.

Recently, metal isolated single-atom site (ISAS) catalysts with metal-N₄ sites have been extensively utilized for CO₂RR, including the Zn-N₄ site (<https://doi.org/10.1002/anie.201805871>). However, the rigid and inflexible structure of metal-N₄ site on carbon plane is unfavourable for regulating the geometric configuration and electronic distribution of catalytic site, which limits the further

improvement of catalytic performance. (*Angew. Chem. Int. Ed.* **2023**, *62*, e202215136; *Nat. Commun.* **2020**, *11*, 3049.) Therefore, how to optimize the traditional metal-N₄ site at atomic level, inducing the transformation from low active site into highly active site and boosting CO₂RR activity by orders of magnitude, is significant and challenging.

Herein, we firstly proposed and designed the planar chlorination engineering of metal ISAS catalysts for successfully converting the traditional Zn-N₄ site with low activity and selectivity for CO₂RR into highly active Zn-N₃ site by breaking the geometric symmetry of traditional metal-N₄ sites.

The optimal catalyst Zn-SA/CNCl-1000 displayed faradaic efficiency for CO (FE_{CO}) above 90% over a broad potential window of 300 mV and excellent stability during 50 h test at high current density of 200 mA/cm², with enormous potential for promising application in industrial catalysis. At -0.93 V vs. RHE, the partial current density of CO (*J*_{CO}) and the turnover frequency (TOF) value catalyzed by Zn-SA/CNCl-1000 were 271.7 mA/cm² and 29325 h⁻¹, around 29 times and 83 times those of Zn-SA/CN-1000 without planar chlorination engineering.

In-situ EXAFS measurements and DFT calculation demonstrated the adjacent C-Cl bond induced the self-reconstruction of Zn-N₄ site with low activity and selectivity for CO₂RR into the highly active Zn-N₃ sites with broken symmetry, strengthening the adsorption of *COOH intermediate, and thus remarkably improving CO₂RR activity.

This work reveals that planar chlorination engineering has enormous potential for improving the CO₂RR activity and promising application in industrial catalysis by breaking the geometric symmetry of traditional metal-N₄ sites.

Comment 1: *There are no details of in-situ XAS measurements such as what is exactly a Lytle detector? What was the gas flow? Did the author perform a potential dependent study? Usually presence of bubble leads to noisy XAS spectrum. How did the authors overcome this? Please see the review by Sarma et al. (<https://doi.org/10.1021/acs.chemrev.2c00495>).*

Reply: We appreciate the reviewer’s careful reading and valuable comments. As shown in **Figure R12a**, to conduct our in-situ EXAFS experiments for CO₂RR, we set up three ionization chambers for I₀, I_t, and I_f, respectively, similar to the regular setup. A Lytle detector was employed to measure the fluorescence signals. The Lytle detector is also designed as an ionization chamber with a large collection area. It contains a fan-shaped slit to block the scattered X-ray and a filter to reduce the K_β fluorescence signals (In our Zn K-edge case, Cu foil was used for the filter). The in-situ cell was mounted between the I₀ and I_t chambers and tilted 45 degrees toward the fluorescence detector. Outside the cell, CO₂ gas was introduced into the 0.5 M KHCO₃ electrolyte at a rate of 60 c.c./min for 30 minutes to produce a saturated solution and kept the flow throughout the entire experiment.

Figure R12. a, The photograph of the in-situ EXAFS experiments for CO₂RR. b-c, The photographs of the in-situ cell. d, The schematic diagram of capillary action, allowing for quick removal of electrochemically generated bubbles.

We sincerely appreciate the comments provided by Reviewer #3 regarding the potential dependent study. As shown in **Figure R13**, in-situ XANES and EXAFS measurements were carried out for CO₂RR catalyzed by Zn-SA/CNCI-1000 catalyst

at OCV, -0.7 V, -0.8 V, -0.9 V and -1.0 V vs. RHE. As shown in **Figure R13a**, compared with the XANES curve at OCV, the intensities of XANES white-line (WL) at -0.7 V and -0.8 V vs. RHE had an obvious decline. The intensities of XANES white-line (WL) at -0.9 V and -1.0 V vs. RHE further decreased compared with those at -0.7 V and -0.8 V vs. RHE. The corresponding FT-EXAFS spectra in R space were shown in **Figure R13b**. As the applied potentials gradually decreased from OCV to -1.0 V vs. RHE, the intensities of main peak from Zn-N bond at around 1.5 Å gradually decreased correspondingly, revealing the gradual transformation from Zn-N₄ site into low-coordinated Zn-N₃ site. As shown in **Fig. 3b** and **Fig. 3c**, as the the applied potentials gradually decreased from -0.57 V to -0.93 V vs. RHE, both the J_{CO} and TOF values were remarkably increased, which was ascribed to the gradual transformation from Zn-N₄ site into low-coordinated Zn-N₃ site. We added the **Figure R13** as **Supplementary Fig. 31** in the revised Supplementary Information and Manuscript (Line 316-320).

Figure R13. **a**, The XANES spectra for CO₂RR catalyzed by Zn-SA/CNCl-1000 at OCV, -0.7 V, -0.8 V, -0.9 V and -1.0 V vs. RHE. **b**, The corresponding FT-EXAFS spectra in R space at OCV, -0.7 V, -0.8 V, -0.9 V and -1.0 V vs. RHE.

Fig. 3. The catalytic performance for CO₂RR of Zn-based catalysts. **b**, The comparison of J_{CO} during CO₂RR. **c**, The comparison of TOF values.

As shown in **Figure R12d**, the peristaltic pump continuously delivers CO₂-saturated electrolytes into the cell from the bottom side and exits from above. We coated the sample on carbon paper and covered it with a Kapton window. An X-ray penetrable tube is inserted close to the sample, leaving a small gap. This small gap will create capillary action, allowing for quick removal of electrochemically generated bubbles. Therefore, the absorption spectrum could maintain its quality.

We added the **Figure R12** as **Supplementary Fig. 29** in the revised Supplementary Information and Manuscript (Line 292-293).

Comment 2: The authors conducted in-situ ATR-SEIRAS measurements over Si prism and claimed that there are C=O and C-OH features. However, when these experiments were conducted authors should keep in mind that Si absorbs the IR strongly in the range of 1500-1000 cm⁻¹. Hence any data in this range might not be so meaningful and might have occurred due to some artifact. A more interesting region will be Metal-Carbonyl which appears in the range of 1900-2100 cm⁻¹. Also in the zero voltage potential why do there are already some features? May be these features come from some organic that is already present in the Nafion or residue from ethanol. For more details please see <https://doi.org/10.1021/acs.nanolett.9b01582>

Reply: We sincerely appreciate the comments provided by Reviewer #3 regarding the in-situ ATR-SEIRAS measurements.

When Si prism is used as the IR window, the oxide layer on the surface of Si prism also has adsorption peaks, such as the adsorption peaks of Si-O-Si and Si-OH. As reported by the reference (*Journal of Cleaner Production* **2022**, 380, 134975), the band of the vibration of Si-O-Si bonds located at 1095 cm^{-1} while the band of the stretching vibration of Si-OH appeared at 950 cm^{-1} . As reported by the reference (*Appl. Phys. A* **2018**, 124, 802), the IR characteristic peaks of the Si-O-Si bond are all strong stretching vibration absorption peaks, which appear at 527 cm^{-1} , 465 cm^{-1} , 1095 cm^{-1} and 1020 cm^{-1} . Thus, we analyze the peaks distribution above 1100 cm^{-1} to avoid the disruptions of Si signals. Besides, in the pre-treatment of Si prism, we have used 40% NH_4F to remove the oxide layer on Si surface.

As shown in **Supplementary Fig. 28**, we further analyze the adsorption range to 2200 cm^{-1} . At the potential of 0 V and -0.3 V vs. RHE, a adsorption at 2110 cm^{-1} was observed, which is attributed the terminally CO (CO_{atop}). This peak quickly disappeared when the applied potential shift to -0.6 V vs. RHE because the CO_{atop} was quickly desorbed from electrode surface and was converted into gaseous CO, which was favourable for the formation of gaseous CO as main product.

Additionally, from -0.6 V to -1.2 V vs. RHE, the peak intensities at 1945 cm^{-1} corresponding to bridge CO ($\text{CO}_{\text{bridge}}$) were rather weak, without obvious enhancement as the potential decreased from -0.6 V to -1.2 V vs. RHE. As confirmed by references (*Proc. Natl. Acad. Sci. USA*. **2016**, 113, E4585-E4593; *Nano Lett.* **2019**, 19, 4817-4826), the $\text{CO}_{\text{bridge}}$ is inert intermediate and hard to be converted to gaseous CO, while CO_{atop} is active and can be quickly desorbed. Therefore, we conclude that CO_2RR on Zn-SA/CNCI-1000 proceeds through CO_{atop} rather than $\text{CO}_{\text{bridge}}$, facilitating the formation of gaseous CO as main product, which was consistent with the experimental results for CO_2RR .

We updated the **Supplementary Fig. 28** and added the above analysis in the revised Supplementary Information.

Supplementary Fig. 28. The in situ ATR-SEIRAS measurement for CO₂RR catalyzed by Zn-SA/CNCl-1000 at OCV, 0.0 V, -0.3 V, -0.6 V, -0.7 V, -0.8 V, -0.9 V, -1.0 V, -1.1 V, and -1.2 V vs. RHE. The potentials are provided without iR correction.

In our ATR-SEIRAS test, we used CO₂ saturated 0.5 M KHCO₃ solution as electrolyte, and high pure CO₂ was continuously introduced into the electrolyte. In potential conversion, we assume that the pH of the electrolyte is 7. Actually, the pH of electrolyte is slightly below 7, and thus the actual potential is slightly lower than the marked potential. So, we may detect some features at 0 V vs. RHE. However, the intensities of these features at 0 V vs. RHE were rather weak, revealing the weak catalytic role of Zn-SA/CNCl-1000 for CO₂RR at 0 V vs. RHE. By comparison, as the potentials decreased from -0.6 V to -1.2 V vs. RHE, the intensities of adsorption peaks at 1215 cm⁻¹ and 1720 cm⁻¹ had an obvious enhancement, revealing the predominant catalytic role of Zn-SA/CNCl-1000 for CO₂RR from -0.6 V to -1.2 V vs. RHE.

Comment 3: The authors should have include a Zn-Cl reference sample in their XAS measurements which would have helped the readers to follow the XAS results. There is consistent presence of a strong feature in the range of 2-3 Å in the figure 2 (d), (e)

and (f). Does the author know the origin of this feature?

Reply: We sincerely appreciate the comments provided by Reviewer #3 regarding including a Zn-Cl reference sample in the XAS measurements.

As shown in **Figure R14a**, we measured the XANES spectrum of ZnCl_2 as the reference sample. The corresponding FT-EXAFS result in R space of ZnCl_2 was exhibited in **Figure R14b**. The comparison of FT-EXAFS spectra in R space of Zn-SA/CNCl samples and ZnCl_2 was exhibited in **Figure R4**. The prominent peak at around 1.8 Å of ZnCl_2 was attributable to the Zn-Cl bond. As the pyrolysis temperatures of Zn-SA/CNCl samples increasing from 850°C to 1000°C, the intensity of main peak gradually decreased and the peak value gradually moved toward lower R space. Considering the Zn-Cl bond length is longer than that of Zn-N bond length, the gradual shift of peak value toward lower R space is attributable to the transformation of $\text{Zn-N}_4\text{Cl}_1$ site into the Zn-N_4 site. The gradually reduced intensity of main peak was attributable to the gradual disappearance of Zn-Cl bond.

We added the XANES spectrum and the corresponding FT-EXAFS result in R space of ZnCl_2 in the revised **Fig. 2**.

Figure R14. a-b, The XANES spectra and corresponding FT-EXAFS results in R space of Zn-SA/CNCl-850, Zn-SA/CNCl-920, Zn-SA/CNCl-1000, Zn foil, ZnO and ZnCl_2 as reference samples.

Figure R4. The comparison of FT-EXAFS spectra in R space of Zn-SA/CNCl-850, Zn-SA/CNCl-920, Zn-SA/CNCl-1000 and ZnCl₂ as reference sample.

We sincerely appreciate the comments provided by Reviewer #3 regarding the wave in the range of 2-3 Å in **Figs. 2d-2f**. As shown in **Figure R15**, the intensity of the wave in the range of 2-3 Å (red box b) was similar to that of the wave in the range of 0-0.75 Å (red box a). The wave in the range of 0-0.75 Å (red box a) was attributable to the noise rather than signal since any chemical bonds less than 1.0 Å were irrational. Considering the similar intensities of waves in the red boxes a and b, the wave in the range of 2-3 Å (red box b) was also attributable to the noise rather than signal.

The boiling point of bulk Zn is 907°C while the boiling point of Zn nanoparticles will remarkably decrease. As revealed by the reference (*Nat. Commun.* **2023**, *14*, 7549.), the Zn nanoparticles will evaporate above 700°C under inert atmosphere. Therefore, at elevated temperature under inert atmosphere, especially above 850°C, the Zn nanoparticles can not exist in the Zn-SA/CNCl samples.

Figure R15. The comparison of FT-EXAFS spectra in R space of Zn-SA/CNCI-850, Zn-SA/CNCI-920 and Zn-SA/CNCI-1000.

Comment 4: The fitting results of the XAS data in supplementary table 1 lack errors and also details. When is the asterisk or star represented? What did the authors fit in achieving those results?

Reply: We sincerely appreciate the comments provided by Reviewer #3. As shown in the revised **Supplementary Table 1**, we added the uncertainties for fitting results of Zn-based samples.

The revised **Supplementary Table 1**. Structural parameters extracted from the Zn K-edge EXAFS fitting. ($S_0^2 = 0.78$)

Sample	Shell	N	R(Å)	$\sigma^2(\text{Å}^2)$	$\Delta E_0(\text{eV})$	R factor
Zn foil	Zn-Zn ¹	6*	2.644 ± 0.005	0.0096 ± 0.0004	3.64 ± 0.66	0.0007218
	Zn-Zn ²	6*	2.766 ± 0.007	0.0185 ± 0.0023		
Zn-SA/CNCl-850	Zn-N	4.07 ± 0.18	2.023 ± 0.003	0.0106 ± 0.0005	0.43 ± 0.31	0.0026757
	Zn-Cl	0.36 ± 0.08	2.217 ± 0.021			
Zn-SA/CNCl-920	Zn-N	3.99 ± 0.12	2.004 ± 0.012	0.0097 ± 0.0003	-1.79 ± 0.26	0.0010394
	Zn-Cl	0.27 ± 0.05	2.215 ± 0.015			
Zn-SA/CNCl-1000	Zn-N	4.02 ± 0.14	2.045 ± 0.003	0.0098 ± 0.0005	4.22 ± 0.39	0.0022350
Zn-SA/CN-1000	Zn-N	4.01 ± 0.08	2.001 ± 0.001	0.0078 ± 0.0002	-2.80 ± 0.24	0.0002944
Zn-SA/CNCl-1000 (CO ₂ RR@OCV)	Zn-N	4.04 ± 0.17	2.030 ± 0.003	0.0094 ± 0.0006	3.69 ± 0.42	0.0021748
Zn-SA/CNCl-1000 (CO ₂ RR@-0.9V)	Zn-N	3.44 ± 0.35	2.008 ± 0.007	0.0134 ± 0.0014	2.24 ± 0.89	0.0052816
Zn-SA/CN-1000 (CO ₂ RR@OCV)	Zn-N	4.31 ± 0.11	2.024 ± 0.002	0.0100 ± 0.0003	2.43 ± 0.23	0.0005161
Zn-SA/CN-1000 (CO ₂ RR@-0.9V)	Zn-N	4.07 ± 0.09	2.021 ± 0.002	0.0093 ± 0.0003	2.70 ± 0.22	0.0005811

The asterisk represented the coordination numbers of Zn-Zn bonds in the Zn foil were fixed during the EXAFS fitting. As shown in **Figure R16**, each Zn atom in Zn foil has two kinds of adjacent Zn coordination atoms, with 6 Zn atoms in the same layer (red hexagon) and 3+3 Zn atoms in the adjacent upper and lower layers (green triangle). Therefore, the coordination numbers of Zn-Zn bonds in the Zn foil were

fixed as 6 during EXAFS fitting.

Figure R16. The optimized structure of Zn foil.

During the EXAFS fitting, we fitted the Zn-N and Zn-Cl bonds in the Zn-SA/CNCl samples. Firstly, we also performed XPS measurement to determine whether the Zn-Cl pathway should be considered during EXAFS fitting. The XPS spectra for the Cl 2*p* of Zn-SA/CNCl samples were shown in **Figs. 1e-1g**. The ionic Cl from Zn-Cl bond of Zn-SA/CNCl-850, Zn-SA/CNCl-920 and Zn-SA/CNCl-1000 were 56.1%, 43.8% and 6.9%, respectively. Since the noticeable peak intensity of ionic Cl from Zn-Cl bond exists in Zn-SA/CNCl-850 and Zn-SA/CNCl-920, the inclusion of Zn-Cl pathway during the EXAFS fitting of Zn-SA/CNCl-850 and Zn-SA/CNCl-920 is necessary and rational.

By contrast, compared with Zn-SA/CNCl-850 and Zn-SA/CNCl-920, the ionic Cl of Zn-SA/CNCl-1000 is only 6.9% with trace amounts and the signal has poor signal-to-noise ratio. Therefore, we conclude that the Zn-Cl bond exists in

Zn-SA/CNCl-850 and Zn-SA/CNCl-920 samples while it is difficult to determine whether the Zn-Cl bond exists in Zn-SA/CNCl-1000 sample.

We also attempt to introduce the Zn-Cl pathway during the EXAFS fitting of Zn-SA/CNCl-1000. As shown in **Table R1**, the corresponding fitting result exhibited that the coordination numbers of Zn-N bond and Zn-Cl bond were 4.18 and -0.16, respectively. The coordination number of Zn-Cl bond was negative (-0.16), which indicated that introducing the Zn-Cl pathway during the EXAFS fitting of Zn-SA/CNCl-1000 was inappropriate.

Similarly, we also introduced the Zn-Cl pathway during the EXAFS fitting of Zn-SA/CNCl-850 and Zn-SA/CNCl-920 by the same fitting method. The coordination numbers of Zn-Cl bond in Zn-SA/CNCl-850 and Zn-SA/CNCl-920 were 0.36 and 0.27, respectively, revealing that introducing the Zn-Cl pathway for fitting Zn-SA/CNCl-850 and Zn-SA/CNCl-920 was appropriate.

Therefore, the Zn-Cl pathway was not involved during the EXAFS fitting of Zn-SA/CNCl-1000, which was different from those of Zn-SA/CNCl-850 and Zn-SA/CNCl-920.

Table R1. The structural parameters for fitting Zn-SA/CNCl-1000 when introducing Zn-Cl pathway. ($S_0^2 = 0.78$)

Sample	Shell	N	R(Å)	$\sigma^2(\text{Å}^2)$	$\Delta E_0(\text{eV})$	R factor
Zn-SA/CNCl-1000	Zn-N	4.18 ± 0.17	2.047 ± 0.003	0.00985 ± 0.0008	4.96 ± 0.75	0.00098
	Zn-Cl	-0.16 ± 0.08	2.192 ± 0.041			
Zn-SA/CNCl-920	Zn-N	3.99 ± 0.12	2.004 ± 0.012	0.0097 ± 0.0003	-1.79 ± 0.26	0.0010394
	Zn-Cl	0.27 ± 0.05	2.215 ± 0.015			
Zn-SA/CNCl-850	Zn-N	4.07 ± 0.18	2.023 ± 0.003	0.0106 ± 0.0005	0.43 ± 0.31	0.0026757
	Zn-Cl	0.36 ± 0.08	2.217 ± 0.021			

Response to the Reviewers

Reviewer #2:

Comment 1: During the revision authors have addressed some of my prior comments. Nonetheless, I feel that two my most important points were not answered.

First, I am still not convinced that there are any Zn-Cl bonds in any of the investigated samples. In my previous review, I have asked authors to carry out EXAFS fitting without Zn-Cl bonds included, with one or two non-equivalent Zn-N paths only, to demonstrate that the inclusion of Cl indeed improves the fits noticeably. This has not been done. In the revised version of the manuscript authors also show XANES spectrum simulated for their structure model. However, it looks rather featureless, and similar to the spectrum of singly dispersed Zn species in general. Does really the presence of Cl in this structure affect the XANES spectrum in any noticeably way? There is also no explanation in the manuscript, how the simulations of XANES spectra were carried out. Which code was used? Was there any attempt to optimize the structure to improve the agreement with experimental data?

Reply: We sincerely appreciate the comments provided by Reviewer #2. We extend our sincere appreciation to the reviewer for the time, expertise, and constructive feedback, which have undoubtedly contributed to the improvement of our manuscript.

Question 1: EXAFS fitting with only one or two non-equivalent Zn-N paths.

We constructed three models for EXAFS fitting of Zn-based samples. (**model 1:** with only one Zn-N path; **model 2:** with both Zn-N and Zn-Cl paths; **model 3:** with two non-equivalent Zn-N paths.) Due to the stronger contribution of Zn-N path in the first coordination shell of Zn, obtaining the accurate coordination number of Zn-Cl bond is challenging by EXAFS fitting at Zn-K edge. Therefore, EXAFS measurement at Cl K-edge was also performed, confirming the C-Cl bond and Zn-Cl bond co-existed in Zn-SA/CNCl-850 and Zn-SA/CNCl-920. While the Cl element mainly existed as C-Cl bond in Zn-SA/CNCl-1000. The detailed analysis is shown as

follows:

The EXAFS fitting results of Zn-SA/CNCI-850 was shown in **Figure R1** and **Table R1**. During fitting, the disorder factors of different paths were not constricted to the same value. As shown in **Table R1**, the R factors of Zn-SA/CNCI-850 fitted by model 1, model 2 and model 3 are 0.001319, 0.001805 and 0.001919, respectively, indicating that the model 1 has the best goodness of fitting result for fitting Zn-SA/CNCI-850.

Figure R1. The EXAFS fitting results of Zn-SA/CNCI-850. **a-b**, The fitting result in k space and R space, respectively, with utilization of model 1 as fitting model. **c-d**, The fitting result in k space and R space, respectively, with utilization of model 2 as fitting model. **e-f**, The fitting result in k space and R space, respectively, with utilization of model 3 as fitting model. (**model 1**: with only one Zn-N path; **model 2**: with both Zn-N and Zn-Cl paths; **model 3**: with two non-equivalent Zn-N paths.)

Table R1. The structural parameters extracted from the Zn K-edge for fitting Zn-SA/CNCI-850 with different models. ($S_0^2 = 0.78$)

Sample	Shell	N	R(\AA)	$\sigma^2(\text{\AA}^2)$	$\Delta E_0(\text{eV})$	R factor
Zn-SA/CNCI-850 (model 1)	Zn-N	4.11 \pm 0.11	2.038 \pm 0.011	0.0094 \pm 0.0007	3.07 \pm 0.28	0.001319
Zn-SA/CNCI-850 (model 2)	Zn-N	4.08 \pm 0.16	2.027 \pm 0.003	0.0099 \pm 0.0011	0.98 \pm 0.33	0.001805
	Zn-Cl	0.39 \pm 0.10	2.217 \pm 0.021	0.0194 \pm 0.0050		
Zn-SA/CNCI-850 (model 3)	Zn-N1	4.09 \pm 0.16	2.044 \pm 0.015	0.0096 \pm 0.0005	4.42 \pm 0.37	0.001919
	Zn-N2	0.34 \pm 0.12	2.214 \pm 0.024	0.0155 \pm 0.0062		

The EXAFS fitting results of Zn-SA/CNCI-920 was shown in **Figure R2** and **Table R2**. During fitting, the disorder factors of different paths were not constricted to the same value. As shown in **Table R2**, the R factors of Zn-SA/CNCI-920 fitted by model 1, model 2 and model 3 are 0.000804, 0.001445 and 0.001520, respectively. Therefore, the model 1 is the best model for fitting Zn-SA/CNCI-920.

Figure R2. The EXAFS fitting results of Zn-SA/CNCI-920. **a-b**, The fitting result in k space and R space, respectively, with utilization of model 1 as fitting model. **c-d**, The fitting result in k space and R space, respectively, with utilization of model 2 as fitting model. **e-f**, The fitting result in k space and R space, respectively, with utilization of model 3 as fitting model. (**model 1**: with only one Zn-N path; **model 2**: with both Zn-N and Zn-Cl paths; **model 3**: with two non-equivalent Zn-N paths.)

Table R2. The structural parameters extracted from the Zn K-edge for fitting Zn-SA/CNCl-920 with different models. ($S_0^2 = 0.78$)

Sample	Shell	N	R(Å)	$\sigma^2(\text{\AA}^2)$	$\Delta E_0(\text{eV})$	R factor
Zn-SA/CNCl-920 (model 1)	Zn-N	4.06 ± 0.12	2.025 ± 0.008	0.0092 ± 0.0005	1.74 ± 0.24	0.000804
Zn-SA/CNCl-920 (model 2)	Zn-N	4.03 ± 0.12	2.011 ± 0.012	0.0099 ± 0.0011	-0.56 ± 0.30	0.001445
	Zn-Cl	0.29 ± 0.05	2.218 ± 0.015	0.0120 ± 0.0050		
Zn-SA/CNCl-920 (model 3)	Zn-N1	4.04 ± 0.11	2.026 ± 0.011	0.0092 ± 0.0015	2.67 ± 0.50	0.001520
	Zn-N2	0.30 ± 0.13	2.217 ± 0.022	0.0094 ± 0.0015		

The EXAFS fitting results of Zn-SA/CNCl-1000 was shown in **Figure R3** and **Table R3**. During fitting, the disorder factors of different paths were not constricted to the same value. As shown in **Table R3**, the R factors of Zn-SA/CNCl-1000 fitted by model 1 and model 2 are 0.002235 and 0.002278, respectively, which are similar to each other. After adding the Zn-Cl path for fitting Zn-SA/CNCl-1000 by model 2, the coordination number of Zn-Cl bond in model 2 is 0.00 ± 0.17 , indicating the Zn-Cl bond is negligible in the Zn-SA/CNCl-1000. Therefore, the model 1 with only Zn-N path is the rational model for fitting Zn-SA/CNCl-1000.

Figure R3. The EXAFS fitting results of Zn-SA/CNCI-1000. **a-b**, The fitting result in k space and R space, respectively, with utilization of model 1 as fitting model. **c-d**, The fitting result in k space and R space, respectively, with utilization of model 2 as fitting model. (**model 1**: with only one Zn-N path; **model 2**: with both Zn-N and Zn-Cl paths.)

Table R3. The structural parameters extracted from the Zn K-edge for fitting Zn-SA/CNCl-1000 with different models. ($S_0^2 = 0.78$)

Sample	Shell	N	R(Å)	$\sigma^2(\text{Å}^2)$	$\Delta E_0(\text{eV})$	R factor
Zn-SA/CNCl-1000 (model 1)	Zn-N	4.02 ± 0.14	2.045 ± 0.003	0.0098 ± 0.0005	4.22 ± 0.39	0.002235
Zn-SA/CNCl-1000 (model 2)	Zn-N	4.02 ± 0.14	2.045 ± 0.005	0.0098 ± 0.0021	4.32 ± 0.88	0.002278
	Zn-Cl	0.00 ± 0.17	2.219 ± 0.011	0.0113 ± 0.0056		

In the Zn-SA/CNCl catalysts, the N element is abundant while the Cl element is in trace amounts. Thus, the Zn elements in both Zn-SA/CNCl-850, Zn-SA/CNCl-920 and Zn-SA/CNCl-1000 mainly existed as Zn-N bond rather than Zn-Cl bond. Therefore, during the EXAFS fitting by model 2 with both Zn-N and Zn-Cl paths, the intensity of Zn-N bond is much stronger than that of the Zn-Cl bond. Thus, on the background of strong contribution of Zn-N bond, obtaining the accurate coordination number of Zn-Cl bond is challenging because the contribution of Zn-N bond can easily cover the contribution of Zn-Cl bond. Considering the strong contribution of Zn-N bond, it is difficult to judge whether the Zn-Cl bond exists in the Zn-SA/CNCl catalysts by only measuring the EXAFS results at Zn K-edge.

We added the **Figures R1-R3** as **Supplementary Figs. 16-18** and **Tables R1-R3** as **Supplementary Tables 2-4** in the revised Supplementary Information and the revised Manuscript (Line 181-185).

In order to study the existence form of Cl element in the Zn-SA/CNCl catalysts, we carry out the EXAFS measurement at Cl K-edge, which can effectively eliminate the effect of the Zn-N bond, as shown in **Figure R4**. The EXAFS measurement at Cl K-edge confirmed the co-existence of C-Cl bond and Zn-Cl bond in Zn-SA/CNCl-850 and Zn-SA/CNCl-920. While the Cl element mainly existed as C-Cl bond in Zn-SA/CNCl-1000.

The XANES spectra of Zn-SA/CNCl catalysts at Cl K-edge are shown in **Figure R4a** and the corresponding FT-EXAFS results in R space are exhibited in **Figure R4b**. As shown in **Figure R4b**, the two prominent peaks at around 1.2 Å and 2.1 Å are assigned to the Cl-C path and Cl-Zn path, respectively. The two prominent Cl-C and Cl-Zn paths co-exist in Zn-SA/CNCl-850 and Zn-SA/CNCl-920. While the intensity of Cl-Zn path in Zn-SA/CNCl-1000 has a noticeable decline, compared with those of Zn-SA/CNCl-850 and Zn-SA/CNCl-920. By contrast, the intensity of Cl-C path in Zn-SA/CNCl-1000 is similar to those of Zn-SA/CNCl-850 and Zn-SA/CNCl-920. These results reveal the gradual transformation from the Zn-N₄Cl₁ site into Zn-N₄ site as the pyrolysis temperatures of Zn-SA/CNCl samples increasing from 850°C to 1000°C.

As the pyrolysis temperatures increasing from 850°C to 1000°C, the intensity of Cl-Zn path gradually decreases and the peak location of Cl-Zn path gradually moves toward higher R space, revealing that the Zn-Cl bonds are gradually elongated and finally are broken, which is attributed to the gradual transformation from the Zn-N₄Cl₁ site into Zn-N₄ site. By contrast, the intensity of Cl-C path has no obvious changes as the pyrolysis temperatures increasing from 850°C to 1000°C, indicating the C-Cl bond is much stronger than Zn-Cl bond under elevated temperatures.

Figure R4. **a**, The XANES spectra of Zn-SA/CNCI catalysts at Cl K-edge. **b**, The corresponding FT-EXAFS results in R space.

We added the **Figure R4** as **Supplementary Fig. 21** in the revised Supplementary Information and **Fig. 2c** in the revised Manuscript (Line 188-207).

The WT analysis of Zn-SA/CNCI-850, Zn-SA/CNCI-920 and Zn-SA/CNCI-1000 at Cl-K edge are compared in **Figure R5**. The prominent peak at around 6 \AA^{-1} in k space and 1.2 \AA in R space is assigned to the Cl-C path. While the prominent peak at around 7 \AA^{-1} in k space and 2.0 \AA in R space is assigned to the Cl-Zn path. Obviously, the Cl-C path and Cl-Zn path co-exist in the Zn-SA/CNCI-850 and Zn-SA/CNCI-920. As the pyrolysis temperatures gradually increasing from 850°C to 1000°C , the peak intensity of Cl-Zn path has a noticeable decay, revealing the breakage of Zn-Cl bond during the gradual transformation from $\text{Zn-N}_4\text{Cl}_1$ sites into Zn-N_4 sites. As shown in **Figure R5c**, the Zn-SA/CNCI-1000 has a prominent peak of Cl-C path while the intensity of Cl-Zn path is rather weak.

Figure R5. **a-c**, The WT analysis of Zn-SA/CNCI-850, Zn-SA/CNCI-920 and Zn-SA/CNCI-1000 at Cl-K edge, respectively.

We added the **Figure R5** as **Figs. 2d-2f** in the revised Manuscript (Line 208-221).

On the other hand, the XPS spectra for the Cl 2*p* of Zn-SA/CNCl samples are compared in **Figure R6**. The ionic Cl from Zn-Cl bond of Zn-SA/CNCl-850, Zn-SA/CNCl-920 and Zn-SA/CNCl-1000 are 56.1%, 43.8% and 6.9%, respectively. The proportion of ionic Cl from Zn-Cl bond gradually decreases as the pyrolysis temperatures increasing from 850°C to 1000°C, which is consistent with the EXAFS results at Cl K-edge.

Figure R6. a-c, The XPS spectra for Cl 2*p* of Zn-SA/CNCl-850, Zn-SA/CNCl-920 and Zn-SA/CNCl-1000, respectively.

Therefore, by combination of EXAFS measurement at Cl K-edge and XPS measurement for Cl 2*p*, we conclude that the C-Cl bond and Zn-Cl bond co-exist in Zn-SA/CNCl-850 and Zn-SA/CNCl-920 samples. While the Cl element mainly exists as C-Cl bond in the Zn-SA/CNCl-1000.

Question 2: Simulation of XANES spectrum.

We sincerely appreciate the comments provided by Reviewer #2 regarding the effect of Cl element for simulating XANES spectrum. As shown in **Supplementary Scheme 2**, the C-Cl bonds, Zn-N₄Cl₁ sites and Zn-N₄ sites co-exist in Zn-SA/CNCl-850 and Zn-SA/CNCl-920 samples. While the C-Cl bonds and Zn-N₄ sites co-exist in Zn-SA/CNCl-1000.

In order to study the effect of Cl element for simulating XANES spectra, we simulate the theoretical XANES spectra of Zn-N₄Cl/CNCl-1 and Zn-N₄/CNCl-1 models, as shown in **Figure R7**.

Supplementary Scheme 2. The evolution of catalytic site from Zn-SA/CNCl-850 to Zn-SA/CNCl-1000 catalysts.

Figure R7. a, The theoretical XANES spectra of Zn-N₄Cl/CNCI-1 and Zn-N₄/CNCI-1 models. **b-c**, The corresponding optimized Zn-N₄Cl/CNCI-1 and Zn-N₄/CNCI-1 models.

As shown in **Figure R7a**, the theoretical XANES spectrum of Zn-N₄Cl/CNCI-1 model with Zn-N₄Cl₁ site is similar to that of Zn-N₄/CNCI-1 model with Zn-N₄ site. Therefore, the gradual transformation from the Zn-N₄Cl₁ site into the Zn-N₄ site can not induce remarkable change of XANES spectrum at Zn-K edge.

Besides, the Cl element is in trace amount in Zn-SA/CNCI catalysts and the Zn element in Zn-SA/CNCI catalysts mainly exists as Zn-N bonds. Therefore, the effect of trace amount Cl element on the XANES spectra of Zn element in Zn-SA/CNCI catalysts is not remarkable. Therefore, the experimental XANES spectra of Zn-SA/CNCI-850, Zn-SA/CNCI-920 and Zn-SA/CNCI-1000 are similar to each other.

We sincerely appreciate the comments provided by Reviewer #2 regarding the method for simulating the XANES spectra. We utilized FDMNES with the 29th reversion to simulate the XANES spectra. We calculated the theoretical XANES spectra by using all the software’s default options within the finite difference method mode. Upon entering the lattice parameters and atomic structure into the software, we incrementally calculated the spectrum with varying “Cluster Radius” from 3 to 7 angstrom and convoluted the spectra.

We added the **Figure R7** as **Supplementary Fig. 23** in the revised Supplementary Information and the revised Manuscript (Line 229-234). The above description of method for simulating the XANES spectra was also added in **Supplementary Fig. 23**.

As shown in **Figure R8a**, the reference (*Angew. Chem. Int. Ed.* **60**, 27324-27329 (2021).) reported the FeN₄Cl₁/NC catalyst with Fe-N₄Cl₁ catalytic sites for catalyzing oxygen reduction reaction (ORR). The XANES spectrum of FeN₄Cl₁/NC catalyst was similar to that of FeN₄/NC catalyst without Fe-Cl bond. As shown in **Figure R8b**, the reference (*ACS Nano* **16**, 15165-15174 (2022).) also reported the FeN₄Cl SAC with Fe-N₄Cl catalytic sites for ORR. The XANES spectra of FeN₄Cl SAC and FeN_x SAC also exhibited similar characteristic to each other. These results revealed that the coordination of metal-Cl bond had no obvious effect on the XANES spectrum of metal element at K-edge.

Figure R8. a, The comparison of XANES spectra of FeN₄Cl₁/NC and FeN₄/NC. (*Angew. Chem. Int. Ed.* **60**, 27324-27329 (2021).) **b**, The comparison of XANES spectra of FeN₄Cl SAC and FeN_x SAC. (*ACS Nano* **16**, 15165-15174 (2022).)

Due to the trace amount of Cl element in Zn-SA/CNCl catalysts and the weak effect of Cl element on the XANES spectra of Zn-SA/CNCl catalysts, the EXAFS measurement at Cl K-edge was also performed for studying the existence form of Cl element. As shown in **Figure R4** and **Figure R5**, the EXAFS measurement at Cl

K-edge confirmed the co-existence of C-Cl bond and Zn-Cl bond in Zn-SA/CNCl-850 and Zn-SA/CNCl-920. While the Cl element mainly existed as C-Cl bond in Zn-SA/CNCl-1000.

Therefore, we added the EXAFS measurement at Cl K-edge in **Fig. 2** in the revised Manuscript (Line 160) as follows:

Fig. 2. The characterization of Zn-SA/CNCl catalysts by XANES and EXAFS measurements at Zn K-edge and Cl K-edge. **a-b**, The XANES spectra and corresponding FT-EXAFS results in R space of Zn-SA/CNCl-850, Zn-SA/CNCl-920, Zn-SA/CNCl-1000, Zn foil, ZnO and ZnCl₂ as reference samples at Zn K-edge. **c**, The FT-EXAFS results in R space of Zn-SA/CNCl-850, Zn-SA/CNCl-920, and Zn-SA/CNCl-1000 at Cl K-edge. **d-f**, The WT analysis of Zn-SA/CNCl-850, Zn-SA/CNCl-920 and Zn-SA/CNCl-1000 at Cl K-edge, respectively.

***Comment 2:** In the revised version of the manuscript, authors also provide XPS data to show the presence of Cl in their catalysts. This is an important observation. However, XPS results cannot be directly compared with XAS, since the former is surface sensitive technique while the latter-bulk sensitive. More importantly, XPS does not show that there indeed are Zn-Cl bonds in the material-just that there is Cl somewhere in this system.*

Reply: We sincerely appreciate the comments provided by Reviewer #2.

The XPS measurement reflects the chemical structure of the surface of catalyst while the EXAFS measurement reveals the overall structural information of catalyst, including internal structural information.

In order to study the existence form of Cl element in the Zn-SA/CNCl catalysts, we carry out the EXAFS measurement at Cl K-edge, which can effectively eliminate the effect of the Zn-N bond, as shown in **Figure R4**.

The XANES spectra of Zn-SA/CNCl catalysts at Cl K-edge are shown in **Figure R4a** and the corresponding FT-EXAFS results in R space are exhibited in **Figure R4b**. As shown in **Figure R4b**, the two prominent peaks at around 1.2 Å and 2.1 Å are assigned to the Cl-C path and Cl-Zn path, respectively. The two prominent Cl-C and Cl-Zn paths co-exist in Zn-SA/CNCl-850 and Zn-SA/CNCl-920. While the intensity of Cl-Zn path in Zn-SA/CNCl-1000 has a noticeable decline, compared with those of Zn-SA/CNCl-850 and Zn-SA/CNCl-920. By contrast, the intensity of Cl-C path in Zn-SA/CNCl-1000 is similar to those of Zn-SA/CNCl-850 and Zn-SA/CNCl-920. These results reveal the gradual transformation from the Zn-N₄Cl₁ site into Zn-N₄ site as the pyrolysis temperatures of Zn-SA/CNCl samples increasing from 850°C to 1000°C.

As the pyrolysis temperatures increasing from 850°C to 1000°C, the intensity of Cl-Zn path gradually decreases and the peak location of Cl-Zn path gradually moves toward higher R space, revealing that the Zn-Cl bonds are gradually elongated and finally are broken, which is attributed to the gradual transformation from the Zn-N₄Cl₁ site into Zn-N₄ site. By contrast, the intensity of Cl-C path has no obvious changes as

the pyrolysis temperatures increasing from 850°C to 1000°C, indicating the C-Cl bond is much stronger than Zn-Cl bond under elevated temperatures.

Figure R4. **a**, The XANES spectra of Zn-SA/CNCl catalysts at Cl K-edge. **b**, The corresponding FT-EXAFS results in R space.

We added the **Figure R4** as **Supplementary Fig. 21** in the revised Supplementary Information and **Fig. 2c** in the revised Manuscript (Line 188-207).

The WT analysis of Zn-SA/CNCl-850, Zn-SA/CNCl-920 and Zn-SA/CNCl-1000 at Cl-K edge are compared in **Figure R5**. The prominent peak at around 6 \AA^{-1} in k space and 1.2 \AA in R space is assigned to the Cl-C path. While the prominent peak at around 7 \AA^{-1} in k space and 2.0 \AA in R space is assigned to the Cl-Zn path. Obviously, the Cl-C path and Cl-Zn path co-exist in the Zn-SA/CNCl-850 and Zn-SA/CNCl-920. As the pyrolysis temperatures gradually increasing from 850°C to 1000°C, the peak intensity of Cl-Zn path has a noticeable decay, revealing the breakage of Zn-Cl bond during the gradual transformation from $\text{Zn-N}_4\text{Cl}_1$ sites into Zn-N_4 sites. As shown in **Figure R5c**, the Zn-SA/CNCl-1000 has a prominent peak of Cl-C path while the intensity of Cl-Zn path is rather weak.

Figure R5. a-c, The WT analysis of Zn-SA/CNCI-850, Zn-SA/CNCI-920 and Zn-SA/CNCI-1000 at Cl-K edge, respectively.

We added the **Figure R5** as **Figs. 2d-2f** in the revised Manuscript (Line 208-221).

On the other hand, the XPS spectra for the Cl 2*p* of Zn-SA/CNCI samples are compared in **Figure R6**. The ionic Cl from Zn-Cl bond of Zn-SA/CNCI-850, Zn-SA/CNCI-920 and Zn-SA/CNCI-1000 are 56.1%, 43.8% and 6.9%, respectively. The proportion of ionic Cl from Zn-Cl bond gradually decreases as the pyrolysis temperatures increasing from 850°C to 1000°C, which is consistent with the EXAFS results at Cl K-edge.

Figure R6. a-c, The XPS spectra for Cl 2*p* of Zn-SA/CNCI-850, Zn-SA/CNCI-920 and Zn-SA/CNCI-1000, respectively.

Therefore, by combination of EXAFS measurement at Cl K-edge and XPS measurement for Cl 2*p*, we conclude that the C-Cl bond and Zn-Cl bond co-exist in Zn-SA/CNCI-850 and Zn-SA/CNCI-920 samples. While the Cl element mainly exists as C-Cl bond in the Zn-SA/CNCI-1000.

Comment 3: *To demonstrate the presence or the absence of Zn-Cl bonds in their EXAFS fits, authors argue in their reply that if EXAFS fitting gives a negative coordination number for Zn-Cl bond, then this surely means that there are no Zn-Cl bonds in the material. However, the negative coordination numbers can also simply mean the chosen structure model is inadequate, e.g., that the imposed constraint that the disorder factors for Zn-Cl and Zn-N bonds is the same cannot be applied. This can also be simply a result of the instability of non-linear fitting, and can disappear, if different initial guesses for structure parameters are chosen.*

Reply: We sincerely appreciate the comments provided by Reviewer #2.

The EXAFS fitting results of Zn-SA/CNCl-1000 was shown in **Figure R3** and **Table R3**. During fitting by model 2 with both Zn-N and Zn-Cl paths, the disorder factors of Zn-N and Zn-Cl paths were not constricted to the same value. We also gave a positive initial guess of the coordination number of Zn-Cl bond in Zn-SA/CNCl-1000 (initial N value = 1). However, after fitting, the coordination number of Zn-Cl bond in Zn-SA/CNCl-1000 was 0.00 ± 0.17 . Besides, the disorder factors of Zn-N and Zn-Cl paths were 0.0098 ± 0.0021 and 0.0113 ± 0.0056 , respectively, which were independent to each other.

After adding the Zn-Cl path for fitting Zn-SA/CNCl-1000 by model 2, the coordination number of Zn-Cl bond in model 2 is 0.00 ± 0.17 , indicating the Zn-Cl bond is negligible in the Zn-SA/CNCl-1000. Therefore, the model 1 with only Zn-N path is the rational model for fitting Zn-SA/CNCl-1000.

Figure R3. The EXAFS fitting results of Zn-SA/CNCl-1000. **a-b**, The fitting result in k space and R space, respectively, with utilization of model 1 as fitting model. **c-d**, The fitting result in k space and R space, respectively, with utilization of model 2 as fitting model. (**model 1**: with only one Zn-N path; **model 2**: with both Zn-N and Zn-Cl paths.)

Table R3. The structural parameters extracted from the Zn K-edge for fitting Zn-SA/CNCl-1000 with different models. ($S_0^2 = 0.78$)

Sample	Shell	N	R(Å)	$\sigma^2(\text{Å}^2)$	$\Delta E_0(\text{eV})$	R factor
Zn-SA/CNCl-1000 (model 1)	Zn-N	4.02 ± 0.14	2.045 ± 0.003	0.0098 ± 0.0005	4.22 ± 0.39	0.002235
Zn-SA/CNCl-1000 (model 2)	Zn-N	4.02 ± 0.14	2.045 ± 0.005	0.0098 ± 0.0021	4.32 ± 0.88	0.002278
	Zn-Cl	0.00 ± 0.17	2.219 ± 0.011	0.0113 ± 0.0056		

Comment 4: *Even more importantly, authors still have not provided a satisfactory answer to my last question from the previous review: if their XAS data suggest that the sample treated at 1000 deg C does not have any Zn-Cl bonds, and XPS also shows that the amount of Cl in this sample is rather low, how can they conclude that chlorine is actually doing something for the catalytic mechanism? How can they claim that the C-Cl bonds are indeed located in the vicinity of Zn species?*

To conclude, major revision is still necessary, since the author's conclusions do not seem to be supported by experimental data.

Reply: We sincerely appreciate the comments provided by Reviewer #2.

Question 1: Proving that the C-Cl bonds are indeed located in the vicinity of Zn species.

In order to prove the C-Cl bonds are indeed located in the vicinity of Zn species in Zn-SA/CNCl-1000, the AC-STEM measurement with EDX spectroscopy elemental mapping was carried out. As shown in **Figure R9a**, the isolated bright dots in Zn-SA/CNCl-1000 catalyst represented the Zn isolated single-atom sites, which were marked by red circles. The corresponding EDX spectroscopy elemental mapping results of C and Cl elements were exhibited in **Figure R9b** and **Figure R9c**, respectively. The overlapping EDX spectroscopy elemental mapping results of C and Cl elements were exhibited in **Figure R9d**.

The C and Cl elements were homogeneously dispersed on the substrate of Zn-SA/CNCl-1000. The white circles in **Figure R9b**, **Figure R9c** and **Figure R9d** had the same locations as the red circles in **Figure R9a**, reflecting the locations of Zn isolated single-atom sites. As shown in **Figure R9d**, the C and Cl signal simultaneously appeared within each white circle, revealing that the C and Cl elements were indeed located in the vicinity of Zn species.

On the other hand, we also performed the AC-STEM measurement with electron energy loss spectroscopy (EELS). As shown in **Figure R10**, the EELS spectrum around the Zn isolated single-atom site confirmed the presence of C and Cl signals in the vicinity of Zn isolated single-atom site, also revealing the C and Cl elements were

indeed located in the vicinity of Zn species.

Figure R9. a, The AC-STEM image of Zn-SA/CNCl-1000. The isolated bright dots in Zn-SA/CNCl-1000 catalyst represent the Zn isolated single-atom sites, which are marked by red circles. b-c, The corresponding EDX spectroscopy elemental mapping results of C and Cl elements. d, The overlapping EDX spectroscopy elemental mapping results of C and Cl elements. The white circles in Figure R9b, Figure R9c and Figure R9d have the same locations as the red circles in Figure R9a, reflecting the locations of Zn isolated single-atom sites.

Figure R10. **a**, The AC-STEM image of Zn-SA/CNCl-1000. **b**, The EELS spectrum around the Zn isolated single-atom site, which is marked by the white circle in Figure R10a.

We added the **Figures R9-R10** as **Supplementary Figs. 40-41** in the revised Supplementary Information and the revised Manuscript (Line 368-371).

As shown in **Figure R11**, the Cl element mainly existed as C-Cl bond in the Zn-SA/CNCl-1000, confirmed by combination of XPS measurement for Cl 2p (**Figure R11a**) and the WT analysis of Zn-SA/CNCl-1000 at Cl-K edge (**Figure R11b**). Therefore, we concluded that the C-Cl bonds are indeed located in the vicinity of Zn species in Zn-SA/CNCl-1000.

Figure R11. a, The XPS spectrum of Zn-SA/CNCl-1000 for Cl 2p. b, The WT analysis of Zn-SA/CNCl-1000 at Cl-K edge.

Besides, in order to demonstrate the C-Cl bond was favourable to be located in the vicinity of Zn-N₄ site, we constructed ten possible Zn-N₄/CNCl models with C-Cl bond at different positions, as shown in **Figure R12**. We compared the relative energies of different Zn-N₄/CNCl models. The Zn-N₄/CNCl-1 was the most stable model with the lowest energy defined as 0 eV while the Zn-N₄/CNCl-2 was the second stable model with energy of 1.51 eV. As the C-Cl bond moved away from the Zn-N₄ sites, the relative energy increased, demonstrating the more stable models with C-Cl bond adjacent to Zn-N₄ sites.

Figure R12. The optimized structures of ten possible Zn-N₄/CNCl models with C-Cl bond at different positions and the comparison of relative energies.

Question 2: The catalytic role of Cl element on Zn catalytic sites for CO₂RR.

As shown in **Table R4**, the atomic ratios of C, N, Zn and Cl elements of Zn-SA/CNCl catalysts are measured by XPS measurement. Although the atomic ratio of Cl element in Zn-SA/CNCl catalysts was much lower than those of C and N elements, the atomic ratio of Cl element in Zn-SA/CNCl catalysts was comparable to that of Zn element as catalytic sites. Therefore, the catalytic role of Cl element on Zn catalytic sites for CO₂RR can not be ignored. As revealed by EXAFS measurement at Cl K-edge in **Figure R4** and **Figure R5**, from Zn-SA/CNCl-850 to Zn-SA/CNCl-1000, the intensity of Cl-Zn path gradually decreased. The C-Cl bond and Zn-Cl bond co-existed in Zn-SA/CNCl-850 and Zn-SA/CNCl-920. While the Cl element mainly existed as C-Cl bond in Zn-SA/CNCl-1000. Thus, the evolution of existence form of Cl element in Zn-SA/CNCl catalysts has an important effect on Zn catalytic sites, especially, the gradual disappearance of Zn-Cl bond.

Table R4. The atomic ratios of C, N, Zn and Cl elements of Zn-SA/CNCl catalysts by XPS measurement.

Catalyst	Zn-SA/CNCl-850	Zn-SA/CNCl-920	Zn-SA/CNCl-1000
C (at%)	85.62	89.94	92.74
N (at%)	12.57	8.87	6.66
Zn (at%)	0.98	0.67	0.35
Cl (at%)	0.83	0.52	0.25

We added the **Table R4** as **Supplementary Table 1** in the revised Supplementary Information and the revised Manuscript (Line 143-146).

Besides, we revealed that the C-Cl bonds were located in the vicinity of Zn species in Zn-SA/CNCl-1000, confirmed by AC-STEM measurement with EDX spectroscopy elemental mapping, AC-STEM measurement with EELS and comparison of the relative energies between different Zn-N₄/CNCl models by DFT

calculation. Therefore, the C-Cl bonds were located in the vicinity of the Zn-N₄ sites in Zn-SA/CNCl-1000 while the Zn-N₄ sites in Zn-SA/CN-1000 were anchored on N-doped graphene without any C-Cl bonds. Although both the Zn element in Zn-SA/CNCl-1000 and Zn-SA/CN-1000 existed as Zn-N₄ sites, their chemical environments were quite different to each other.

In order to study the catalytic role of Cl element on Zn-N₄ catalytic sites in Zn-SA/CNCl-1000 for CO₂RR, in-situ XANES and EXAFS measurements were carried out at Zn K-edge. As shown in **Figure R13a-b**, in-situ XANES and EXAFS measurements were carried out for CO₂RR catalyzed by Zn-SA/CNCl-1000 catalyst at OCV, -0.7 V, -0.8 V, -0.9 V and -1.0 V *vs.* RHE. As shown in **Figure R13a**, compared with the XANES curve at OCV, the intensities of XANES white-line (WL) at -0.7 V and -0.8 V *vs.* RHE had an obvious decline. The intensities of XANES white-line (WL) at -0.9 V and -1.0 V *vs.* RHE further decreased compared with those at -0.7 V and -0.8 V *vs.* RHE. The corresponding FT-EXAFS spectra in R space were shown in **Figure R13b**. As the applied potentials gradually decreased from OCV to -1.0 V *vs.* RHE, the intensities of main peak from Zn-N bond at around 1.5 Å gradually decreased correspondingly, revealing the gradual transformation from Zn-N₄ site into low-coordinated Zn-N₃ site.

As shown in **Figure R13c-d**, in-situ XANES and EXAFS measurements were carried out for CO₂RR catalyzed by Zn-SA/CN-1000 catalyst without Cl-doping at OCV and -0.9 V *vs.* RHE. By contrast, for Zn-SA/CN-1000 without Cl-doping, there was no obvious changes in both XANES spectra and FT-EXAFS spectra in R space from OCV to -0.9 V *vs.* RHE, which was quite different from that of Zn-SA/CNCl-1000 catalyst.

Figure R13. The in-situ XANES and EXAFS measurements of Zn-SA/CNCI-1000 and Zn-SA/CN-1000 at Zn-K edge. **a**, The XANES spectra for CO₂RR catalyzed by Zn-SA/CNCI-1000 at OCV, -0.7 V, -0.8 V, -0.9 V and -1.0 V vs. RHE. **b**, The corresponding FT-EXAFS spectra in R space catalyzed by Zn-SA/CNCI-1000 at OCV, -0.7 V, -0.8 V, -0.9 V and -1.0 V vs. RHE. **c**, The XANES spectra for CO₂RR catalyzed by Zn-SA/CN-1000 at OCV and -0.9 V vs. RHE. **d**, The corresponding FT-EXAFS spectra in R space catalyzed by Zn-SA/CN-1000 at OCV and -0.9 V vs. RHE.

Figure R14. The corresponding WT analysis of in-situ XANES and EXAFS measurements for CO₂RR catalyzed by Zn-SA/CNCl-1000 and Zn-SA/CN-1000 at Zn-K edge. **a-b**, The WT analysis of Zn-SA/CNCl-1000 at OCV and -0.9 V vs. RHE during in-situ CO₂RR, respectively. **c-d**, The WT analysis of Zn-SA/CN-1000 at OCV and -0.9 V vs. RHE during in-situ CO₂RR, respectively. The potentials are provided without iR correction.

The corresponding WT analysis of in-situ XANES and EXAFS measurements also confirmed the above results. Compared with the WT contour plot of Zn-SA/CNCl-1000 at OCV in **Figure R14a**, the intensity of main peak (4.2 \AA^{-1} in k space and 1.5 \AA in R space) of Zn-SA/CNCl-1000 at -0.9 V vs. RHE remarkably decreased, as shown in **Figure R14b**, revealing the evolution from Zn-N₄ sites at OCV into Zn-N₃ sites at -0.9 V vs. RHE. By contrast, there was no obvious changes in the WT contour plots of Zn-SA/CN-1000 at OCV in **Figure R14c** and -0.9 V vs. RHE in **Figure R14d**. These results revealed that the planar chlorination engineering

effectively induced the self-reconstruction of Zn-N₄ sites into low-coordinated Zn-N₃ sites during CO₂RR.

Fig. 3b, The comparison of J_{CO} during CO₂RR. **Fig. 3c**, The comparison of TOF values.

As shown in **Fig. 3b** and **Fig. 3c**, compared with the Zn-SA/CN-1000 without Cl-doping, both the J_{CO} and TOF values catalyzed by Zn-SA/CNCl-1000 with adjacent C-Cl bonds remarkably increased, as the applied potentials gradually decreased from -0.57 V to -0.93 V vs. RHE. Especially, at -0.93 V vs. RHE, the maximum J_{CO} catalyzed by Zn-SA/CNCl-1000 was 271.7 mA/cm², which was 28.9 times that of Zn-SA/CN-1000 without Cl element. Besides, at -0.93 V vs. RHE, the TOF value catalyzed by Zn-SA/CNCl-1000 was 29325 h⁻¹, which was 82.8 times that of Zn-SA/CN-1000 (354.0 h⁻¹).

These results revealed that the Zn-N₄ catalytic sites with adjacent C-Cl bonds in Zn-SA/CNCl-1000 had much higher activity for CO₂RR, compared with the traditional Zn-N₄ sites without adjacent C-Cl bonds in Zn-SA/CN-1000. Combining the in-situ XANES and EXAFS measurements at Zn-K edge, we associated the much higher activity of Zn-SA/CNCl-1000 for CO₂RR with the in-situ structural evolution from Zn-N₄ sites into low-coordinated Zn-N₃ sites.

The different chemical environments at atomic-level between Zn-N₄ catalytic

sites with adjacent C-Cl bonds in Zn-SA/CNCl-1000 and the Zn-N₄ sites without adjacent C-Cl bonds in Zn-SA/CN-1000 resulted in different catalytic performance and structural evolution during CO₂RR. In order to understand the important role of adjacent C-Cl bonds on the much higher catalytic activity and structural evolution during CO₂RR catalyzed by Zn-SA/CNCl-1000, DFT calculation was performed.

As shown in **Figure R15**, we constructed three Zn-N₄/CNCl models with adjacent C-Cl bond and the pristine Zn-N₄/CN model without C-Cl bond.

Figure R15. a-d, The optimized Zn-N₄/CNCl-1, Zn-N₄/CNCl-2, Zn-N₄/CNCl-3 and Zn-N₄/CN models, respectively.

The CO₂RR into CO was the combination of reduction and protonation. Due to the large electronegativity of Cl and N element, the proton will enrich around the coordinated N atom with adjacent C-Cl bond. Therefore, the Gibbs energy changes (ΔG) for the protonation of coordinated N atom on Zn-N₄/CNCl-1, Zn-N₄/CNCl-2, Zn-N₄/CNCl-3 and Zn-N₄/CN models were compared in **Figure R16**. We denoted the protonation of coordinated N atom on Zn-based models as Zn-N₄/CNCl-1(H), Zn-N₄/CNCl-2(H) and Zn-N₄/CNCl-3(H), respectively. The ΔG for the protonation of

coordinated N atom on Zn-N₄/CNCl-1, Zn-N₄/CNCl-2, Zn-N₄/CNCl-3 and Zn-N₄/CN models were 0.79 eV, -0.66 eV, 1.06 eV and 1.20 eV, respectively, indicating the adjacent C-Cl bond induced the easier protonation of coordinated N atom than the pristine Zn-N₄ site.

Figure R16. The Gibbs energy changes (ΔG) for the protonation of coordinated N atom on Zn-N₄/CNCl-1, Zn-N₄/CNCl-2, Zn-N₄/CNCl-3 and Zn-N₄/CN models and the corresponding optimized structures.

As shown in **Figure R17**, we calculated the energy changes for CO₂RR on Zn-N₄/CNCl and Zn-N₄/CN models without the protonation of coordinated N atom, revealing the direct CO₂RR pathways on Zn-N₄/CNCl models were unable to induce the self-reconstruction of Zn-N₄ site into the low-coordinated Zn-N₃ catalytic sites.

Figure R17. a, The energy changes for CO₂RR on Zn-N₄/CNCI-1, Zn-N₄/CNCI-2, Zn-N₄/CNCI-3 and Zn-N₄/CN models without the protonation of coordinated N atom. b, The comparison of energy changes for RDS. c, The optimized structures of *COOH and *CO on Zn-N₄/CNCI-1, Zn-N₄/CNCI-2 and Zn-N₄/CNCI-3 models. The direct CO₂RR pathways on Zn-N₄/CNCI-1, Zn-N₄/CNCI-2, Zn-N₄/CNCI-3 models were unable to induce the self-reconstruction of Zn-N₄ site into the low-coordinated Zn-N₃ catalytic sites.

Besides, as exhibited in **Figure R18**, the energy changes of protonation of coordinated N atom on Zn-N₄/CNCl-1 and Zn-N₄/CNCl-2 models were lower than their energy barriers of rate-determining steps (RDS) for direct CO₂RR, indicating the protonation of coordinated N atom was easier to occur than direct CO₂RR on Zn-N₄/CNCl-1 and Zn-N₄/CNCl-2 models. Therefore, we considered the CO₂RR pathways on Zn-N₄/CNCl(H) models after the protonation of coordinated N atom.

Figure R18. **a**, The comparison between energy changes of the protonation of coordinated N atom on Zn-N₄/CNCl and Zn-N₄/CN models. **b**, The comparison between the energy barriers of RDS for direct CO₂RR.

The energy barriers of RDS for direct CO₂RR on Zn-N₄/CNCl-1 and Zn-N₄/CNCl-2 models were 1.38 eV and 0.82 eV, respectively, much higher than their protonation of coordinated N atom (0.79 eV and -0.66 eV, respectively), indicating the protonation of coordinated N atom was easier to occur than direct CO₂RR on Zn-N₄/CNCl-1 and Zn-N₄/CNCl-2 models. For Zn-N₄/CNCl-3 model, the energy barrier of RDS for direct CO₂RR was 0.70 eV, lower than the protonation of coordinated N atom on Zn-N₄/CNCl-3 model. However, considering the Zn-N₄/CNCl-1 and Zn-N₄/CNCl-2 were more stable than Zn-N₄/CNCl-3, Zn-N₄/CNCl-1 and Zn-N₄/CNCl-2 as dominant models were more easier formed rather than Zn-N₄/CNCl-3 model. Besides, after the protonation of coordinated N atom on Zn-N₄/CNCl-3 model (Zn-N₄/CNCl-3(H)), the energy barrier of RDS for CO₂RR on Zn-N₄/CNCl-3(H) was 0.65 eV, lower than that of direct CO₂RR (0.70 eV)

on Zn-N₄/CNCl-3 model. For Zn-N₄/CN model, both the energy barrier of RDS for direct CO₂RR and the protonation of coordinated N atom on Zn-N₄/CN model were 1.20 eV, indicating that there was no advantage for the protonation of coordinated N atom on Zn-N₄/CN model compared with direct CO₂RR pathway. Therefore, the protonation of coordinated N atom on Zn-N₄/CNCl models was more advantageous than their direct CO₂RR pathway.

Therefore, we considered the CO₂RR pathways on Zn-N₄/CNCl(H) models after the protonation of coordinated N atom. The optimized structures of *COOH and *CO intermediates on Zn-N₄/CNCl(H) models were shown in **Figure R19**. The protonation of coordinated N atom on Zn-N₄/CNCl(H) models induced the break of Zn-N bond and the self-reconstruction of Zn-N₄ site into Zn-N₃ site with broken planar-like symmetry during CO₂RR, which was also confirmed by in-situ EXAFS measurement.

Figure R19. The optimized structures of *COOH and *CO on Zn-N₄/CNCl-1(H), Zn-N₄/CNCl-2(H) and Zn-N₄/CNCl-3(H) models.

The energy changes for CO₂RR on Zn-N₄/CNCl(H) and Zn-N₄/CN models were shown in **Figure R20**. The energy barriers of RDS on Zn-N₄/CNCl-1(H), Zn-N₄/CNCl-2(H), Zn-N₄/CNCl-3(H) and Zn-N₄/CN models were 0.55 eV, 0.59 eV, 0.65 eV and 1.20 eV, respectively, revealing the self-reconstruction of Zn-N₄ site into Zn-N₃ site remarkably improved the CO₂RR activity. Compared with pristine Zn-N₄ site, after self-reconstruction of Zn-N₄ site into Zn-N₃ site, the low-coordinated Zn-N₃ site from Zn-N₄/CNCl-1(H), Zn-N₄/CNCl-2(H) and Zn-N₄/CNCl-3(H) effectively strengthened the adsorption of *COOH intermediate and accelerated the reaction rate, which was consistent with the experimental results.

Figure R20. The catalytic pathways for CO₂RR and corresponding energy changes on Zn-N₄/CNCl-1(H), Zn-N₄/CNCl-2(H), Zn-N₄/CNCl-3(H) and Zn-N₄/CN models.

Therefore, the catalytic pathways on Zn-N₄/CNCl models were summarized in **Figure R21**. The adjacent C-Cl bond induced the easier protonation of coordinated N atom of Zn-N₄/CNCl models. The protonation of coordinated N atom subsequently induced the self-reconstruction of Zn-N₄ site into highly active Zn-N₃ site, which remarkably strengthened the adsorption of *COOH intermediate and lowered the energy barriers of RDS.

Figure R21. The schematic illustration of catalytic mechanism on Zn-N₄/CNCl models for CO₂RR.

In summary, we revealed that the C-Cl bonds were indeed located in the vicinity of Zn species in Zn-SA/CNCl-1000 by AC-STEM measurement with EDX spectroscopy elemental mapping, EELS and DFT calculation. As confirmed by in-situ EXAFS measurement and DFT calculation, the adjacent C-Cl bonds induced the self-reconstruction of Zn-N₄ site into highly active Zn-N₃ site, which remarkably boosted their catalytic activity for CO₂RR.

Once again, we extend our sincere appreciation to the reviewer for the time, expertise, and constructive feedback, which have undoubtedly contributed to the improvement of our manuscript.

Reviewer #3:

Comment : In this revised manuscript, Wei et al. has conducted significant revision of the concerns raised by the reviewers which I highly appreciate. However, I am still little bit puzzled by their explanation of existence of only single atom as there are still some additional features (between 2.5 -3.5 Å) in the EXAFS regions of figure 20 of supplementary information. Keep in mind that EXAFS is less sensitive with distances. Hence, any features that you observe in the high radial distance might be low in intensity but can contribute significantly. The contribution from Cl cannot be ignored as there is clear evidence of presence of Cl from the XPS results. Following are some additional comments.

Reply: We sincerely appreciate the comments provided by Reviewer #3. We extend our sincere appreciation to the reviewer for the time, expertise, and constructive feedback, which have undoubtedly contributed to the improvement of our manuscript.

In the FT-EXAFS fitting result in R space of Zn-SA/CN-1000 (the previous Supplementary Fig. 20), the features beyond 2.0 Å was attributed to the Zn-C path from the second coordination shell of Zn-N-C due to the absence of Cl element in the Zn-SA/CN-1000 as reference sample.

For the metal single-atom anchored on N-doped carbon, the main contribution of the second coordination shell of metal was assigned to the metal-C path. For instance, the reference (*Nat. Catal.* **4**, 407-417 (2021).) reported the FeN₃P-SAzyme. The features at around 2.8 Å were attributed to the Fe-C path from the second coordination shell of Fe-N-C.

Supplementary Table 1. Fe K-edge EXAFS curve Fitting Parameters^a

sample	path	N	R (Å)	σ^2 (Å ²)	ΔE_0 (eV)	R , %
Fe foil ^b	Fe-Fe1	8	2.47	0.006	0.7	0.01
	Fe-Fe2	6	2.85	0.007		
FePc ^c	Fe-N	4	1.93	0.004	3.2	0.01
FeN ₃ P-SAzyme ^d	Fe-N/O	3.6	1.97	0.007	0.6	0.3
	Fe-P	1.1	2.26	0.008		
	Fe-C	4.3	2.80	0.021		
FeN ₄ -SAzyme ^d	Fe-N/O	5.8	2.00/	0.009	4.1	0.1
	Fe-C	4.2	2.76	0.040		

The Supplementary Table 1 from the reference (*Nat. Catal.* **4**, 407-417 (2021)).

As shown in **Figure R22**, we utilized the Zn-N₄/CN model to simulate the Zn-SA/CN-1000 without Cl-doping. The first coordination shell of Zn atom was four N atoms. The second coordination shell of Zn atom included eight C atoms, with four C atoms with shorter Zn-C distance (2.67 Å) marked by red circles and four C atoms with longer Zn-C distance (3.06 Å) marked by blue circles.

Figure R22. The optimized Zn-N₄/CN model to simulate the Zn-SA/CN-1000 without Cl-doping.

Therefore, we utilized two non-equivalent Zn-C paths to fit the second coordination shell of Zn atom in Zn-SA/CN-1000 without Cl-doping. The fitting results were shown in **Figure R23** and **Table R5**.

Figure R23. The FT-EXAFS fitting results of the first and second coordination shells of Zn atom in Zn-SA/CN-1000. **a**, The fitting result in k space. **b**, The fitting result in R space.

Table R5. The structural parameters extracted from the Zn K-edge for fitting the first and second coordination shells of Zn atom in Zn-SA/CN-1000. ($S_0^2 = 0.78$)

Sample	Shell	N	R(Å)	$\sigma^2(\text{Å}^2)$	$\Delta E_0(\text{eV})$	R factor
Zn-SA/CN-1000	Zn-N	4.00 ± 0.09	2.001 ± 0.003	0.0081 ± 0.0003	-2.77 ± 0.31	0.000887
	Zn-C-1	0.85 ± 0.53	2.495 ± 0.023	0.0161 ± 0.0033		
	Zn-C-2	1.03 ± 0.69	3.259 ± 0.049	0.0164 ± 0.0052		

As shown in **Table R5**, the coordination numbers of Zn-C1 and Zn-C2 were 0.85 ± 0.53 and 1.03 ± 0.69 , respectively, with larger range of error, revealing the second coordination shell of Zn atom in Zn-SA/CN-1000 was irregular. For example, during pyrolysis at 1000°C for 3 h, partial carbon atoms in the second coordination shell of Zn atom will evaporate with formation of carbon-vacancies. Therefore, the carbon atoms and carbon-vacancies randomly co-existed in the second coordination shell of Zn atom in Zn-SA/CN-1000, which led to the lower coordination numbers of Zn-C path. Considering the Zn-SA/CN-1000 lacked the periodic crystal structure like inorganic crystal and the chemical structure of the second coordination shell of Zn

atom was irregular, thus most reported articles mainly focused on fitting the first coordination shell of metal atom.

We added the **Figures R22-R23** as **Supplementary Figs. 24-25** and **Table R5** as **Supplementary Table 6** in the revised Supplementary Information and the revised Manuscript (Line 234-236).

***Comment 1:** (a) Did the authors performed any catalytic test with ZnO_x clusters over the same support?*

Reply: We sincerely appreciate the comments provided by Reviewer #3.

The ZnO_x/CN catalyst was synthesized by heating the Zn-SA/CN-1000 catalyst at 350°C for 1h in air with heating rate of 5°C/min. As shown in **Figure R24**, ZnO_x clusters appeared on the N-doped carbon substrate after heating Zn-SA/CN-1000 catalyst in air.

Figure R24. The HAADF-STEM image of ZnO_x/CN catalyst.

We also tested the catalytic performance of ZnO_x/CN catalyst for CO₂RR. As shown in **Figure R25**, the main product for CO₂RR catalyzed by ZnO_x/CN was H₂ rather than CO, ranging from -0.57 V to -0.93 V vs. RHE. The maximum faradaic efficiency of CO catalyzed by ZnO_x/CN was 43.31%, achieved at -0.75 V vs. RHE. These results revealed the poor catalytic activity of ZnO_x/CN catalyst for CO₂RR.

Figure R25. The faradaic efficiency of CO for CO₂RR catalyzed by ZnO_x/CN at different potentials.

We added the **Figures R24-R25** as **Supplementary Figs. 34-35** in the revised Supplementary Information and the revised Manuscript (Line 305-308).

Comment 2:

(b) Since the authors already have the DFT models, I suggest also fit of the second shell to make sure that there is no contribution of other component beyond 2 Å.

(c) The authors claim that the intensity of the wave in the range of 2-3 Å (red box b) was similar to that of the wave in the range of 0-0.75 Å. I agree that there is no bond that can be in the range 0-1 Å. However, as I said the intensity of the EXAFS decreases with radial distance. So, to compare between 0-1 Å and 1.5-2.5 Å is not the same. The features in figure 2 (d), (e) and (f) cannot be simply ignored.

As the novelty of the work lies on the restructuring of the Zn coordination, this is very important that the authors carefully make their conclusion to be published in such a prestigious journal.

Reply: We sincerely appreciate the comments provided by Reviewer #3.

As shown in **Figure R26**, we analyzed the coordinate structure of Zn atom of the DFT models in **Figure 5a**. As shown in **Figure R26a**, four N atoms located in the first coordination shell of Zn atom in Zn-N₄/CNCl-1 model. While eight C atoms located in the second coordination shell of Zn atom. There were two non-equivalent Zn-C paths with three shorter Zn-C distance marked in red circles and five longer Zn-C distance marked in blue circles. Theoretically, the Cl atom located in the third coordination shell of Zn atom with Zn-N-C-Cl structure.

As shown in **Figure R26b**, four N atoms located in the first coordination shell of Zn atom in Zn-N₄/CNCl-2 model. While eight C atoms located in the second coordination shell of Zn atom. There were two non-equivalent Zn-C paths with four shorter Zn-C distance marked in red circles and four longer Zn-C distance marked in blue circles. Theoretically, the Cl atom located in the fourth coordination shell of Zn atom with Zn-N-C-C-Cl structure.

As shown in **Figure R26c**, four N atoms located in the first coordination shell of Zn atom in Zn-N₄/CNCl-3 model. While eight C atoms located in the second coordination shell of Zn atom. There were two non-equivalent Zn-C paths with four shorter Zn-C distance marked in red circles and four longer Zn-C distance marked in blue circles. Theoretically, the Cl atom located in the third coordination shell of Zn

atom with Zn-N-C-Cl structure.

Figure R26. a-c, The analysis the coordinate structure of Zn atom in Zn-N₄/CNCl-1, Zn-N₄/CNCl-2 and Zn-N₄/CNCl-3 models, respectively.

Therefore, theoretically, only two non-equivalent Zn-C paths should be considered for fitting the second coordination shell of Zn atom in Zn-N₄/CNCl models without Zn-Cl path. However, considering the Zn-Cl distance was comparable to that of Zn-C paths, fitting the second coordination shell of Zn atom in Zn-N₄/CNCl models with both Zn-C path and Zn-Cl path was also achievable.

When considering both the Zn-C path and Zn-Cl path as the second coordination shell of Zn atom, the main contribution of the second coordination shell of Zn atom was from Zn-C path rather than Zn-Cl path since eight C atoms and only one Cl atom were in the second coordination shell of Zn atom. Therefore, simultaneously fitting the Zn-C path and Zn-Cl path in the second coordination shell of Zn atom is challenging because the contribution of Zn-C path is much stronger than that of Zn-Cl path. Thus, on the background of strong contribution of Zn-C path, obtaining the accurate

coordination number of Zn-Cl path is challenging because the contribution of Zn-C path can easily cover the contribution of Zn-Cl path.

Besides, the Zn-SA/CNCl catalysts lacked the periodic crystal structure like inorganic crystal and the chemical structure of the second coordination shell of Zn atom was irregular. The Cl atoms were randomly located in the second or higher coordination shells of Zn atom. Therefore, simultaneously fitting Zn-C and Zn-Cl paths as the second coordination shell of Zn atom and obtaining the accurate coordination information of Zn-Cl path, such as coordination number and Zn-Cl distance is challenging with larger errors.

Thus, only two non-equivalent Zn-C paths were considered as the second coordination shell of Zn atom. The fitting results were exhibited in **Figure R27** and **Table R6**.

Figure R27. The FT-EXAFS fitting results of the first and second coordination shells of Zn atom in Zn-SA/CNCl catalysts. **a-b**, The fitting result of Zn-SA/CNCl-850 in k space and R space, respectively. **c-d**, The fitting result of Zn-SA/CNCl-920 in k space and R space, respectively. **e-f**, The fitting result of Zn-SA/CNCl-1000 in k space and R space, respectively.

Table R6. The structural parameters extracted from the Zn K-edge for fitting the first and second coordination shells of Zn atom in Zn-SA/CNCl catalysts. ($S_0^2 = 0.78$)

Sample	Shell	N	R(Å)	$\sigma^2(\text{Å}^2)$	$\Delta E_0(\text{eV})$	R factor
Zn-SA/CNCl-850	Zn-N	4.02 ± 0.08	2.034 ± 0.002	0.0092 ± 0.0004	2.49 ± 0.21	0.002692
	Zn-C-1	1.60 ± 0.46	2.899 ± 0.025	0.0127 ± 0.0037		
	Zn-C-2	4.41 ± 0.62	3.271 ± 0.013	0.0129 ± 0.0018		
Zn-SA/CNCl-920	Zn-N	4.01 ± 0.09	2.021 ± 0.003	0.0090 ± 0.0002	0.97 ± 0.19	0.001147
	Zn-C-1	1.21 ± 0.35	2.909 ± 0.020	0.0103 ± 0.0029		
	Zn-C-2	3.89 ± 0.44	3.262 ± 0.010	0.0109 ± 0.0013		
Zn-SA/CNCl-1000	Zn-N	3.98 ± 0.09	2.040 ± 0.002	0.0100 ± 0.0003	3.54 ± 0.16	0.001284
	Zn-C-1	1.69 ± 0.35	2.887 ± 0.015	0.0122 ± 0.0026		
	Zn-C-2	4.45 ± 0.47	3.278 ± 0.009	0.0125 ± 0.0013		

We added the **Figures R26-R27** as **Supplementary Figs. 19-20** and **Table R6** as **Supplementary Table 5** in the revised Supplementary Information and the revised Manuscript (Line 185-187).

The above fitting results of the second coordination shell revealed that the main contribution of the second coordination shell was from Zn-C path, in both Zn-SA/CNCl-1000 and Zn-SA/CN-1000 without Cl-doping. However, they exhibited quite different structural evolution for CO₂RR, confirmed by in-situ XANES and EXAFS measurements. The Zn-SA/CNCl-1000 underwent the gradual transformation from Zn-N₄ site into low-coordinated Zn-N₃ site for CO₂RR. While the Zn-N₄ site of

Zn-SA/CN-1000 without Cl-doping had no obvious changes for CO₂RR. During CO₂RR, compared with Zn-SA/CN-1000 without Cl-doping, Zn-SA/CNCl-1000 exhibited much higher catalytic activity, with J_{CO} of 271.7 mA/cm² and TOF value of 29325 h⁻¹ at -0.93 V vs. RHE, which were 28.9 times and 82.8 times those of Zn-SA/CN-1000 without Cl-doping.

Therefore, we concluded that the unique structural evolution from Zn-N₄ site into Zn-N₃ site in Zn-SA/CNCl-1000 and the much higher catalytic activity for CO₂RR were attributable to the Cl-doping rather than the carbon atoms in the second coordination shell of Zn atom since the Zn-C path was the main contribution of the second coordination shell in both Zn-SA/CNCl-1000 and Zn-SA/CN-1000.

As shown in **Figure R13** and **Figure R14**, the in-situ XANES and EXAFS measurements revealed the introduction of Cl element in Zn-SA/CNCl-1000 effectively induced the structural evolution from Zn-N₄ site into Zn-N₃ site. Therefore, the much higher catalytic activity for CO₂RR catalyzed by Zn-SA/CNCl-1000 was attributable to the change of coordination number of Zn-N bond in the first coordination shell.

Figure R13. The in-situ XANES and EXAFS measurements of Zn-SA/CNCI-1000 and Zn-SA/CN-1000 at Zn-K edge. **a**, The XANES spectra for CO₂RR catalyzed by Zn-SA/CNCI-1000 at OCV, -0.7 V, -0.8 V, -0.9 V and -1.0 V vs. RHE. **b**, The corresponding FT-EXAFS spectra in R space catalyzed by Zn-SA/CNCI-1000 at OCV, -0.7 V, -0.8 V, -0.9 V and -1.0 V vs. RHE. **c**, The XANES spectra for CO₂RR catalyzed by Zn-SA/CN-1000 at OCV and -0.9 V vs. RHE. **d**, The corresponding FT-EXAFS spectra in R space catalyzed by Zn-SA/CN-1000 at OCV and -0.9 V vs. RHE.

Figure R14. The corresponding WT analysis of in-situ XANES and EXAFS measurements for CO₂RR catalyzed by Zn-SA/CNCl-1000 and Zn-SA/CN-1000 at Zn-K edge. **a-b**, The WT analysis of Zn-SA/CNCl-1000 at OCV and -0.9 V vs. RHE during in-situ CO₂RR, respectively. **c-d**, The WT analysis of Zn-SA/CN-1000 at OCV and -0.9 V vs. RHE during in-situ CO₂RR, respectively. The potentials are provided without iR correction.

As shown in **Figure R9** and **Figure R10**, we revealed that the C-Cl bonds were indeed located in the vicinity of Zn species in Zn-SA/CNCl-1000, confirmed by AC-STEM measurement with EDX spectroscopy elemental mapping and EELS. As shown in **Figure R21**, the DFT calculation revealed that the adjacent C-Cl bond induced the easier protonation of coordinated N atom of Zn-N₄/CNCl models. The protonation of coordinated N atom subsequently induced the self-reconstruction of Zn-N₄ site into highly active Zn-N₃ site, which remarkably strengthened the adsorption of *COOH intermediate and lowered the energy barriers of RDS.

Figure R21. The schematic illustration of catalytic mechanism on Zn-N₄/CNCI models for CO₂RR.

Once again, we extend our sincere appreciation to the reviewer for the time, expertise, and constructive feedback, which have undoubtedly contributed to the improvement of our manuscript.

Response to the Reviewers

Reviewer #2:

Comment : During the revision, authors have made the manuscript much more convincing, in particular, by adding Cl K-edge XAS data. I appreciate authors' efforts in improving the manuscript. I have just a few additional comments.

- 1) Please add Cl K-edge EXAFS data in k -space to Supplementary Information
- 2) Please provide fits of Cl K-edge EXAFS data that would confirm the presence of Cl-Zn bonds.
- 3) Please provide details of Cl K-edge XAS data processing in the Methods section. Also, currently it is stated in the Methods section that that only XANES data were collected, while in the manuscript EXAFS data are discussed as well.

Reply: We sincerely appreciate the comments provided by Reviewer #2. We extend our sincere appreciation to the reviewer for the time, expertise, and constructive feedback, which have undoubtedly contributed to the improvement of our manuscript.

As shown in **Figure R1** and **Table R1**, we provided the Cl K-edge EXAFS data in k space and the fitting results of Zn-SA/CNCl samples.

Figure R1. a-b, The fitting results of Zn-SA/CNCI-850 at Cl K-edge in R space and k space, respectively. c-d, The fitting results of Zn-SA/CNCI-920 at Cl K-edge in R space and k space, respectively. e-f, The fitting results of Zn-SA/CNCI-1000 at Cl K-edge in R space and k space, respectively.

Table R1. The structural parameters extracted from the Cl K-edge for fitting Zn-SA/CNCl samples.

Sample	Shell	N	R(Å)	$\sigma^2(10^{-3}\text{Å}^2)$	$\Delta E_0(\text{eV})$	R factor
Zn-SA/CNCl-850	Cl-C	0.47	1.67	2.12	-2.5	0.008
	Cl-Zn	0.55	2.38	1.51		
Zn-SA/CNCl-920	Cl-C	0.77	1.67	0.35	-4.4	0.017
	Cl-Zn	0.32	2.37	7.31		
Zn-SA/CNCl-1000	Cl-C	0.92	1.73	5.43	-0.4	0.025
	Cl-Zn	0.13	2.65	9.17		

Error bounds that characterize the structural parameters obtained by EXAFS spectroscopy were estimated as $N \pm 10\%$; $R \pm 1\%$; $\sigma^2 \pm 15\%$; $\Delta E_0 \pm 20\%$.

Zn-SA/CNCl-850 (k range:3.0-10.0 Å⁻¹; R range: 1.0-3.0 Å)

Zn-SA/CNCl-920 (k range:3.0-10.0 Å⁻¹; R range: 1.0-3.0 Å)

Zn-SA/CNCl-1000 (k range:3.0-10.0 Å⁻¹; R range: 1.0-3.0 Å)

The details of Cl K-edge XAS data processing was exhibited as follows:

For fitting the EXAFS data at Cl K-edge, k^3 weights, the k range (3.0-10.0 Å⁻¹), and R range (1.0-3.0 Å) were applied for Zn-SA/CNCl samples.

During the Wavelet Transform analysis, the $\chi(k)$ data format was imported into the Larch Python code. The parameters were listed as follows: R range was 1-5 Å; k range was 2-10 Å⁻¹; k -weight was 2; and the Morlet function ($\kappa=6$, $\sigma=1$) was used as the mother wavelet to provide the overall distribution.

We added the details of Cl K-edge XAS data processing in the Methods section in the revised Manuscript. (Line 549-554)

We revised the description of XANES and EXAFS measurement at Cl K-edge as follows: The XANES and EXAFS measurement at Cl K-edge was recorded at the 4B7A station in Beijing Synchrotron Radiation Facility in TEY mode. (Line 536-538 in the revised Manuscript)

We added the **Figure R1** and **Table R1** as **Supplementary Fig. 22** and **Supplementary Table 6** in the revised manuscript (Line 207-209) and Supplementary Information.

Once again, we extend our sincere appreciation to the reviewer for the time, expertise, and constructive feedback, which have undoubtedly contributed to the improvement of our manuscript.